# Feature Accompaniment: Is it Feasible to Learn Out-of-Distribution Generalizable Representations with In-Distribution Data?

## Abstract

Learning representations that generalize out-of-distribution (OOD) is critical for machine learning models to be deployed in the real world. However, despite the significant effort in the last decade, algorithmic advances in this direction have been limited. In this work, we seek to answer the fundamental question: is learning OOD generalizable representations with only in-distribution data really feasible? We first empirically show that perhaps surprisingly, even with an "oracle" representation learning objective that allows the model to *explicitly fit good representations* on the training set, the learned model still underperforms OOD in a wide range of distribution shift benchmarks. To explain the gap, we then formally study the OOD generalization of two-layer ReLU networks trained by stochastic gradient descent (SGD) in a structured setting, unveiling an unexplored OOD generalization failure mode that we refer to as *feature accompaniment*. We show that this failure mode essentially stems from the inductive biases of non-linear neural networks and fundamentally differs from the prevailing narrative of spurious correlations. Overall, our results imply that it may be generally not feasible to learn OOD generalizable representations without explicitly considering the inductive biases of SGD-trained neural networks and provide new insights into the OOD generalization failure, suggesting that OOD generalization in practice may behave *very differently* from existing theoretical models and explanations.

## 1 Introduction

Robustness to distribution shifts is a critical requirement for machine learning systems to be deployed in the wild (Amodei et al., 2016; Koh et al., 2021). In the last decade, it has proved that the conventional principle of empirical risk minimization (ERM), when combined with (deep) neural networks optimized by stochastic gradient descent (SGD), can lead to remarkable *in-distribution (ID) generalization* performance with sufficient training data. Unfortunately, this powerful paradigm can also fail catastrophically in *out-of-distribution (OOD) generalization* (Torralba & Efros, 2011; Beery et al., 2018; Geirhos et al., 2018; DeGrave et al., 2021), where the test data exhibits distribution shifts stemming from data variations that are not well-covered in training. Due to their ubiquity in the real world, distribution shifts have posed significant challenges to machine learning.

As a result, recent years have witnessed a surge of developing novel learning algorithms to train models that can generalize OOD. Nevertheless, the effectiveness of many of those algorithms has been called into question by several studies (Gulrajani & Lopez-Paz, 2021; Koh et al., 2021; Wiles et al., 2022), where no tested algorithm exhibits consistent and significant advantage compared with the most "vanilla" baseline ERM. On the other hand, an important observation made by recent work is that *increasing the diversity of training data*, either through pre-training or though special diverse data augmentation, often yields representations with significant improvement in OOD generalization (Taori et al., 2020; Hendrycks et al., 2021a; Wiles et al., 2022). For example, properly fine-tuning CLIP representations (Radford et al., 2021) has yielded state-of-the-art performance on various distribution shift benchmarks (Wortsman et al., 2022; Kumar et al., 2022), and it has been empirically observed that such distribution shift robustness heavily depends on the amount and diversity of pre-training data (Fang et al., 2022; Santurkar et al., 2023). This overarching trend leaves some important open questions to be answered, which motivates this work.

On the empirical side, a notable facet of OOD generalization revealed by recent work is that *data may outweigh algorithms*. However, increasing (pre-)training data also blurs the notion of "OOD" itself since it essentially expands the training distribution: for example, CLIP is trained using a dataset of 400 million image-text pairs, which is at least hundreds of times larger than any existing OOD generalization dataset. Therefore, given the somewhat pessimistic empirical results without more training data, a fundamental question arises regarding the ongoing pursuit of OOD generalization:

*Is it really feasible to learn OOD generalizable representations by training on only ID data, in particular, when (i) ID and OOD data do have structural similarities that enable generalization and (ii) ID data is informative enough for extracting such generalizable structures?*

On the theoretical side, a large body of work has been devoted to understanding and addressing the OOD generalization failure caused by *spurious correlations*, which represent the failure mode caused by the model using features that have non-causal relationships with desired outputs. These studies, however, do not give satisfying answers to the above question due to the following two reasons: (i) the majority of existing theory either only considers *linear models* such as linear classification over prescribed features or neural tangent kernels (NTKs) (Arjovsky et al., 2019; Sagawa et al., 2020b; Nagarajan et al., 2021; Xu et al., 2021; Ahuja et al., 2021b;a; Pezeshki et al., 2021; Chen et al., 2022; Wang et al., 2022; Rosenfeld et al., 2022; Abbe et al., 2023), or also considers non-linear models but is *optimization-independent* (Rosenfeld et al., 2021; Kamath et al., 2021; Ye et al., 2021). Hence, these results may fail to capture the inductive biases of the most widely used model class in practice, i.e., *overparametrized non-linear neural networks*, for which it is well-known that the implicit biases of SGD optimization is vital to generalization (Zhang et al., 2017). (ii) As we will show in Section 2, the viewpoint of spurious correlations itself is unable to explain some important observations in OOD generalization. Indeed, it has been shown that many OOD generalization algorithms that enjoy provable guarantees in their specific settings do not excel in real-world benchmarks (Gulrajani & Lopez-Paz, 2021; Koh et al., 2021). Motivated by the gap between theory and practice, we argue that taking into account the inductive biases of non-linear neural networks and SGD may be not only important but also *necessary* for OOD generalization.

## 1.1 SUMMARY OF OUR RESULTS

In this work, we take steps toward formally answering the above question:

**Empirically,** we show on 8 common distribution shift datasets that, perhaps surprisingly, even with an "oracle" representation learning objective that *allows the model to explicitly fit OOD generalizable pre-trained representations on the training set*, the learned representaions still perform *much worse* in OOD generalization than their pre-trained counterparts. This indicates that it may be generally **not feasible** to learn OOD generalizable representations without explicitly taking into account the inductive biases of non-linear neural networks in many existing benchmarks. Our results challenge the common belief in the community that the empirically observed OOD failure in existing benchmarks is mainly caused by spurious correlations, suggesting a large OOD generalization gap that *cannot* be explained by spurious correlations or other existing explanations of OOD failure.

**Theoretically,** we prove that in certain binary classification tasks where the data is generated from OOD generalizable *core features* and other *background features* (formally defined in Section 3), a randomly initialized *two-layer ReLU neural network* trained by SGD can achieve good ID generalization given sufficient data and SGD iterations, yet still fails to generalize OOD. We also show that *OOD generalization using neural networks with non-linear activation can be **provably different** from linear models*, which allows us to draw several new conclusions in OOD generalization. Notably, we demonstrate that the above OOD generalization failure differs fundamentally from those in prior work as it holds even when (i) background features are not correlated with the label at all (nullifying the possibility of spurious correlations), (ii) ground-truth labels are perfectly determined by core features (nullifying the possibility of lacking training data or informative features), and (iii) core and background features are not correlated (nullifying the possibility of the innate non-linear entanglement or correlation between core and background features).

Instead, we theoretically prove that at the core of this OOD failure is an unexplored *feature learning proclivity* of non-linear neural networks that we refer to as **feature accompaniment**. In brief, feature accompaniment refers to the process where during the learning of core features, SGD-trained neural networks also provably learn a portion of background features simultaneously—even when

background features have *no correlation* with the label and in the presence of weight decay regularization. We formally show that the reason for this phenomenon is that *neurons in the network tend to have **asymmetric activation** for examples from different classes* during training, resulting in non-zero gradient projections onto the span of background features. This further causes the accumulation of background features in the weights of the network during SGD, leading to large OOD risk under the distribution shifts on background features due to the non-linear coupling of core and background features in the neurons' activation. We provide more detailed explanations of why feature accompaniment happens in Section 4, and present empirical evidence suggesting that this explanation matches the observed OOD failure in our experiments. At a high level, we believe that our theoretical finding of feature accompaniment as a **novel inductive bias of SGD-trained neural networks** can also serve as a new perspective for understanding the existing success and shortcomings of deep learning by characterizing its learned features, complementing known inductive biases such as the simplicity bias of neural networks (Arpit et al., 2017; Shah et al., 2020; Pezeshki et al., 2021).

## 2 OOD Generalizable Representations are Hard to Learn Even When Explicitly Given on ID Data

We begin our analysis by an experiment motivated by recent algorithmic explorations in OOD generalization: existing algorithms have made various attempts to learn good representations that generalizes OOD by designing auxiliary objectives beyond the original training objective of minimizing the empirical prediction risk (see Gulrajani & Lopez-Paz (2021); Koh et al. (2021) and Section E for some examples). However, a major downside of those objectives is that their minimizers may not readily lead to ideal representations. Indeed, many critisisms on those objectives construct hard instances where incorporating spurious features can induce even smaller training risk than the ideal representation that only extracts generalizable features (Kamath et al., 2021; Rosenfeld et al., 2021). Therefore, the limited empirical success of existing objectives does not nullify the possibility that good representations may be learned with "better" objectives. Motivated by this, we would like to ask: *what is the very best we can get by representation learning given a finite training set?*

**An "oracle" representation learning objective.** In this work, we empirically approach the above question by introducing an "oracle" representation learning objective that allows the network to *explicitly fit given good representations* that can generalize OOD. Note that without further prior knowledge on the inductive biases of the model or the task, this is already the *best* objective we can possibly define since its minimizer can *uniquely* recover the "right" representations for all examples in the training set. We implement this idea by leveraging large-scale pre-trained models such as CLIP, whose representations have shown remarkable robustness under distribution shifts (Radford et al., 2021). Concretely, given a pre-trained "teacher" encoder, we randomly initialize another "student" encoder with an *identical architecture* to the teacher encoder. We then train the student encoder by minimizing the Euclidean distance between its output representations and the representations extracted by the teacher encoder on the training set, a process also known as *representation distillation*. The main difference between our work and existing work on representation distillation is that we focus on OOD generalization with teacher and student models sharing the same architecture, while existing work mainly considers model compression and knowledge transfer between a teacher model and a smaller student model under ID evaluations (Hinton et al., 2014; Tian et al., 2020). In total, our experiments span 8 distribution shift datasets that are extensively benchmarked by the community, including 5 large-scale ImageNet-based natural distribution shift datasets (Taori et al., 2020), 2 in-the-wild distribution shift datasets from WILDS (Koh et al., 2021), and a domain generalization dataset DomainNet (Peng et al., 2019). In those experiments, we employed fully-fledged neural network architectures including Vision Transformers (ViTs) (Dosovitskiy et al., 2021) and ResNets (He et al., 2016). Details of our experiments are provided in Section E.

**Evaluation protocol.** We evaluate the ID and OOD performance of the pre-trained and distilled representations by training linear probes on top of the representations on the ID training set and then evaluate those linear probes on both ID and OOD test sets. Note that under our protocol, the trained linear probes still need to complete an *OOD generalization task* on the OOD test sets, albeit with representations instead of raw image pixels as input. To compare the OOD generalization ability of different models, we follow the evaluation protocol of *effective robustness* proposed by Taori et al. (2020), which quantifies a model's distribution shift robustness as its OOD performance advantage over a baseline representing standard models trained on ID data. We follow Taori et al. (2020);

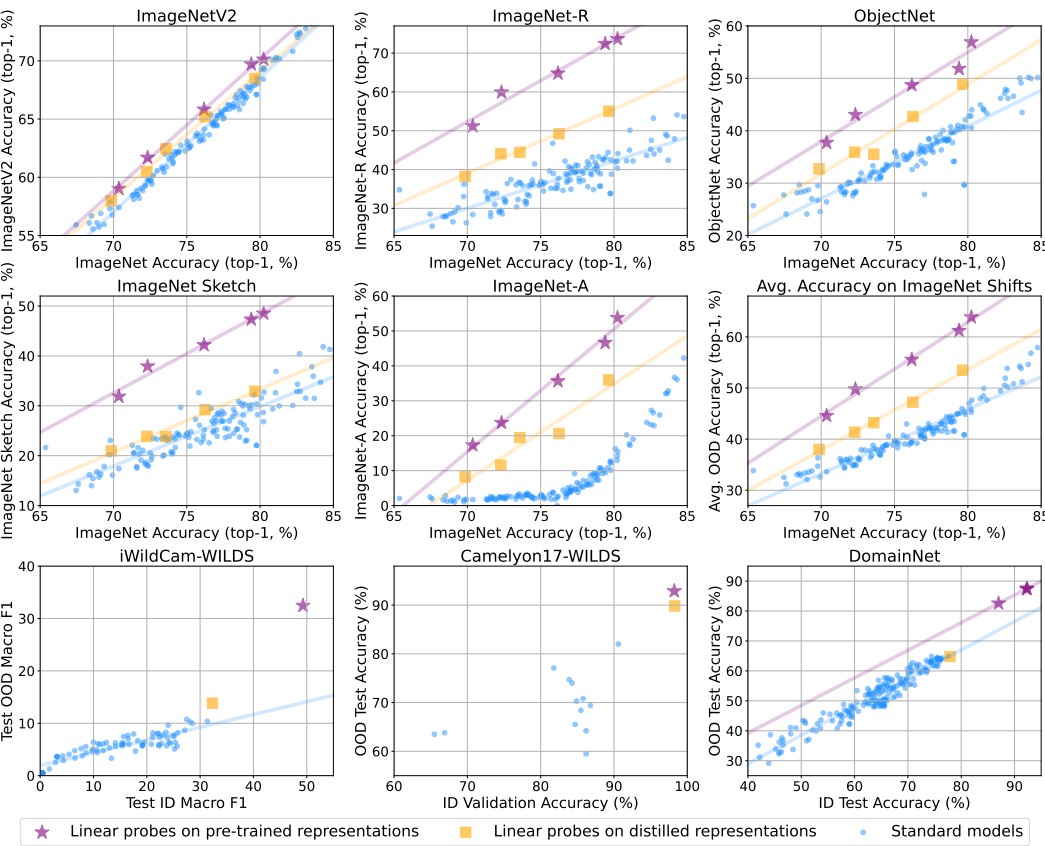

Figure 1: OOD performance ($y$-axes) v.s. ID performance ($x$-axes) for three model families including (i) linear probes on pre-trained representations (purple stars), (ii) linear probes on the representations distilled on the training set (orange squares), and (iii) standard models trained on the training set (blue circles). The $y$-axis of the 6-th panel stands for the average accuracy on ImageNet-based OOD test sets, averaged from the first 5 panels. See Section E for more experimental details.

Miller et al. (2021) and illustrate the effective robustness of the models using scatter plots, with their $x$-axes representing ID performance and $y$-axes representing OOD performance.

**Results.** As shown in Figure 1, linear probes on distilled representations exhibit consistent OOD generalization improvements compared to standard models especially for large datasets such as ImageNet.[1] This is not surprising since our "oracle" distillation objective uses additional representation-level supervision that standard models have no access to. However, we also observe that even with such supervision, *the OOD generalization ability of distilled representations still lags far behind compared to pre-trained representations.* For example, distilled representations only close about half of the effective robustness gap between standard models and pre-trained representations on average in ImageNet-based datasets, with even worse performance on some datasets with fewer data such as iWildCam and DomainNet. Given the fact that the representation learning objective itself cannot be further improved in general, our result implies that **OOD generalizable representations may not be learnable using only ID data without explicitly taking into account the inductive biases of the model or the task**. This is consistent with existing observations that even a standard ERM often remains strong in OOD generalization (Gulrajani & Lopez-Paz, 2021; Koh et al., 2021).

**Why does the distilled model still underperform OOD?** We first argue that this failure mode is *not likely due to spurious correlations* since we do *not* use any label in representation distillation—unlike the scenario where the model picks up spurious features due to their correlations with the

---

[1]The main exception is on ImageNet Sketch where some standard models surpass the linear fit of the results of distilled representations. By manually checking those models, we found that they use intensive, custom data augmentations (Hendrycks et al., 2020; 2021a), which essentially lead to a more diverse training distribution.

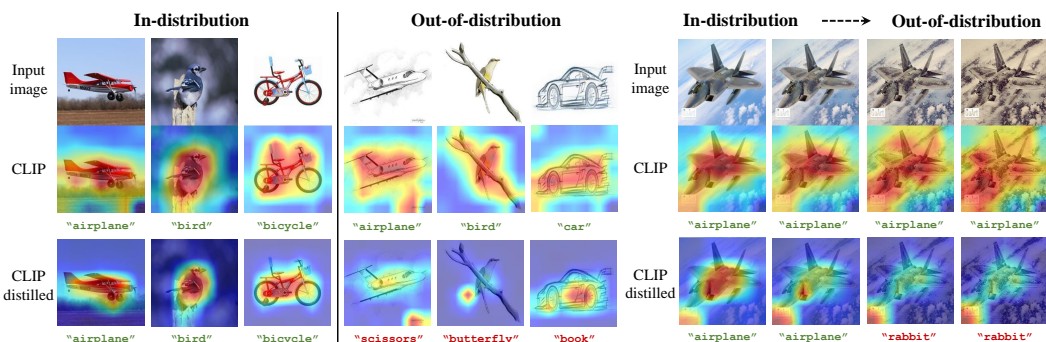

Figure 2: Prediction heatmaps for the linear probes on CLIP representations and distilled representations. **Left:** images from "real" (ID) and "sketch" (OOD) domains in DomainNet. **Right:** synthetic image style transfer simulating gradual distribution shifts. While the core objects are focused in ID images by the distilled model, their importance are gradually weakened under distribution shifts.

label. Also, due to our experimental protocol, our results can neither be explained by CLIP extracting richer representations of ID data (Zhang et al., 2022; Zhang & Bottou, 2023) (since representation distillation will also learn those rich representations), nor by CLIP extracting more "OOD core features" that are absent in ID data (since the linear probe *ID-trained* on top of CLIP representations cannot leverage them to achieve OOD generalization). To further understand this failure, we visualize the prediction heatmaps (using Grad-CAM++ (Chattopadhay et al., 2018)) of our models in DomainNet and a synthetic "gradual" distribution shift scenario based on image style transfer, as shown in Figure 2. An intriguing phenomenon revealed by the visualization is that while the distilled model indeed correctly focuses on the core objects for ID images, its attention to the core objects is gradually *weakened* under distribution shifts, resulting in OOD failure. Being unaware of any theoretical result that can explain this "weakening" phenomenon, we argue that our observations suggest **the existence of a OOD generalization failure mode that is beyond the reach of existing OOD generalization theory** and is likely to be tied to the non-linear feature learning process of neural networks. In the following sections, we will formally prove that a novel OOD generalization failure mode, which we refer to as feature accompaniment, indeed exists in certain binary classification tasks with two-layer ReLU networks and has strong connections to our empirical observations.

## 3 THEORETICAL MODEL ON OOD GENERALIZATION

**Notation.** We use $[d]$ to denote the set $\{1, \ldots, d\}$ for positive integers $d$, $\boldsymbol{I}_d$ to denote the $d \times d$ identity matrix, and $\mathcal{N}(\boldsymbol{\mu}, \boldsymbol{\Sigma})$ to denote the Gaussian distribution with mean $\boldsymbol{\mu}$ and covariance $\boldsymbol{\Sigma}$. For a set $\mathcal{S}$, we denote its cardinality by $|\mathcal{S}|$. For a vector $\mathbf{u}$, we denote its $\ell_2$ norm by $\|\mathbf{u}\|_2$. We denote the inner product of two vectors $\mathbf{u}$ and $\mathbf{v}$ by $\langle \mathbf{u}, \mathbf{v} \rangle$. We use the standard big-O notation: $O(\cdot), \Omega(\cdot), \Theta(\cdot), o(\cdot)$, as well as their soft-O variants such as $\widetilde{\Theta}(\cdot)$ to hide logarithmic factors. For some parameter $d$, we use $\mathrm{poly}(d)$ to denote $\Theta(d^C)$ with some unspecified large constant $C$.

### 3.1 OOD GENERALIZATION PROBLEM AND DATA GENERATION MODEL

We consider a binary classification setting with an input space $\mathcal{X} \subseteq \mathbb{R}^d$, a label space $\mathcal{Y} = \{-1, 1\}$, a model class $\mathcal{H} : \mathcal{X} \to \mathbb{R}$, and a loss function $\ell : \mathcal{Y} \times \mathcal{Y} \to \mathbb{R}$. For every distribution $\mathcal{D}$ over $\mathcal{X} \times \mathcal{Y}$ and model $h \in \mathcal{H}$, the expected risk of $h \in \mathcal{H}$ on $\mathcal{D}$ is given by $\mathcal{R}_{\mathcal{D}}(h) := \mathbb{E}_{(\mathbf{x}, \mathbf{y}) \sim \mathcal{D}} \ell(h(\mathbf{x}), \mathbf{y})$. We consider an OOD generalization regime where there are a set of distributions $\mathbb{D}$ consisting of all distributions to which we would like our model to generalize. Training examples are drawn from a set of training distributions $\mathbb{D}_{\mathrm{train}} \subsetneq \mathbb{D}$, where $\mathbb{D}_{\mathrm{train}}$ may contain one or multiple distributions available. Following the most common objective studied by prior work (Arjovsky et al., 2019; Sagawa et al., 2020a; Nagarajan et al., 2021; Rosenfeld et al., 2021; Weber et al., 2022), we aim to select a model $h \in \mathcal{H}$ to minimize the *OOD risk* defined as the worst-case expected risk on $\mathbb{D}$:

$$\mathcal{R}_{\mathrm{OOD}}(h) := \max_{\mathcal{D} \in \mathbb{D}} \mathcal{R}_{\mathcal{D}}(h). \tag{1}$$

It is clear that without assumptions on $\mathbb{D}_{\mathrm{train}}$ and $\mathbb{D}$, OOD generalization is impossible since no model can generalize to an arbitrary distribution. Fortunately, real-world distribution shifts are often

*structured* with some structural similarities shared by different distributions. We can thus hope that such structures can be captured by certain algorithms, leading to models that generalize OOD.

To formalize this, in this work we assume that both ID and OOD data are generated by a dictionary $M = (m_1, \ldots, m_{d_0}) \in \mathbb{R}^{d \times d_0}$ consisting of $d_0$ features with each feature $m_i \in \mathbb{R}^d$. Throughout the paper, we work with the case where $d_0$ is sufficiently large and $d \in [\Omega(d_0^{2.5}), \text{poly}(d_0)]$. For simplicity, we assume that every feature has $\ell_2$ norm $\|m_i\|_2 = 1$ and different features are orthogonal: $\forall i \neq j \in [d_0], \langle m_i, m_j \rangle = 0$, although our results can also be extended to more general cases.[2]

Among all features in $M$, we assume that there are $d_{\text{core}}$ features consistently correlating with the label in all distributions from $\mathbb{D}$. We denote the index set of those features by $\mathcal{S}_{\text{core}} \subsetneq [d_0]$ and refer to them as ***core features*** since they are predictive of the label regardless of distribution shifts. We will refer to the remaining features as ***background features*** and denote their index set by $\mathcal{S}_{\text{bg}} = [d_0] \setminus \mathcal{S}_{\text{core}}$ with $d_{\text{bg}} := |\mathcal{S}_{\text{bg}}| = d_0 - d_{\text{core}}$. We assume that $d_{\text{core}} = \Theta(d_0)$ and $d_{\text{bg}} = \Theta(d_0)$. With the above definitions, we now introduce the concrete ID and OOD data generation process.

**Definition 1** (ID and OOD data generation). *Consider an OOD generalization problem with a training distribution (ID data distribution) $\mathcal{D}_{\text{train}} \in \mathbb{D}_{\text{train}}$ and a test distribution (OOD data distribution) $\mathcal{D}_{\text{test}} \in \mathbb{D} \setminus \mathbb{D}_{\text{train}}$.[3] Each example $(\mathbf{x}, \mathbf{y}) \sim \mathcal{D} \in \{\mathcal{D}_{\text{train}}, \mathcal{D}_{\text{test}}\}$ is generated as follows:*

1. *Sample a label $\mathbf{y}$ from the uniform distribution over $\mathcal{Y}$.*

2. *Sample a weight vector $\mathbf{z} = (\mathbf{z}_1, \ldots, \mathbf{z}_{d_0}) \in \mathcal{Z} \subseteq \mathbb{R}^{d_0}$ where different coordinates of $\mathbf{z}$ are independent random variables generated via the following process:*

   - **ID data** *($\mathcal{D} = \mathcal{D}_{\text{train}}$): for every $j \in [d_0]$, sample $\mathbf{z}_j$ from some distribution $\mathcal{D}_j$ over $[0, 1]$ such that its moments satisfy $\mu_{jp} := \mathbb{E}_{\mathcal{D}_j} \mathbf{z}_j^p = \Theta(1)$ for $p \in [3]$, and the expected total weight of core features is not less than background features: $\sum_{j \in \mathcal{S}_{\text{core}}} \mu_{j1}^2 - \sum_{j \in \mathcal{S}_{\text{bg}}} \mu_{j1}^2 \geq 0$.*
   - **OOD data** *($\mathcal{D} = \mathcal{D}_{\text{test}}$): for every $j \in [d_0]$, if $j \in \mathcal{S}_{\text{core}}$, sample $\mathbf{z}_j$ from $\mathcal{D}_j$ over $[0, 1]$; if $j \in \mathcal{S}_{\text{bg}}$, sample $\mathbf{z}_j$ from some distribution $\mathcal{D}'_j$ over $[-1, 0]$ such that $\mathbb{E}_{\mathcal{D}'_j} \mathbf{z}_j = -\Theta(1)$.*

3. *Generate $\mathbf{x} = \sum_{j \in \mathcal{S}_{\text{core}}} \mathbf{y} \mathbf{z}_j m_j + \sum_{j \in \mathcal{S}_{\text{bg}}} \mathbf{z}_j m_j$.*

**Remarks on data generation.** Our data generation process formalize a natural OOD generalization setting that reflects several important aspects of real-world OOD generalization problems:

- The explicit separation of core and background features captures structural assumptions ensuring that OOD generalization is realistic: under the distribution shifts on background features, there exists a set of core features that enable robust classification. Hence, a model that is insensitive to background features and retains core features can generalize OOD. This rules out the ill-posed cases where the ID data is not informative enough to learn a generalizable model (Tripuraneni et al., 2020; Xu et al., 2021; Kumar et al., 2022) and is also the key intuition of many OOD generalization algorithms aiming to learn invariances (Gulrajani & Lopez-Paz, 2021).

- The weights of background features are assumed to be independent of the label in generation, rendering background features and labels *uncorrelated*. This fundamentally differs from prior OOD generalization analysis (Arjovsky et al., 2019; Sagawa et al., 2020b; Nagarajan et al., 2021; Rosenfeld et al., 2021) where it is assumed that non-core features are spuriously correlated with the label during training and hence may be used by the model.

**Connection to experiments.** In the prediction heatmap visualization of our models (Figure 2), we include a synthetic OOD scenario based on image style transfer, which *closely matches our data model in Definition 1* (keeping core features intact while *only* changing background features). As shown in our visualization, the prediction heatmaps on synthetic OOD images exhibit visually similar "weakening" patterns as in those of natural OOD images, suggesting that our data model can indeed capture important characteristics of real-world data with distribution shifts.

### 3.2 MODEL AND TRAINING

We consider a model class $\mathcal{H}$ representing width-$m$ two-layer neural networks with ReLU activation. Formally, given hidden-layer weights $\mathbf{W} = (\mathbf{w}_1, \ldots, \mathbf{w}_m) \in \mathbb{R}^{d \times m}$ and output-layer weights $\mathbf{a} =$

---

[2]Another advantage of assuming orthogonal features is that this nullifies the possibility of the network learning background features due to its correlation with core features (as we will define in the next paragraph).

[3]Note that $\mathcal{D}_{\text{train}}$ and $\mathcal{D}_{\text{test}}$ can also be (weighted) *mixtures* of multiple distributions in $\mathbb{D}_{\text{train}}$ and $\mathbb{D} \setminus \mathbb{D}_{\text{train}}$.

$(a_1, \ldots, a_m)^\top \in \mathbb{R}^m$, the output of a model $h : \mathbb{R}^d \to \mathbb{R}$ given an input $\mathbf{x} \in \mathcal{X}$ is

$$h(\mathbf{x}) = \sum_{k \in [m]} a_k \cdot \mathrm{ReLU}(\langle \mathbf{w}_k, \mathbf{x} \rangle), \tag{2}$$

where $\mathrm{ReLU}(u) = \max\{u, 0\}, u \in \mathbb{R}$. Similar to practical design choices, we consider an overpa-rameterized setting where $m \in [\Theta(d_0), \Theta(d)]$ and each weight vector $\mathbf{w}_k, k \in [m]$ is independently initialized by sampling $\mathbf{w}_i^{(0)} \sim \mathcal{N}(\mathbf{0}, \sigma_0^2 \mathbf{I}_d)$ with $\sigma_0^2 = \frac{1}{d}$. We randomly initialize output-layer weights $\mathbf{a}$ by sampling $a_k \sim \mathrm{Uniform}\{-\frac{1}{m}, \frac{1}{m}\}$ independently for each $k \in [m]$. To simplify our analysis, we keep output-layer weights $\mathbf{a}$ fixed throughout training, which is a common assumption in analyzing the optimization and generalization of two-layer neural networks in ID settings (Allen-Zhu & Li, 2021; Karp et al., 2021; Wen & Li, 2021; Allen-Zhu & Li, 2023).

We train the network using SGD over the standard hinge loss $\ell(y, y') = \max\{1 - yy', 0\}$ with step size $\eta > 0$ for $T$ iterations. We consider the most common $\ell_2$ weight decay with strength $\lambda = O(\frac{d_0}{m^{1.01}})$ for regularization. At each iteration $t \in \{0, \ldots, T-1\}$, we sample a batch of examples $\{(\mathbf{x}_i^{(t)}, \mathbf{y}_i^{(t)})\}_{i \in [N]} \sim \mathcal{D}_{\mathrm{train}}^N$ with batch size $N = \mathrm{poly}(d)$. The empirical loss is then

$$\widehat{\mathcal{L}}(h^{(t)}) = \frac{1}{N} \sum_{i \in [N]} \ell\left(h^{(t)}(\mathbf{x}_i^{(t)}), \mathbf{y}_i^{(t)}\right) + \frac{\lambda}{2} \sum_{k \in [m]} \left\|\mathbf{w}_k^{(t)}\right\|_2^2, \tag{3}$$

where we use $h^{(t)}$ to denote the model parameterized by weights $\mathbf{W}^{(t)} = (\mathbf{w}_1^{(t)}, \ldots, \mathbf{w}_m^{(t)})$ at iteration $t$. The corresponding SGD update for each weight vector $\mathbf{w}_k, k \in [m]$ is then given by

$$\mathbf{w}_k^{(t+1)} = \mathbf{w}_k^{(t)} - \eta \nabla_{\mathbf{w}_k^{(t)}} \widehat{\mathcal{L}}(h^{(t)}) = (1 - \eta\lambda)\mathbf{w}_k^{(t)} - \eta \nabla_{\mathbf{w}_k^{(t)}} \frac{1}{N} \sum_{i \in [N]} \ell\left(h^{(t)}(\mathbf{x}_i^{(t)}), \mathbf{y}_i^{(t)}\right). \tag{4}$$

## 4 MAIN THEORETICAL RESULTS

In this section, we present our main theoretical results.

**Technical challenges.** As we have discussed in Section 1, most of existing theoretical work on OOD generalization *separates generalization and optimization*, studying the *global minimizers* of their training objecives without considering exact optimization dynamics. By contrast, our setup requires an explicit analysis on the optimization trajectory of SGD, which is known to be challenging due to its *non-convex* and *non-linear* nature. Prior work has studied fine-tuning pre-trained models for OOD generalization using two-layer *linear networks* (Kumar et al., 2022; Lee et al., 2023). However, analyzing non-linear networks further requires a careful treatment on the activation property of the neurons, which results in SGD dynamics that fundamentally deviate from linear networks.

**Our approach.** At a high level, our analysis is based on the construction of two neuron subsets with cardinality $\Theta(m)$ that are randomly initialized to have large enough expected correlations with the examples from the two classes (i.e., "winning the lottery tickets" (Frankle & Carbin, 2019; Allen-Zhu & Li, 2021)). Based on this construction, we apply Berry-Esseen theorem to bound the activation probabilities of the ReLU functions for the neurons in the constructed subsets, iteratively tracking their gradient updates throughout training. This treatment allows us to characterize the output of the network up to constant factors while avoiding the nuisance of analyzing the activation probability of every single neuron in the network, which turns out to be very challenging due to fine-grained SGD dynamics. For ease of presentation, in the sequel we separate our main results into four parts and introduce them progressively. The proofs of all theorems are deferred to Appendix I.

**4.1. Neuron activation is asymmetric.** Our key insight is that during training, every neuron in the network has the incentive to be positively correlated with examples from *at most one class* $\mathbf{y}_{\mathrm{pos}}$ (depending on the random initialization of the neuron); we refer to those examples as *positive examples* $(\mathbf{x}_{\mathrm{pos}}, \mathbf{y}_{\mathrm{pos}}) \sim \mathcal{D}_{\mathrm{train}}|\mathbf{y} = \mathbf{y}_{\mathrm{pos}}$ for this neuron. Correspondingly, we refer to examples from the other class $\mathbf{y}_{\mathrm{neg}}$ as *negative examples* $(\mathbf{x}_{\mathrm{neg}}, \mathbf{y}_{\mathrm{neg}}) \sim \mathcal{D}_{\mathrm{train}}|\mathbf{y} = \mathbf{y}_{\mathrm{neg}}$. We can then show that during SGD, at least $\Theta(m)$ neurons will accumulate (in expectation) positive correlations with $\mathbf{x}_{\mathrm{pos}}$ and negative correlations with $\mathbf{x}_{\mathrm{neg}}$.[4] Since ReLU only activates for positive inputs, the activation probability of those neurons would become much larger for $\mathbf{x}_{\mathrm{pos}}$ than for $\mathbf{x}_{\mathrm{neg}}$, which we refer to as *activation asymmetry* and formally demonstrate by the following theorem.

---

[4]We note that activation asymmetry also emerges in more general settings beyond our current data model such as when different classes have distinct subsets of core features. The essential reason for this is that a neuron that has positive correlations with both classes is not informative enough for classification.

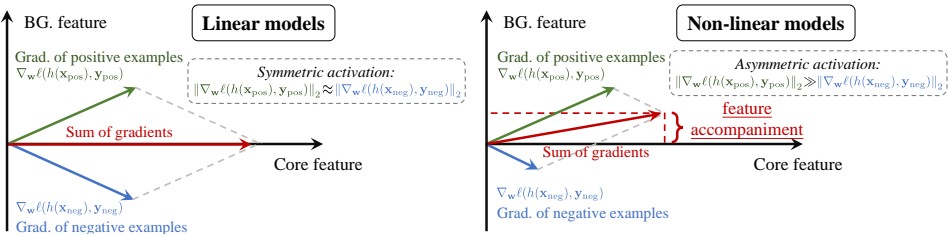

Figure 3: A diagram of feature accompaniment in non-linear models: activation asymmetry leads to non-cancelling gradient projections onto background features, resulting in their accumulation.

**Theorem 1** (Activation asymmetry). *For every $\eta \leq \frac{1}{\text{poly}(d_0)}$ and every $y \in \mathcal{Y}$, there exists $T_0 = \widetilde{\Theta}(\frac{m}{\eta\sqrt{d}})$ such that with high probability, there are $\Theta(m)$ neurons whose weights $\mathbf{w}_k$ satisfy*

$$\mathbf{Pr}_{\mathbf{x}|\mathbf{y}=y\sim\mathcal{D}_{\text{train}}}[\langle\mathbf{w}_k^{(t)},\mathbf{x}\rangle \geq 0] = 1 - o(1), \quad \mathbf{Pr}_{\mathbf{x}|\mathbf{y}=-y}[\langle\mathbf{w}_k^{(t)},\mathbf{x}\rangle \geq 0] = o(1), \quad \forall t \geq T_0. \quad (5)$$

**4.2. Activation asymmetry leads to feature accompaniment.** We first note that for every $k \in [m]$, the weight vector of the $k$-th neuron (in what follows we will also refer to it as the *learned feature* of the neuron) after $t$ iterations can be equivalently written as

$$\mathbf{w}_k^{(t)} = \sum_{j\in\mathcal{S}_{\text{core}}}\langle\mathbf{w}_k^{(t)},\boldsymbol{m}_j\rangle\boldsymbol{m}_j + \sum_{j\in\mathcal{S}_{\text{bg}}}\langle\mathbf{w}_k^{(t)},\boldsymbol{m}_j\rangle\boldsymbol{m}_j + \text{residual}, \quad (6)$$

where $\langle\text{residual},\boldsymbol{m}_j\rangle = 0$ for every $j \in [d_0]$ and thus can be neglected. Intuitively, Eq. (6) indicates that the learned feature can be decomposed into its projections onto different feature vectors. Meanwhile, as we will prove in Section A, the gradient projection onto background features is

$$\langle-\nabla_{\mathbf{w}_k^{(t)}}\widehat{\mathcal{L}}(h^{(t)}),\boldsymbol{m}_j\rangle \approx \frac{1}{m}\mathbb{E}_{(\mathbf{x},\mathbf{y})\sim\mathcal{D}_{\text{train}}}(\mathbb{1}_{\mathbf{y}=y} - \mathbb{1}_{\mathbf{y}=-y})\cdot\mathbb{1}_{\langle\mathbf{w}_k^{(t)},\mathbf{x}\rangle\geq 0}\mathbf{z}_j, \quad \forall j \in \mathcal{S}_{\text{bg}}, \forall t \quad (7)$$

for some $y \in \mathcal{Y}$, where we omit the weight decay term here for simplicity. By Theorem 1, we have for at least $\Theta(m)$ neurons, $\mathbb{E}_{\mathbf{x}|\mathbf{y}=y}\mathbb{1}_{\langle\mathbf{w}_k^{(t)},\mathbf{x}\rangle\geq 0}$ would be much larger than $\mathbb{E}_{\mathbf{x}|\mathbf{y}=-y}\mathbb{1}_{\langle\mathbf{w}_k^{(t)},\mathbf{x}\rangle\geq 0}$, resulting in a quite positive gradient projection to every background feature $\boldsymbol{m}_j, j \in \mathcal{S}_{\text{bg}}$ *regardless of its correlation with the label*. We refer to this phenomenon as **feature accompaniment** and illustrate it in Figure 3. Formally, we show that such accumulation would result in learned features containing both core features and some "coupled" background features after enough SGD iterations.

**Theorem 2** (Learned features). *For every $\eta \leq \frac{1}{\text{poly}(d_0)}$ and every $y \in \mathcal{Y}$, after $T_1 = \Theta(\frac{m}{\eta d_0})$ iterations with high probability, there are $\Theta(m)$ neurons whose weights $\mathbf{w}_k^{(T_1)}$ satisfy*

$$\sum_{j\in\mathcal{S}_{\text{core}}}\mu_{j1}\langle\mathbf{w}_k^{(T_1)},\boldsymbol{m}_j\rangle = y\cdot\Theta(1), \quad \sum_{j\in\mathcal{S}_{\text{bg}}}\mu_{j1}\langle\mathbf{w}_k^{(T_1)},\boldsymbol{m}_j\rangle = \Theta(1). \quad (8)$$

**4.3. Feature accompaniment has negligible impact on ID risk, yet causes large OOD risk.** Our next theorem characterizes the impact of feature accompaniment on both ID and OOD risks.

**Theorem 3** (ID and OOD risks). *For every $\eta \leq \frac{1}{\text{poly}(d_0)}$, after at most $T_2 = \widetilde{\Theta}(\frac{m}{\eta d_0})$ iterations with high probability, the trained model $h^{(T_2)}$ satisfies the following:*

$$\mathcal{R}_{\mathcal{D}_{\text{train}}}(h^{(T_2)}) \leq o(1), \quad \mathcal{R}_{\text{OOD}}(h^{(T_2)}) = \Theta(1). \quad (9)$$

Intuitively, the reason for this result is that the learned model $h^{(T_2)}$ predicts the label of ID examples using *both* the learned core features and the "accompanied" background features due to their non-linear coupling in the neuron's activation. Due to this coupling, negative shift on the magnitude of background features also *reduces the overall activation of the neuron*, resulting in OOD risk.[5]

**Connection to experiments.** We note that the OOD failure mode articulated above also explains our empirical observations in Figure 2, where the model's attention on the core objects gets weakened under distribution shifts—since the Grad-CAM score of each feature map is proportional to

---

[5]While our theorems are proved in a binary classification setting since it is more amenable to analysis, our numerical experiments suggest that feature accompaniment also happens in representation distillation. Thus, we believe that our analysis can be extended to the setting that exactly matches our experimental setup in Section 2.

the neuron's activation (Selvaraju et al., 2019), if core and background features are coupled in the activation as shown in Theorem 2, then the shift of *background features* can make the activation less positive or even negative (i.e., removing the contribution of this neuron to classification), which in turn reduces the Grad-CAM score of *core features*. Since higher Grad-CAM score corresponds to more saliency in the prediction heatmap, this would result in weakened attention to core objects.

**4.4. Linear models are provably free from feature accompaniment.** Finally, to further understand the role of non-linearity, we prove that if we "remove" the non-linearity in the model by replacing ReLU with identity functions, then feature accompaniment will no longer exist.

**Theorem 4** (Linear networks)**.** *If we replace the ReLU functions in the network with identity functions and keep other conditions the same as in Theorem 2, then with high probability, we have* $|\langle \mathbf{w}_k^{(T_1)}, \boldsymbol{m}_j \rangle| \leq \widetilde{O}(\frac{1}{\sqrt{d}})$ *for every* $k \in [m]$ *and every* $j \in \mathcal{S}_{\mathrm{bg}}$.

The intuition is that without non-linearity, the activation magnitude for different examples will be no longer asymmetric: for two-layer linear networks, we have the gradient projection akin to Eq. (7) but without the activation derivative $\mathbb{1}_{\langle \mathbf{w}_k^{(t)}, \mathbf{x} \rangle \geq 0}$. This immediately leads to $\langle -\nabla_{\mathbf{w}_k^{(t)}} \widehat{\mathcal{L}}(h^{(t)}), \boldsymbol{m}_j \rangle \approx 0$ for every $j \in \mathcal{S}_{\mathrm{bg}}$, meaning that the background features will not be accumulated during SGD.

As more **empirical evidence** that corroborates our theory, in Section F.1, we provide numerical experiments in both synthetic classification and representation distillation tasks. In Section F.2, we visualize the features learned by a ResNet-32 on a modified CIFAR-10 dataset, showing that feature accompaniment also happens in *deep features* learned by neural networks used in practice.

## 5 DISCUSSION

### 5.1 TAKEAWAYS

**Takeaway 1: OOD generalization algorithms need to consider inductive biases.** Prior algorithmic studies in OOD generalization often motivate and analyze their algorithms in simplified linear settings, which may fail to capture the inductive biases of non-linear neural networks. Our work implies that OOD generalization may not be feasible without considering such inductive biases, calling for explicitly incorporating them into the development of principled OOD generalization algorithms.

**Takeaway 2: Non-linearity in neural networks elicits new OOD generalization challenges beyond spurious correlations.** As we formally show in Section 4, feature accompaniment is a new OOD generalization challenge that is essentially induced by the non-linearity of SGD-trained neural networks, being orthogonal to spurious correlations. We believe that this result provides a new perspective on OOD generalization in practice and may inspire new algorithmic designs.

**Takeaway 3: Learned features may behave very differently from prescribed ones.** Many existing studies on OOD generalization explicitly or implicitly assume that we can directly work on a set of *well-separated* features. While this assumption helps build intuitions, our results highlight that it can also be misleading since the features *learned* by neural networks may manifest in a *non-linearly coupled* manner, thus often diverging from the intuitions for prescribed, well-separated features.

### 5.2 LIMITATIONS AND FUTURE WORK

While our work takes a step towards fully understanding OOD generalization in practice, our results still leave much room for improvement such as extensions to more general data distributions, multi-class classification, and more complicated network architectures. More importantly, while our results indicate the innate difficulty in achieving OOD generalization with *only* ID data, they do not readily explain *how* pre-training on *more diverse data* consistently helps OOD generalization as observed in practice. Based on our preliminary experiments, we have the following conjecture:

**Conjecture 1.** *Pre-training on a sufficiently large and diverse dataset alleviates feature accompaniment and leads to more linearized representations, hence improving OOD genenralization.*

We provide preliminary empirical evidence that supports this conjecture in Section G. However, we believe that formally proving this conjecture may require more fine-grained treatment in the (pre-training) data generation process and the dynamics of SGD, which we leave as future work.

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

# APPENDIX

The appendix is divided into two parts to maximize readability. We provide complete proofs of our theoretical results in Appendix I, while presenting the experimental details and additional empirical results in Appendix II.

## APPENDIX I: PROOFS OF THEORETICAL RESULTS

In this part of the appendix, we provide complete proofs of our theorems in the main text. A quick overview of the structure of this part is as follows:

- In Section A, we introduce the preliminaries and some lemmas that characterize the neuron properties at network initialization.

- In Section B, we provide the proofs of our main theorems on activation asymmetry (Theorem 1), feature accompaniment (Theorem 2), and the induced ID and OOD risks (Theorem 3).

- In Section C, we provide the proof of our theorem on linear neural networks (Theorem 4).

- In Section D, we provide the basic probability theory lemmas used in the proofs for completeness.

## A    PRELIMINARIES

**Notation.**    Throughout the appendix, we overload $\mathcal{D}_{\mathrm{train}}$ and $\mathcal{D}_{\mathrm{test}}$ to allow them to denote (joint) training and test distributions on both $\mathcal{X} \times \mathcal{Y}$ and $\mathcal{Z} \times \mathcal{Y}$. We also use $\mathcal{D}_{\mathrm{train}}$ and $\mathcal{D}_{\mathrm{test}}$ to denote the corresponding marginal distributions on $\mathcal{X}$, $\mathcal{Y}$ and $\mathcal{Z}$. For presentation brevity, unless otherwise stated, we use the shorthand $\mathbb{E}_{(\cdot)}$ and $\mathbf{Pr}_{(\cdot)}$ to denote $\mathbb{E}_{(\cdot)\sim\mathcal{D}_{\mathrm{train}}}$ and $\mathbf{Pr}_{(\cdot)\sim\mathcal{D}_{\mathrm{train}}}$, respectively, and use the shorthand $h$ to denote $h^{(t)}$ when it is clear from context. As in Definition 1, we denote the moments of each $\mathbf{z}_j$ on the training distribution by $\mu_{jp} := \mathbb{E}_{\mathbf{z}\sim\mathcal{D}_{\mathrm{train}}}\mathbf{z}_j^p$ for every $j \in [d_0]$ and $p \in [3]$, and use the shorthand $\mu_j$ to denote $\mu_{j1}$ when it is clear from the context.

### A.1    WEIGHT DECOMPOSITION AND GRADIENT CALCULATIONS

We begin by recalling that each weight vector $\mathbf{w}_k \in \mathbb{R}^d, k \in [m]$ (i.e., the learned feature of the $k$-th neuron) in the network can be decomposed into the sum of its projections to different feature vectors:

$$\mathbf{w}_k^{(t)} = \sum_{j\in\mathcal{S}_{\mathrm{core}}} \langle \mathbf{w}_k^{(t)}, \boldsymbol{m}_j \rangle \boldsymbol{m}_j + \sum_{j\in\mathcal{S}_{\mathrm{bg}}} \langle \mathbf{w}_k^{(t)}, \boldsymbol{m}_j \rangle \boldsymbol{m}_j + \sum_{j\in[d]\setminus[d_0]} \langle \mathbf{w}_k^{(t)}, \boldsymbol{m}_j \rangle \boldsymbol{m}_j, \qquad (10)$$

where $(\boldsymbol{m}_{d_0+1}, \ldots, \boldsymbol{m}_d)$ are an orthogonal complement of $\boldsymbol{M}$. Since all possible inputs are generated to be in $\mathrm{span}\{\boldsymbol{m}_1, \ldots, \boldsymbol{m}_{d_0}\}$ as in Definition 1, the last term in the RHS of Eq. (10) (i.e., the residual term in Eq. (6) in the main text) can be neglected due to the orthogonality of different feature vector $\boldsymbol{m}_j$s. Therefore, throughout the following analysis, we will overload the notation $\mathbf{w}_k^{(t)}$ and let

$$\mathbf{w}_k^{(t)} = \sum_{j\in\mathcal{S}_{\mathrm{core}}} \langle \mathbf{w}_k^{(t)}, \boldsymbol{m}_j \rangle \boldsymbol{m}_j + \sum_{j\in\mathcal{S}_{\mathrm{bg}}} \langle \mathbf{w}_k^{(t)}, \boldsymbol{m}_j \rangle \boldsymbol{m}_j. \qquad (11)$$

A direct consequence of Eq. (10) is that we can analyze the feature learning process of the network by tracking the correlations between each weight vector $\mathbf{w}_k^{(t)}$ and different feature vectors $\boldsymbol{m}_j, j \in [d]$ as the training proceeds. To this end, we need to first analyze the gradient of each neuron at every iteration.

**Gradient of each neuron.**    Recall that at each iteration $t = 0, \ldots, T-1$, the SGD update for each weight vector $\mathbf{w}_k, k \in [m]$ is given by

$$\mathbf{w}_k^{(t+1)} \leftarrow (1-\eta\lambda)\mathbf{w}_k^{(t)} - \eta\nabla_{\mathbf{w}_k^{(t)}}\frac{1}{N}\sum_{i\in[N]} \ell\left(h^{(t)}(\mathbf{x}_i^{(t)}), \mathbf{y}_i^{(t)}\right),$$

where

$$h^{(t)}(\mathbf{x}_i^{(t)}) = \sum_{k \in [m]} a_k \cdot \mathrm{ReLU}\big(\langle \mathbf{w}_k^{(t)}, \mathbf{x}_i^{(t)} \rangle\big)$$

and $\ell(y, y') = \max\{1 - yy', 0\}$. We can then calculate the gradient of each neuron $\mathbf{w}_k^{(t)}$ with regard to a certain example $(\mathbf{x}, \mathbf{y})$:

**Lemma 1** (Gradient). *For every example $(x, y) \in \mathcal{X} \times \mathcal{Y}$, every $k \in [m]$, and every iteration $t$, the following holds:*

$$\nabla_{\mathbf{w}_k^{(t)}} \ell\left(h(x), y\right) = -a_k y \mathbb{1}_{h(x) \le 1} \mathbb{1}_{\langle \mathbf{w}_k^{(t)}, x \rangle \ge 0} x. \tag{12}$$

*Proof.* The proof follows from simple calculation. $\qquad\square$

We then introduce a lemma that bounds the empirical growth of the correlation between each neuron $\mathbf{w}_k^{(t)}$ and each feature vector $\boldsymbol{m}_j$ after an SGD update using population gradients.

**Lemma 2** (Gap between empirical and population gradients). *For every $k \in [m]$, every $j \in [d]$, and every iteration $t$, if the batch size $N = \mathrm{poly}(d)$ for some sufficiently large polynomial, then the following holds with probability at least $1 - e^{-\Omega(d)}$:*

$$\left| \left\langle \nabla_{\mathbf{w}_k^{(t)}} \frac{1}{N} \sum_{i \in [N]} \ell\left(h^{(t)}(\mathbf{x}_i^{(t)}), \mathbf{y}_i^{(t)}\right), \boldsymbol{m}_j \right\rangle - \left\langle \nabla_{\mathbf{w}_k^{(t)}} \mathbb{E}_{(\mathbf{x},\mathbf{y}) \sim \mathcal{D}_{\mathrm{train}}} \ell\left(h(\mathbf{x}), \mathbf{y}\right), \boldsymbol{m}_j \right\rangle \right| \le \frac{1}{\mathrm{poly}(d)}. \tag{13}$$

*Proof.* Recall that $\|\boldsymbol{m}_j\|_2 = 1$. Applying Cauchy-Schwarz inequality gives

$$\left| \left\langle \nabla_{\mathbf{w}_k^{(t)}} \frac{1}{N} \sum_{i \in [N]} \ell\left(h^{(t)}(\mathbf{x}_i^{(t)}), \mathbf{y}_i^{(t)}\right), \boldsymbol{m}_j \right\rangle - \left\langle \nabla_{\mathbf{w}_k^{(t)}} \mathbb{E}_{(\mathbf{x},\mathbf{y}) \sim \mathcal{D}_{\mathrm{train}}} \ell\left(h(\mathbf{x}), \mathbf{y}\right), \boldsymbol{m}_j \right\rangle \right|$$

$$\le \underbrace{\left\| \nabla_{\mathbf{w}_k^{(t)}} \frac{1}{N} \sum_{i \in [N]} \ell\left(h^{(t)}(\mathbf{x}_i^{(t)}), \mathbf{y}_i^{(t)}\right) - \nabla_{\mathbf{w}_k^{(t)}} \mathbb{E}_{(\mathbf{x},\mathbf{y}) \sim \mathcal{D}_{\mathrm{train}}} \ell\left(h(\mathbf{x}), \mathbf{y}\right) \right\|_2}_{\mathbf{S}^{(t)}}.$$

We define

$$\mathbf{Z}_i^{(t)} := \frac{1}{N} \nabla_{\mathbf{w}_k^{(t)}} \ell\left(h^{(t)}(\mathbf{x}_i^{(t)}), \mathbf{y}_i^{(t)}\right) - \frac{1}{N} \nabla_{\mathbf{w}_k^{(t)}} \mathbb{E}_{(\mathbf{x},\mathbf{y}) \sim \mathcal{D}_{\mathrm{train}}} \ell\left(h(\mathbf{x}), \mathbf{y}\right), \ \forall i \in [N].$$

It is easy to see that $\mathbf{S}^{(t)} = \sum_{i \in [N]} \mathbf{Z}_i^{(t)}$, $\mathbb{E} \mathbf{Z}_i^{(t)} = 0$ for every $i \in [N]$, and $\forall i \ne j \in [N]$, $\mathbf{Z}_i^{(t)}$ and $\mathbf{Z}_j^{(t)}$ are independent. By Lemma 1, we have

$$\mathbf{Z}_i^{(t)} = \frac{1}{N} \mathbb{E}_{(\mathbf{x},\mathbf{y}) \sim \mathcal{D}_{\mathrm{train}}} a_k \mathbf{y} \mathbb{1}_{h(\mathbf{x}) \le 1} \mathbb{1}_{\langle \mathbf{w}_k^{(t)}, \mathbf{x} \rangle \ge 0} \cdot \mathbf{x} - \frac{1}{N} a_k \mathbf{y}_i^{(t)} \mathbb{1}_{h^{(t)}(\mathbf{x}_i^{(t)}) \le 1} \mathbb{1}_{\langle \mathbf{w}_k^{(t)}, \mathbf{x}_i^{(t)} \rangle \ge 0} \cdot \mathbf{x}_i^{(t)}.$$

Recall that $a_k \in \{-\frac{1}{m}, \frac{1}{m}\}$ and $\mathbf{x}$ is generated by $\mathbf{x} = \sum_{j \in \mathcal{S}_{\mathrm{core}}} \mathbf{y} \mathbf{z}_j \boldsymbol{m}_j + \sum_{j \in \mathcal{S}_{\mathrm{bg}}} \mathbf{z}_j \boldsymbol{m}_j$ according to Definition 1. We then have $\|\mathbf{Z}_i^{(t)}\|_2 \le \frac{2\sqrt{d_0}}{mN}$, which also indicates that $\mathbb{E} \langle \mathbf{Z}_i^{(t)}, \mathbf{Z}_i^{(t)} \rangle \le \frac{4d_0}{m^2 N^2}$. This gives

$$\mathbb{E} \langle \mathbf{S}^{(t)}, \mathbf{S}^{(t)} \rangle = \sum_{i \in [N]} \mathbb{E} \langle \mathbf{Z}_i^{(t)}, \mathbf{Z}_i^{(t)} \rangle \le \frac{4d_0}{m^2 N}.$$

Applying matrix Bernstein's inequality (Lemma 20), we have

$$\mathbf{Pr}\left[ \|\mathbf{S}^{(t)}\|_2 \ge \delta \right] \le (d+1) \exp\left( -\frac{3m^2 N^2 \delta^2}{24d_0 + 4\sqrt{d_0} \delta mN} \right)$$

hold with every $\delta = \frac{1}{\mathrm{poly}(d)}$. Therefore, we have that for $N = \mathrm{poly}(d)$ with some sufficiently large polynomial, the following holds with probability at least $1 - e^{-\Omega(d)}$:

$$\|\mathbf{S}^{(t)}\|_2 \le \frac{1}{\mathrm{poly}(d)}.$$

This gives the desired result. $\qquad\square$

Lemma 2 directly leads to the following corollary:

**Lemma 3** (Projection approximation). *For every $k \in [m]$, every $j \in [d_0]$, and every iteration $t$, the following holds:*

$$\left\langle \nabla_{\mathbf{w}_k^{(t)}} \frac{1}{N} \sum_{i \in [N]} \ell\left(h^{(t)}(\mathbf{x}_i^{(t)}), \mathbf{y}_i^{(t)}\right), \boldsymbol{m}_j \right\rangle$$

$$= \left\langle \nabla_{\mathbf{w}_k^{(t)}} \mathbb{E}_{(\mathbf{x},\mathbf{y}) \sim \mathcal{D}_{\text{train}}} \ell\left(h(\mathbf{x}), \mathbf{y}\right), \boldsymbol{m}_j \right\rangle \pm \frac{1}{\text{poly}(d)} \tag{14}$$

$$= -a_k \mathbb{E}_{(\mathbf{x},\mathbf{y}) \sim \mathcal{D}_{\text{train}}} \mathbf{y} \mathbb{1}_{h(\mathbf{x}) \leq 1} \mathbb{1}_{\langle \mathbf{w}_k^{(t)}, \mathbf{x} \rangle \geq 0} \cdot \mathbf{z}_j \pm \frac{1}{\text{poly}(d)}, \quad j \in [d_0].$$

*Proof.* The proof directly follows from combining Lemma 1 and the generation process of $\mathbf{x}$ in Definition 1. $\qquad \square$

Lemma 3 allows us to directly work with population gradients instead of empirical gradients when analyzing the trajectory of SGD iterations in the subsequent sections.

## A.2 NEURON CHARACTERIZATION

In this section, we define two subsets of neurons that will be used throughout our proofs.

**Definition 2** (Neuron characterization). *For each label $y \in \mathcal{Y} = \{-1, 1\}$ and every iteration $t$, we define the set $\mathcal{N}_y^{(t)} \subseteq [m]$ as:*

$$\mathcal{N}_y^{(t)} := \left\{ k \in [m] : \sum_{j \in \mathcal{S}_{\text{core}}} y\mu_j \langle \mathbf{w}_k^{(t)}, \boldsymbol{m}_j \rangle + \sum_{j \in \mathcal{S}_{\text{bg}}} \mu_j \langle \mathbf{w}_k^{(t)}, \boldsymbol{m}_j \rangle \geq \Theta\left(\sqrt{\frac{d_0}{d}}\right), \right.$$

$$\left. \text{sign}\left(a_k\right) = y \right\}. \tag{15}$$

**Intuition.** For each label $y \in \mathcal{Y}$ and iteration $t$, Definition 2 characterizes a subset of neurons $\mathcal{N}_y^{(t)}$ in which

- each neuron has (in expectation) enough positive correlations with the examples from class $y$ (recall that $\mathbf{x} = \sum_{j \in \mathcal{S}_{\text{core}}} \mathbf{y} \mathbf{z}_j \boldsymbol{m}_j + \sum_{j \in \mathcal{S}_{\text{bg}}} \mathbf{z}_j \boldsymbol{m}_j$);
- each neuron positively contributes to the classification of examples from class $y$ (i.e., $\text{sign}\left(a_k\right) = y$).

In our main proof, we will show in an iterative fashion that each neuron in $\mathcal{N}_y^{(t)}$ will accumulate either positive (if random initialization gives $a_k = \frac{1}{m}$) or negative (if random initialization gives $a_k = -\frac{1}{m}$) correlations with features in $\mathcal{S}_{\text{core}}$ (*core feature learning*), while also accumulating positive correlations with features in $\mathcal{S}_{\text{bg}}$ (*feature accompaniment*).

For each neuron, we formally define the notion of *positive examples* and *negative examples* which are informally mentioned in Section 4:

**Definition 3** (Positive examples and negative examples). *Let $(x, y) \in \mathcal{X} \times \mathcal{Y}$ be an example. For every $k \in [m]$, we say that $(x, y)$ is a **positive example** of neuron $k$ if $\text{sign}(a_k) = y$, and say that $(x, y)$ is a **negative example** of neuron $k$ if $\text{sign}(a_k) = -y$.*

## A.3 PROPERTIES AT INITIALIZATION

In this section, we introduce some useful properties of the neurons at initialization $t = 0$, which serve as a basis for our inductive proofs in the subsequent sections.

**Lemma 4.** *For every $j \in [d_0]$, every $\mathcal{S} \subseteq [d_0]$, and every $\{y_j\}_{j \in \mathcal{S}} \in \{-1, 1\}^{|\mathcal{S}|}$, the following holds for every $\delta > 0$ over random initialization:*

$$
\mathbf{Pr}_{\mathbf{W}^{(0)}}\left[\sum_{j \in \mathcal{S}} y_j \mu_j \langle \mathbf{w}_k^{(0)}, \boldsymbol{m}_j \rangle \geq \frac{\delta}{\sqrt{d}}\right] \geq \frac{1}{\sqrt{2\pi}} \frac{\delta \sqrt{\sum_{j \in \mathcal{S}} \mu_j^2}}{\delta^2 + \sum_{j \in \mathcal{S}} \mu_j^2} \exp\left(-\frac{\delta^2}{2 \sum_{j \in \mathcal{S}} \mu_j^2}\right),
$$

$$
\mathbf{Pr}_{\mathbf{W}^{(0)}}\left[\sum_{j \in \mathcal{S}} y_j \mu_j \langle \mathbf{w}_k^{(0)}, \boldsymbol{m}_j \rangle \geq \frac{\delta}{\sqrt{d}}\right] \leq \frac{1}{\sqrt{2\pi}} \frac{\sqrt{\sum_{j \in \mathcal{S}} \mu_j^2}}{\delta} \exp\left(-\frac{\delta^2}{2 \sum_{j \in \mathcal{S}} \mu_j^2}\right).
$$

(16)

*Proof.* Recall that different neurons are independently initialized by $\mathbf{w}_k^{(0)} \sim \mathcal{N}(\mathbf{0}, \sigma_0^2 \boldsymbol{I}_d), \forall k \in [m]$ with $\sigma_0^2 = \frac{1}{d}$. Using the fact that $\|\boldsymbol{m}_j\|_2 = 1, \forall j \in [d_0]$ and $y_j^2 = 1, \forall j \in \mathcal{S}$, we have

$$
\sum_{j \in \mathcal{S}} y_j \mu_j \langle \mathbf{w}_k^{(0)}, \boldsymbol{m}_j \rangle \sim \mathcal{N}\left(0, \frac{1}{d} \sum_{j \in \mathcal{S}} \mu_j^2\right)
$$

Applying standard bounds for the Gaussian distribution function (Lemma 21) gives that for every $\delta > 0$,

$$
\frac{1}{\sqrt{2\pi}} \frac{\delta}{\delta^2 + 1} \exp\left(-\frac{\delta^2}{2}\right) \leq \mathbf{Pr}_{\mathbf{W}^{(0)}}\left[\frac{\sqrt{d} \sum_{j \in \mathcal{S}} y_j \mu_j \langle \mathbf{w}_k^{(0)}, \boldsymbol{m}_j \rangle}{\sqrt{\sum_{j \in \mathcal{S}} \mu_j^2}} \geq \delta\right] \leq \frac{1}{\sqrt{2\pi}} \frac{1}{\delta} \exp\left(-\frac{\delta^2}{2}\right).
$$

A simple transformation completes the proof. □

**Lemma 5** (Neuron properties at initialization). *For each label $y \in \mathcal{Y}$, the following holds with probability at least $1 - e^{-\Omega(m)}$ over random initialization:*

$$
\left|\mathcal{N}_y^{(0)}\right| = \Theta(m).
$$

(17)

*Proof.* For each neuron $k \in [m]$, define events $E_{k1}$ and $E_{k2}$ to be

$$
E_{k1} := \left\{ \sum_{j \in \mathcal{S}_{\text{core}}} y \mu_j \langle \mathbf{w}_k^{(0)}, \boldsymbol{m}_j \rangle + \sum_{j \in \mathcal{S}_{\text{bg}}} \mu_j \langle \mathbf{w}_k^{(0)}, \boldsymbol{m}_j \rangle \geq \Theta\left(\sqrt{\frac{d_0}{d}}\right) \right\},
$$

$$
E_{k2} := \left\{ \text{sign}(a_k) = y \right\}.
$$

By $a_k \sim \text{Uniform}\{-\frac{1}{m}, \frac{1}{m}\}$, we immediately have $\mathbf{Pr}[E_{k2}] = \frac{1}{2}$ for every $k \in [m]$. For $E_{k1}$, by applying Lemma 4 with $\delta = \Theta(\sqrt{d_0})$ we obtain

$$
\mathbf{Pr}_{\mathbf{W}^{(0)}}[E_{k1}] \geq \frac{1}{\sqrt{2\pi}} \frac{\Theta\left(\sqrt{d_0 \sum_{j \in [d_0]} \mu_j^2}\right)}{\Theta(d_0) + \sum_{j \in [d_0]} \mu_j^2} \exp\left(-\Theta\left(\frac{d_0}{\sum_{j \in [d_0]} \mu_j^2}\right)\right),
$$

$$
\mathbf{Pr}_{\mathbf{W}^{(0)}}[E_{k1}] \leq \frac{1}{\sqrt{2\pi}} \Theta\left(\sqrt{\frac{\sum_{j \in [d_0]} \mu_j^2}{d_0}}\right) \exp\left(-\Theta\left(\frac{d_0}{\sum_{j \in [d_0]} \mu_j^2}\right)\right).
$$

Together with $\mu_j^2 = \Theta(1)$ for every $j \in [d_0]$, we have that $\mathbf{Pr}_{\mathbf{W}^{(0)}}[E_{k1}] = \Theta(1)$ for every $k \in [m]$. Since events $E_{k1}$ and $E_{k2}$ are independent, we have that for each neuron $k \in [m]$, the probability of it belonging to $\mathcal{N}_y^{(0)}$ is given by $\mathbf{Pr}(k \in \mathcal{N}_y^{(0)}) = \mathbf{Pr}(E_{k1} \cap E_{k2}) = \Theta(1)$.

Since different neurons are independently initialized, $|\mathcal{N}_y^{(0)}|$ follows a binomial distribution with trial number $m$ and some success probability $\Theta(1)$. Applying the standard tail bound for binomial variables (Lemma 22) then gives $|\mathcal{N}_y^{(0)}| \geq \Theta(m)$ with probability at least $1 - e^{-\Omega(m)}$. Together with the trivial upper bound that $|\mathcal{N}_y^{(0)}| \leq m$, we have that $|\mathcal{N}_y^{(0)}| = \Theta(m)$ with probability at least $1 - e^{-\Omega(m)}$. □

**Lemma 6** (Neuron properties at initialization, continued). *With probability at least $1 - O(\frac{1}{m})$ over random initialization, for every $y \in \mathcal{Y}$, the following holds:*

$$\max_{k \in [m]} \left| \mathbb{E}_{\mathbf{x}|\mathbf{y}=y \sim \mathcal{D}_{\mathrm{train}}} \langle \mathbf{w}_k^{(0)}, \mathbf{x} \rangle \right| \leq O\left( \sqrt{\frac{d_0 \log m}{d}} \right).$$

*Proof.* Recall that different neurons are independently initialized by $\mathbf{w}_k^{(0)} \sim \mathcal{N}(\mathbf{0}, \sigma_0^2 \mathbf{I}_d), \forall k \in [m]$ with $\sigma_0^2 = \frac{1}{d}$. By $\|\boldsymbol{m}_j\|_2 = 1$, we have

$$\mathbb{E}_{\mathbf{x}|\mathbf{y}=y \sim \mathcal{D}_{\mathrm{train}}} \langle \mathbf{w}_k^{(0)}, \mathbf{x} \rangle = \sum_{j \in \mathcal{S}_{\mathrm{core}}} y\mu_j \langle \mathbf{w}_k^{(0)}, \boldsymbol{m}_j \rangle + \sum_{j \in \mathcal{S}_{\mathrm{bg}}} \mu_j \langle \mathbf{w}_k^{(0)}, \boldsymbol{m}_j \rangle$$

$$\sim \mathcal{N}\left(0, \frac{1}{d} \sum_{j \in [d_0]} \mu_j^2\right).$$

Applying Lemma 23 over the i.i.d. random variables $\langle \mathbf{w}_1^{(0)}, \mathbf{x} \rangle, \ldots, \langle \mathbf{w}_m^{(0)}, \mathbf{x} \rangle$ gives

$$\mathbf{Pr}_{\mathbf{W}^{(0)}}\left[ \mathbb{E}_{\mathbf{x}|\mathbf{y}=y \sim \mathcal{D}_{\mathrm{train}}} \langle \mathbf{w}_k^{(0)}, \mathbf{x} \rangle \geq 2\sqrt{\frac{\sum_{j \in [d_0]} \mu_j^2}{d} \log m} \right] \leq \frac{1}{m}.$$

Finally, using $\sum_{j \in [d_0]} \mu_j^2 = \Theta(d_0)$ and $m \in [\Theta(d_0), \Theta(d)]$ completes the proof. $\square$

**Lemma 7** (Output magnitude at initialization). *For every $x \in \mathcal{X}$, the following holds with probability at least $1 - e^{-\Omega(d_0)}$ over random initialization:*

$$\left| h^{(0)}(x) \right| = O\left( \frac{1}{\sqrt{d_0}} \right). \tag{18}$$

*Proof.* By $\mathbf{w}_k^{(0)} \sim \mathcal{N}(\mathbf{0}, \sigma_0^2 \mathbf{I}_d)$ with $\sigma_0^2 = \frac{1}{d}$ and $\|\boldsymbol{m}_j\|_2 = 1$, we have

$$\sum_{k \in [m]} \frac{1}{m} \sum_{j \in [d_0]} \langle \mathbf{w}_k^{(0)}, \boldsymbol{m}_j \rangle \sim \mathcal{N}\left(0, \frac{d_0}{md}\right).$$

Applying standard bounds for the Gaussian distribution function (Lemma 21) gives

$$\frac{1}{\sqrt{2\pi}} \frac{\delta}{\delta^2 + 1} \exp\left(-\frac{\delta^2}{2}\right) \leq \mathbf{Pr}_{\mathbf{W}^{(0)}}\left[ \sum_{k \in [m]} \frac{1}{m} \sum_{j \in [d_0]} \langle \mathbf{w}_k^{(0)}, \boldsymbol{m}_j \rangle \geq \delta \sqrt{\frac{d_0}{md}} \right] \leq \frac{1}{\sqrt{2\pi}} \frac{1}{\delta} \exp\left(-\frac{\delta^2}{2}\right)$$

for every $\delta > 0$. Substituting $\delta$ by $\Theta(\sqrt{d_0})$ and using the symmetry of Gaussian then yield

$$\mathbf{Pr}_{\mathbf{W}^{(0)}}\left[ \left| \sum_{k \in [m]} \frac{1}{m} \sum_{j \in [d_0]} \langle \mathbf{w}_k^{(0)}, \boldsymbol{m}_j \rangle \right| \geq \frac{\Theta(d_0)}{\sqrt{md}} \right] \leq \exp(-\Omega(d_0)).$$

We then have

$$\left| h^{(0)}(x) \right| = \left| \sum_{k \in [m]} a_k \cdot \mathrm{ReLU}\left( \langle \mathbf{w}_k^{(0)}, \mathbf{x} \rangle \right) \right|$$

$$\leq \left| \sum_{k \in [m]} \frac{1}{m} \langle \mathbf{w}_k^{(0)}, \mathbf{x} \rangle \right|$$

$$\leq \left| \sum_{k \in [m]} \frac{1}{m} \sum_{j \in [d_0]} \langle \mathbf{w}_k^{(0)}, \boldsymbol{m}_j \rangle \right|$$

$$\leq \frac{\Theta(d_0)}{\sqrt{md}} = O\left( \frac{1}{\sqrt{d_0}} \right).$$

holds with probability at least $1 - e^{-\Omega(d_0)}$, where in the last equality we use the fact that $m = \Omega(d_0)$ and $d = \Omega(d_0^{2.5})$. $\square$

In what follows, we will always assume that the high-probability events at initialization in Lemma 5, Lemma 6, and Lemma 7 hold—by a union bound argument and the fact that $m = \Omega(d_0)$, the probability that they all hold is at least $1 - O(\frac{1}{m}) - e^{-\Omega(d_0)}$.

# B ACTIVATION ASYMMETRY, FEATURE ACCOMPANIMENT, AND OOD FAILURE: PROOFS OF THEOREM 1, THEOREM 2, AND THEOREM 3

Before we delve into the main proofs, we first introduce some technical lemmas that characterize the gradient updates starting from random initialization. We begin by introducing an important lemma that characterizes the activation probability of the ReLU function for each neuron in $\mathcal{N}_y^{(t)}$:

**Lemma 8** (Activation probability). *Assume that the training (ID) data is generated according to Definition 1. Then, for every label $y \in \mathcal{Y}$, every $k \in [m]$, and every iteration t, the following holds:*

$$\mathbf{Pr}_{\mathbf{x}|\mathbf{y}=y\sim\mathcal{D}_{\text{train}}}[\langle \mathbf{w}_k^{(t)}, \mathbf{x}\rangle \geq 0] = \Phi\left(\frac{\mathbb{E}_{\mathbf{x}|\mathbf{y}=y}\langle \mathbf{w}_k^{(t)}, \mathbf{x}\rangle}{\Theta(1)\sqrt{\sum_{j\in[d_0]}\langle \mathbf{w}_k^{(t)}, \mathbf{m}_j\rangle^2}}\right) \pm O\left(\frac{1}{\sqrt{d_0}}\right), \qquad (19)$$

*where $\Phi$ denotes the cumulative distribution function of $\mathcal{N}(0,1)$.*

*Proof.* Recall Definition 1 that given label $y \in \mathcal{Y}$, $\mathbf{x}$ is generated by

$$\mathbf{x} = \sum_{j\in\mathcal{S}_{\text{core}}} y\mathbf{z}_j\mathbf{m}_j + \sum_{j\in\mathcal{S}_{\text{bg}}} \mathbf{z}_j\mathbf{m}_j.$$

Therefore,

$$\langle \mathbf{w}_k^{(t)}, \mathbf{x}\rangle = \sum_{j\in\mathcal{S}_{\text{core}}} y\mathbf{z}_j\langle \mathbf{w}_k^{(t)}, \mathbf{m}_j\rangle + \sum_{j\in\mathcal{S}_{\text{bg}}} \mathbf{z}_j\langle \mathbf{w}_k^{(t)}, \mathbf{m}_j\rangle.$$

For every $j \in [d_0]$, define the random variable

$$\mathbf{r}_j := y_j(\mathbf{z}_j - \mu_j)\langle \mathbf{w}_k^{(t)}, \mathbf{m}_j\rangle,$$

where $y_j := \begin{cases} y, & j \in \mathcal{S}_{\text{core}} \\ 1, & j \in \mathcal{S}_{\text{bg}} \end{cases}$. Recall that $\mu_j := \mathbb{E}_{\mathbf{z}\sim\mathcal{D}_{\text{train}}}\mathbf{z}_j$. It is then easy to derive that $\mathbb{E}\mathbf{r}_j = 0$ and $\mathbb{E}\mathbf{r}_j^2 = \Theta(1)\langle \mathbf{w}_k^{(t)}, \mathbf{m}_j\rangle^2$. We now upper bound $\mathbb{E}|\mathbf{r}_j^3|$: first recall that by Definition 1 we have $\mathbb{E}\mathbf{z}_j^3 = \Theta(1)$. For every $p \geq 1$, denote the $\ell_p$ norm of the random variable $\mathbf{z}_j$ by $\|\mathbf{z}_j\|_p := (\mathbb{E}|\mathbf{z}_j|^p)^{\frac{1}{p}}$. Applying Minkowsky's inequality gives

$$\|\mathbf{z}_j - \mu_j\|_p \leq \|\mathbf{z}_j\|_p + \|\mu_j\|_p$$
$$\overset{(a)}{=} \|\mathbf{z}_j\|_p + \|\mathbf{z}_j\|_1$$
$$\overset{(b)}{\leq} 2\|\mathbf{z}_j\|_p,$$

where $(a)$ is due to the fact that $\|\mu_j\|_p = |\mu_j| = \|\mathbf{z}_j\|_1$ and $(b)$ is due to the power norm inequality indicating that $\|\cdot\|_p$ is non-decreasing with regard to $p$. Letting $p = 3$ and cubing the above inequality gives

$$\mathbb{E}|\mathbf{z}_j - \mu_j|^3 \leq 8\mathbb{E}|\mathbf{z}_j^3| = 8\mathbb{E}\mathbf{z}_j^3 = \Theta(1),$$

from which we obtain $\mathbb{E}|\mathbf{r}_j^3| = O(1) \cdot |\langle \mathbf{w}_k^{(t)}, \mathbf{m}_j\rangle|^3$.

We then define the normalized sum of $\mathbf{r}_j$:

$$\mathbf{s}_{d_0} := \frac{\sum_{j\in[d_0]} \mathbf{r}_j}{\sqrt{\sum_{j\in[d_0]} \mathbb{E}\mathbf{r}_j^2}}.$$

Since $\mathbf{r}_i$ and $\mathbf{r}_j$ are independent and zero-mean for every $i \neq j \in [d_0]$, we can apply Berry-Esseen theorem (Lemma 24) to the normalized sum $\mathbf{s}_{d_0}$ and obtain

$$\sup_{\delta\in\mathbb{R}} \left|\mathbf{Pr}_{\mathbf{x}|\mathbf{y}=y}[\mathbf{s}_{d_0} < \delta] - \Phi(\delta)\right| \leq C_0 \left(\sum_{i=1}^{d_0} \mathbb{E}\mathbf{r}_j^2\right)^{-\frac{3}{2}} \sum_{i=1}^{d_0} \mathbb{E}|\mathbf{r}_j^3|$$
$$= O\left(\frac{1}{\sqrt{d_0}}\right).$$

Note that $\sum_{j\in[d_0]}\mathbf{r}_j = \langle\mathbf{w}_k^{(t)},\mathbf{x}\rangle - \mathbb{E}_{\mathbf{x}|\mathbf{y}=y}\langle\mathbf{w}_k^{(t)},\mathbf{x}\rangle$. We then have for every $\delta\in\mathbb{R}$,

$$\mathbf{Pr}_{\mathbf{x}|\mathbf{y}=y}\left[\langle\mathbf{w}_k^{(t)},\mathbf{x}\rangle \geq \mathbb{E}_{\mathbf{x}|\mathbf{y}=y}\langle\mathbf{w}_k^{(t)},\mathbf{x}\rangle + \delta\sqrt{\sum_{j\in[d_0]}\mathbb{E}\mathbf{r}_j^2}\right] = 1 - \Phi(\delta)\pm O\left(\frac{1}{\sqrt{d_0}}\right).$$

Finally, letting $\delta = -\frac{\mathbb{E}_{\mathbf{x}|\mathbf{y}=y}\langle\mathbf{w}_k^{(t)},\mathbf{x}\rangle}{\sqrt{\sum_{j\in[d_0]}\mathbb{E}\mathbf{r}_j^2}}$ and using the symmetry of unit Gaussian $1 - \Phi(\delta) = \Phi(-\delta)$ give the desired result. $\square$

We then define two terms that will be frequently used when analyzing gradients.

**Definition 4.** *For each label $y\in\mathcal{Y}$, every $k\in[m]$, every feature vector $\boldsymbol{m}_j, j\in[d_0]$, every iteration $t$, and every subset $\mathcal{S}\subseteq[d_0]$, define*

$$\begin{aligned}g_{k,y,j}^{(t)} &:= \frac{1}{m}\mathbb{E}_{(\mathbf{x},\mathbf{y})\sim\mathcal{D}_{\text{train}}}\mathbb{1}_{h^{(t)}(\mathbf{x})\leq 1}\mathbb{1}_{\mathbf{y}=y}\mathbb{1}_{\langle\mathbf{w}_k^{(t)},\mathbf{x}\rangle\geq 0}\mu_j\mathbf{z}_j,\\ g_{k,y,\mathcal{S}}^{(t)} &:= \sum_{j\in\mathcal{S}}g_{k,y,j}^{(t)}.\end{aligned}$$

(20)

Given the above notation, we now introduce two lemmas that separately bound the gradient projection onto the core features and the gradient projection onto the background features for neurons in $\mathcal{N}_y^{(t)}$, which will be helpful for us to track the trajectory of SGD starting from network initialization.

**Lemma 9** (Gradient projection onto core features, neurons in $\mathcal{N}_y^{(t)}$). *For every iteration $t\leq O(\frac{m}{\eta d_0})$, the following holds for every $y\in\mathcal{Y}$ and every neuron $k\in\mathcal{N}_y^{(t)}$:*

$$-\left\langle\nabla_{\mathbf{w}_k^{(t)}}\frac{1}{N}\sum_{i\in[N]}\ell\left(h^{(t)}(\mathbf{x}_i^{(t)}),\mathbf{y}_i^{(t)}\right),\sum_{j\in\mathcal{S}_{\text{core}}}\mu_j\boldsymbol{m}_j\right\rangle = y\left(g_{k,y,\mathcal{S}_{\text{core}}}^{(t)}+g_{k,-y,\mathcal{S}_{\text{core}}}^{(t)}\right),$$

(21)

*where*

$$g_{k,y,\mathcal{S}_{\text{core}}}^{(t)} = \Theta\left(\frac{d_0}{m}\right).$$

(22)

*Proof.* Recall Definition 1 that given a label $\mathbf{y}\in\mathcal{Y}$, $\mathbf{x}$ is generated by

$$\mathbf{x} = \sum_{j\in\mathcal{S}_{\text{core}}}\mathbf{y}\mathbf{z}_j\boldsymbol{m}_j + \sum_{j\in\mathcal{S}_{\text{bg}}}\mathbf{z}_j\boldsymbol{m}_j.$$

Then, applying Lemma 3 to the LHS of Eq. (21) and using $\text{sign}(a_k) = y$ for every $k\in\mathcal{N}_y^{(t)}$, we obtain

$$\begin{aligned}&-\left\langle\nabla_{\mathbf{w}_k^{(t)}}\frac{1}{N}\sum_{i\in[N]}\ell\left(h^{(t)}(\mathbf{x}_i^{(t)}),\mathbf{y}_i^{(t)}\right),\sum_{j\in\mathcal{S}_{\text{core}}}\mu_j\boldsymbol{m}_j\right\rangle\\ &= -\left\langle\nabla_{\mathbf{w}_k^{(t)}}\mathbb{E}_{(\mathbf{x},\mathbf{y})\sim\mathcal{D}_{\text{train}}}\ell\left(h^{(t)}(\mathbf{x}),\mathbf{y}\right),\sum_{j\in\mathcal{S}_{\text{core}}}\mu_j\boldsymbol{m}_j\right\rangle\pm\frac{O(d_0)}{\text{poly}(d)}\\ &= a_k\mathbb{E}_{(\mathbf{x},\mathbf{y})\sim\mathcal{D}_{\text{train}}}\mathbf{y}\mathbb{1}_{h^{(t)}(\mathbf{x})\leq 1}\mathbb{1}_{\langle\mathbf{w}_k^{(t)},\mathbf{x}\rangle\geq 0}\sum_{j\in\mathcal{S}_{\text{core}}}\mathbf{y}\mu_j\mathbf{z}_j\pm\frac{O(d_0)}{\text{poly}(d)}\\ &= a_k\mathbb{E}_{(\mathbf{x},\mathbf{y})\sim\mathcal{D}_{\text{train}}}\mathbb{1}_{h^{(t)}(\mathbf{x})\leq 1}\mathbb{1}_{\mathbf{y}=y}\mathbb{1}_{\langle\mathbf{w}_k^{(t)},\mathbf{x}\rangle\geq 0}\sum_{j\in\mathcal{S}_{\text{core}}}\mu_j\mathbf{z}_j\\ &\quad + a_k\mathbb{E}_{(\mathbf{x},\mathbf{y})\sim\mathcal{D}_{\text{train}}}\mathbb{1}_{h^{(t)}(\mathbf{x})\leq 1}\mathbb{1}_{\mathbf{y}=-y}\mathbb{1}_{\langle\mathbf{w}_k^{(t)},\mathbf{x}\rangle\geq 0}\sum_{j\in\mathcal{S}_{\text{core}}}\mu_j\mathbf{z}_j\pm\frac{O(d_0)}{\text{poly}(d)}\\ &= y\left(g_{k,y,\mathcal{S}_{\text{core}}}^{(t)}+g_{k,-y,\mathcal{S}_{\text{core}}}^{(t)}\right)\pm\frac{O(d_0)}{\text{poly}(d)}.\end{aligned}$$

(23)

For $g_{k,y,\mathcal{S}_{\text{core}}}^{(t)}$, by the law of total expectation we have

$$
\begin{aligned}
g_{k,y,\mathcal{S}_{\text{core}}}^{(t)} &= \frac{1}{m}\mathbb{E}_{(\mathbf{x},\mathbf{y})\sim\mathcal{D}_{\text{train}}}\mathbb{1}_{h^{(t)}(\mathbf{x})\leq 1}\mathbb{1}_{\mathbf{y}=y}\mathbb{1}_{\langle\mathbf{w}_k^{(t)},\mathbf{x}\rangle\geq 0}\sum_{j\in\mathcal{S}_{\text{core}}}\mu_j\mathbf{z}_j \\
&= \frac{1}{2m}\mathbb{E}_{\mathbf{x}|\mathbf{y}=y}\left[\mathbb{1}_{h^{(t)}(\mathbf{x})\leq 1}\sum_{j\in\mathcal{S}_{\text{core}}}\mu_j\mathbf{z}_j\ \Big|\ \langle\mathbf{w}_k^{(t)},\mathbf{x}\rangle\geq 0\right]\mathbf{Pr}_{\mathbf{x}|\mathbf{y}=y}[\langle\mathbf{w}_k^{(t)},\mathbf{x}\rangle\geq 0] \\
&= \frac{1}{2m}\mathbb{E}_{\mathbf{x}|\mathbf{y}=y}\left[\mathbb{1}_{h^{(t)}(\mathbf{x})\leq 1}\sum_{j\in\mathcal{S}_{\text{core}}}\mu_j\mathbf{z}_j\right] \\
&\quad - \frac{1}{2m}\mathbb{E}_{\mathbf{x}|\mathbf{y}=y}\left[\mathbb{1}_{h^{(t)}(\mathbf{x})\leq 1}\sum_{j\in\mathcal{S}_{\text{core}}}\mu_j\mathbf{z}_j\ \Big|\ \langle\mathbf{w}_k^{(t)},\mathbf{x}\rangle < 0\right]\mathbf{Pr}_{\mathbf{x}|\mathbf{y}=y}[\langle\mathbf{w}_k^{(t)},\mathbf{x}\rangle < 0].
\end{aligned}
$$

Applying Lemma 8 gives

$$
\mathbf{Pr}_{\mathbf{x}|\mathbf{y}=y}[\langle\mathbf{w}_k^{(t)},\mathbf{x}\rangle < 0] = \Phi\left(-\frac{\mathbb{E}_{\mathbf{x}|\mathbf{y}=y}\langle\mathbf{w}_k^{(t)},\mathbf{x}\rangle}{\Theta(1)\sqrt{\sum_{j\in[d_0]}\langle\mathbf{w}_k^{(t)},\boldsymbol{m}_j\rangle^2}}\right) \pm O\left(\frac{1}{\sqrt{d_0}}\right).
$$

Recall that for $\mathbf{x}\sim\mathcal{D}_{\text{train}}|\mathbf{y}=y$,

$$
\langle\mathbf{w}_k^{(t)},\mathbf{x}\rangle = \sum_{j\in\mathcal{S}_{\text{core}}}y\mathbf{z}_j\langle\mathbf{w}_k^{(t)},\boldsymbol{m}_j\rangle + \sum_{j\in\mathcal{S}_{\text{bg}}}\mathbf{z}_j\langle\mathbf{w}_k^{(t)},\boldsymbol{m}_j\rangle.
$$

By Definition 2, we have for every $k\in\mathcal{N}_y^{(t)}$, $\mathbb{E}_{\mathbf{x}|\mathbf{y}=y}\langle\mathbf{w}_k^{(t)},\mathbf{x}\rangle\geq\Theta\left(\sqrt{\frac{d_0}{d}}\right) > 0$, which indicates that $\Phi\left(-\frac{\mathbb{E}_{\mathbf{x}|\mathbf{y}=y}\langle\mathbf{w}_k^{(t)},\mathbf{x}\rangle}{\Theta(1)\sqrt{\sum_{j\in[d_0]}\langle\mathbf{w}_k^{(t)},\boldsymbol{m}_j\rangle^2}}\right) < \frac{1}{2}$. Together with $h^{(t)}(\mathbf{x})\leq 1$, this gives

$$
\begin{aligned}
g_{k,y,\mathcal{S}_{\text{core}}}^{(t)} &\geq \frac{1}{2m}\mathbb{E}_{\mathbf{x}|\mathbf{y}=y}\left[\mathbb{1}_{h^{(t)}(\mathbf{x})\leq 1}\sum_{j\in\mathcal{S}_{\text{core}}}\mu_j\mathbf{z}_j\right] \\
&\quad - \frac{1}{2m}\mathbb{E}_{\mathbf{x}|\mathbf{y}=y}\left[\mathbb{1}_{h^{(t)}(\mathbf{x})\leq 1}\sum_{j\in\mathcal{S}_{\text{core}}}\mu_j\mathbf{z}_j\ \Big|\ \langle\mathbf{w}_k^{(t)},\mathbf{x}\rangle < 0\right]\cdot\left(\frac{1}{2}\pm O\left(\frac{1}{\sqrt{d_0}}\right)\right) \\
&\geq \sum_{j\in\mathcal{S}_{\text{core}}}\frac{\mu_j^2}{2m} - \sum_{j\in\mathcal{S}_{\text{core}}}\frac{\mu_j^2}{4m} \pm \sum_{j\in\mathcal{S}_{\text{core}}}\frac{\mu_j^2}{\Theta(m\sqrt{d_0})} \\
&= \Theta\left(\frac{d_0}{m}\right).
\end{aligned}
\tag{24}
$$

Meanwhile, we also have the upper bound

$$
\begin{aligned}
g_{k,y,\mathcal{S}_{\text{core}}}^{(t)} &= \frac{1}{m}\mathbb{E}_{(\mathbf{x},\mathbf{y})}\mathbb{1}_{h^{(t)}(\mathbf{x})\leq 1}\mathbb{1}_{\mathbf{y}=j}\mathbb{1}_{\langle\mathbf{w}_k^{(t)},\mathbf{x}\rangle\geq 0}\sum_{j\in\mathcal{S}_{\text{core}}}\mu_j\mathbf{z}_j \\
&\leq \frac{1}{2m}\mathbb{E}_{\mathbf{x}|\mathbf{y}=y}\sum_{j\in\mathcal{S}_{\text{core}}}\mu_j\mathbf{z}_j \\
&= \Theta\left(\frac{d_0}{m}\right).
\end{aligned}
\tag{25}
$$

Combining Eqs. (24) and (25) gives

$$
g_{k,y,\mathcal{S}_{\text{core}}}^{(t)} = \Theta\left(\frac{d_0}{m}\right).
$$

Finally, plugging the above equation and $m = O(d)$ into Eq. (23) completes the proof. $\qquad\square$

**Lemma 10** (Gradient projection onto background features, neurons in $\mathcal{N}_y^{(t)}$). *For every iteration* $t \leq O(\frac{m}{\eta d_0})$, *the following holds for every* $y \in \mathcal{Y}$ *and every neuron* $k \in \mathcal{N}_y^{(t)}$:

$$-\left\langle \nabla_{\mathbf{w}_k^{(t)}} \frac{1}{N} \sum_{i \in [N]} \ell\left(h^{(t)}(\mathbf{x}_i^{(t)}), \mathbf{y}_i^{(t)}\right), \sum_{j \in \mathcal{S}_{\text{bg}}} \mu_j \boldsymbol{m}_j \right\rangle = g_{k,y,\mathcal{S}_{\text{bg}}}^{(t)} - g_{k,-y,\mathcal{S}_{\text{bg}}}^{(t)}, \quad (26)$$

*where*

$$g_{k,y,\mathcal{S}_{\text{bg}}}^{(t)} = \Theta\left(\frac{d_0}{m}\right). \quad (27)$$

*Proof.* Similar to the proof of Lemma 9, we apply Lemma 3 to the LHS of Eq. (26) and using $\text{sign}(a_k) = y$ for every $k \in \mathcal{N}_y^{(t)}$, which gives

$$-\left\langle \nabla_{\mathbf{w}_k^{(t)}} \frac{1}{N} \sum_{i \in [N]} \ell\left(h^{(t)}(\mathbf{x}_i^{(t)}), \mathbf{y}_i^{(t)}\right), \sum_{j \in \mathcal{S}_{\text{bg}}} \mu_j \boldsymbol{m}_j \right\rangle$$

$$= -\left\langle \nabla_{\mathbf{w}_k^{(t)}} \mathbb{E}_{(\mathbf{x},\mathbf{y}) \sim \mathcal{D}_{\text{train}}} \ell\left(h^{(t)}(\mathbf{x}), \mathbf{y}\right), \sum_{j \in \mathcal{S}_{\text{bg}}} \mu_j \boldsymbol{m}_j \right\rangle \pm \frac{O(d_0)}{\text{poly}(d)}$$

$$= a_k \mathbb{E}_{(\mathbf{x},\mathbf{y}) \sim \mathcal{D}_{\text{train}}} \mathbf{y} \mathbb{1}_{h^{(t)}(\mathbf{x}) \leq 1} \mathbb{1}_{\langle \mathbf{w}_k^{(t)}, \mathbf{x} \rangle \geq 0} \sum_{j \in \mathcal{S}_{\text{bg}}} \mu_j \mathbf{z}_j \pm \frac{O(d_0)}{\text{poly}(d)}$$

$$= \frac{1}{m} \mathbb{E}_{(\mathbf{x},\mathbf{y}) \sim \mathcal{D}_{\text{train}}} \mathbb{1}_{h^{(t)}(\mathbf{x}) \leq 1} \mathbb{1}_{\mathbf{y}=y} \mathbb{1}_{\langle \mathbf{w}_k^{(t)}, \mathbf{x} \rangle \geq 0} \sum_{j \in \mathcal{S}_{\text{bg}}} \mu_j \mathbf{z}_j \quad (28)$$

$$- \frac{1}{m} \mathbb{E}_{(\mathbf{x},\mathbf{y}) \sim \mathcal{D}_{\text{train}}} \mathbb{1}_{h^{(t)}(\mathbf{x}) \leq 1} \mathbb{1}_{\mathbf{y}=-y} \mathbb{1}_{\langle \mathbf{w}_k^{(t)}, \mathbf{x} \rangle \geq 0} \sum_{j \in \mathcal{S}_{\text{bg}}} \mu_j \mathbf{z}_j \pm \frac{O(d_0)}{\text{poly}(d)}$$

$$= g_{k,y,\mathcal{S}_{\text{bg}}}^{(t)} - g_{k,-y,\mathcal{S}_{\text{bg}}}^{(t)} \pm \frac{O(d_0)}{\text{poly}(d)}.$$

Also, by a nearly identical argument to Lemma 9, we can obtain

$$g_{k,y,\mathcal{S}_{\text{bg}}}^{(t)} = \Theta\left(\frac{d_0}{m}\right). \quad (29)$$

This completes the proof. $\square$

Next, we also introduce a lemma that bound the gradient projection onto core features for all neurons:

**Lemma 11** (Gradient projection onto core features, all neurons). *For every iteration* $t \leq O(\frac{m}{\eta d_0})$, *the following holds for every* $y \in \mathcal{Y}$ *and every neuron* $k \in [m]$ *with* $\text{sign}(a_k) = y$:

$$-\left\langle \nabla_{\mathbf{w}_k^{(t)}} \frac{1}{N} \sum_{i \in [N]} \ell\left(h^{(t)}(\mathbf{x}_i^{(t)}), \mathbf{y}_i^{(t)}\right), \sum_{j \in \mathcal{S}_{\text{core}}} \mu_j \boldsymbol{m}_j \right\rangle = y \cdot O\left(\frac{d_0}{m}\right). \quad (30)$$

*Proof.* By an identical proof to Lemma 9, we have

$$-\left\langle \nabla_{\mathbf{w}_k^{(t)}} \frac{1}{N} \sum_{i \in [N]} \ell\left(h^{(t)}(\mathbf{x}_i^{(t)}), \mathbf{y}_i^{(t)}\right), \sum_{j \in \mathcal{S}_{\text{core}}} \mu_j \boldsymbol{m}_j \right\rangle = y\left(g_{k,y,\mathcal{S}_{\text{core}}}^{(t)} + g_{k,-y,\mathcal{S}_{\text{core}}}^{(t)}\right) \pm \frac{\Theta(d_0)}{\text{poly}(d)}.$$

By Eq. (25), we have the upper bound $g_{k,y,\mathcal{S}_{\text{core}}}^{(t)} \leq \Theta\left(\frac{d_0}{m}\right)$. By a similar argument, we also have $g_{k,-y,\mathcal{S}_{\text{core}}}^{(t)} \leq \Theta\left(\frac{d_0}{m}\right)$. Plugging those upper bounds and $m = O(d)$ into the above equation completes the proof. $\square$

We then introduce a lemma that bounds the expected correlation between each neuron in $\mathcal{N}_y^{(t)}$ and its positive examples.

**Lemma 12** (Correlation with positive examples, neurons in $\mathcal{N}_y^{(t)}$). *For every iteration $t \leq O(\frac{m}{\eta d_0})$, every $y \in \mathcal{Y}$, and every $k \in \mathcal{N}_y^{(t)}$, the following holds:*

$$\mathbb{E}_{\mathbf{x}|\mathbf{y}=y \sim \mathcal{D}_{\text{train}}}[\langle \mathbf{w}_k^{(t+1)}, \mathbf{x} \rangle] \geq (1 - \lambda\eta)\mathbb{E}_{\mathbf{x}|\mathbf{y}=y \sim \mathcal{D}_{\text{train}}}[\langle \mathbf{w}_k^{(t)}, \mathbf{x} \rangle] + \Theta\left(\frac{\eta d_0}{m}\right). \tag{31}$$

*Proof.* Recall Definition 1 that given the label $y \in \mathcal{Y}$, $\mathbf{x}$ is generated by

$$\mathbf{x} = \sum_{j \in \mathcal{S}_{\text{core}}} y\mathbf{z}_j \boldsymbol{m}_j + \sum_{j \in \mathcal{S}_{\text{bg}}} \mathbf{z}_j \boldsymbol{m}_j.$$

We can thus obtain

$$\mathbb{E}_{\mathbf{x}|\mathbf{y}=y \sim \mathcal{D}_{\text{train}}}[\langle \mathbf{w}_k^{(t+1)}, \mathbf{x} \rangle] - \mathbb{E}_{\mathbf{x}|\mathbf{y}=y \sim \mathcal{D}_{\text{train}}}[\langle \mathbf{w}_k^{(t)}, \mathbf{x} \rangle]$$

$$= \underbrace{y\left(\langle \mathbf{w}_k^{(t+1)}, \sum_{j \in \mathcal{S}_{\text{core}}} \mu_j \boldsymbol{m}_j \rangle - \langle \mathbf{w}_k^{(t)}, \sum_{j \in \mathcal{S}_{\text{core}}} \mu_j \boldsymbol{m}_j \rangle\right)}_{\Delta_{\text{core}}^{(t)}}$$

$$+ \underbrace{\left(\langle \mathbf{w}_k^{(t+1)}, \sum_{j \in \mathcal{S}_{\text{bg}}} \mu_j \boldsymbol{m}_j \rangle - \langle \mathbf{w}_k^{(t)}, \sum_{j \in \mathcal{S}_{\text{bg}}} \mu_j \boldsymbol{m}_j \rangle\right)}_{\Delta_{\text{bg}}^{(t)}}.$$

For $\Delta_{\text{core}}^{(t)}$, by the SGD iteration (4) we have

$$\Delta_{\text{core}}^{(t)} = -\eta y \langle \nabla_{\mathbf{w}_k^{(t)}} \frac{1}{N} \sum_{i \in [N]} \ell\left(h^{(t)}(\mathbf{x}_i^{(t)}), \mathbf{y}_i^{(t)}\right), \sum_{j \in \mathcal{S}_{\text{core}}} \mu_j \boldsymbol{m}_j \rangle - \lambda\eta y \langle \mathbf{w}_k^{(t)}, \sum_{j \in \mathcal{S}_{\text{core}}} \mu_j \boldsymbol{m}_j \rangle.$$

Applying Lemma 9 gives

$$\Delta_{\text{core}}^{(t)} = \eta\left(g_{k,y,\mathcal{S}_{\text{core}}}^{(t)} + g_{k,-y,\mathcal{S}_{\text{core}}}^{(t)}\right) - \lambda\eta y \langle \mathbf{w}_k^{(t)}, \sum_{j \in \mathcal{S}_{\text{core}}} \mu_j \boldsymbol{m}_j \rangle,$$

which results in the iterative expression

$$y \langle \mathbf{w}_k^{(t+1)}, \sum_{j \in \mathcal{S}_{\text{core}}} \mu_j \boldsymbol{m}_j \rangle = y(1 - \lambda\eta) \langle \mathbf{w}_k^{(t)}, \sum_{j \in \mathcal{S}_{\text{core}}} \mu_j \boldsymbol{m}_j \rangle + \eta\left(g_{k,y,\mathcal{S}_{\text{core}}}^{(t)} + g_{k,-y,\mathcal{S}_{\text{core}}}^{(t)}\right). \tag{32}$$

For $\Delta_{\text{bg}}^{(t)}$, by the SGD iteration (4) we have

$$\Delta_{\text{bg}}^{(t)} = -\eta \langle \nabla_{\mathbf{w}_k^{(t)}} \frac{1}{N} \sum_{i \in [N]} \ell\left(h^{(t)}(\mathbf{x}_i^{(t)}), \mathbf{y}_i^{(t)}\right), \sum_{j \in \mathcal{S}_{\text{bg}}} \mu_j \boldsymbol{m}_j \rangle - \lambda\eta \langle \mathbf{w}_k^{(t)}, \sum_{j \in \mathcal{S}_{\text{bg}}} \mu_j \boldsymbol{m}_j \rangle.$$

Applying Lemma 10 gives

$$\Delta_{\text{bg}}^{(t)} = \eta\left(g_{k,y,\mathcal{S}_{\text{bg}}}^{(t)} - g_{k,-y,\mathcal{S}_{\text{bg}}}^{(t)}\right) - \lambda\eta \langle \mathbf{w}_k^{(t)}, \sum_{j \in \mathcal{S}_{\text{bg}}} \mu_j \boldsymbol{m}_j \rangle,$$

which results in the iterative expression

$$\langle \mathbf{w}_k^{(t+1)}, \sum_{j \in \mathcal{S}_{\text{bg}}} \mu_j \boldsymbol{m}_j \rangle = (1 - \lambda\eta) \langle \mathbf{w}_k^{(t)}, \sum_{j \in \mathcal{S}_{\text{bg}}} \mu_j \boldsymbol{m}_j \rangle + \eta\left(g_{k,y,\mathcal{S}_{\text{bg}}}^{(t)} - g_{k,-y,\mathcal{S}_{\text{bg}}}^{(t)}\right). \tag{33}$$

Combining Eqs. (36) and (37) gives

$$\mathbb{E}_{\mathbf{x}|\mathbf{y}=y\sim\mathcal{D}_{\text{train}}}[\langle\mathbf{w}_k^{(t+1)},\mathbf{x}\rangle] = y\langle\mathbf{w}_k^{(t+1)}, \sum_{j\in\mathcal{S}_{\text{core}}}\mu_j\boldsymbol{m}_j\rangle + \langle\mathbf{w}_k^{(t+1)}, \sum_{j\in\mathcal{S}_{\text{bg}}}\mu_j\boldsymbol{m}_j\rangle$$

$$= y(1-\lambda\eta)\langle\mathbf{w}_k^{(t)}, \sum_{j\in\mathcal{S}_{\text{core}}}\mu_j\boldsymbol{m}_j\rangle + \eta\left(g_{k,y,\mathcal{S}_{\text{core}}}^{(t)} + g_{k,-y,\mathcal{S}_{\text{core}}}^{(t)}\right)$$

$$+ (1-\lambda\eta)\langle\mathbf{w}_k^{(t)}, \sum_{j\in\mathcal{S}_{\text{bg}}}\mu_j\boldsymbol{m}_j\rangle + \eta\left(g_{k,y,\mathcal{S}_{\text{bg}}}^{(t)} - g_{k,-y,\mathcal{S}_{\text{bg}}}^{(t)}\right)$$

$$= y(1-\lambda\eta)\mathbb{E}_{\mathbf{x}|\mathbf{y}=y\sim\mathcal{D}_{\text{train}}}[\langle\mathbf{w}_k^{(t)},\mathbf{x}\rangle]$$

$$+ \eta\left(g_{k,y,\mathcal{S}_{\text{core}}}^{(t)} + g_{k,-y,\mathcal{S}_{\text{core}}}^{(t)} + g_{k,y,\mathcal{S}_{\text{bg}}}^{(t)} - g_{k,-y,\mathcal{S}_{\text{bg}}}^{(t)}\right)$$

$$\geq (1-\lambda\eta)\mathbb{E}_{\mathbf{x}|\mathbf{y}=y\sim\mathcal{D}_{\text{train}}}[\langle\mathbf{w}_k^{(t)},\mathbf{x}\rangle] + \Theta\left(\frac{\eta d_0}{m}\right),$$

where in the last inequality we use

$$g_{k,-y,\mathcal{S}_{\text{core}}}^{(t)} - g_{k,-y,\mathcal{S}_{\text{bg}}}^{(t)} = \frac{1}{m}\mathbb{E}_{(\mathbf{x},\mathbf{y})}\mathbb{1}_{h^{(t)}(\mathbf{x})\leq 1}\mathbb{1}_{\mathbf{y}=-y}\mathbb{1}_{\langle\mathbf{w}_k^{(t)},\mathbf{x}\rangle\geq 0}\left(\sum_{j\in\mathcal{S}_{\text{core}}}\mu_j\mathbf{z}_j - \sum_{j\in\mathcal{S}_{\text{bg}}}\mu_j\mathbf{z}_j\right) \geq 0$$

as well as $g_{k,y,\mathcal{S}_{\text{core}}}^{(t)} = \Theta\left(\frac{d_0}{m}\right)$ and $g_{k,y,\mathcal{S}_{\text{bg}}}^{(t)} = \Theta\left(\frac{d_0}{m}\right)$ from Lemma 9 and Lemma 10. $\qquad\square$

We also introduce a general upper bound on the expected correlation between every neuron in the network and its positive examples.

**Lemma 13** (Correlation with positive examples, all neurons). *For every iteration $t \leq O(\frac{m}{\eta d_0})$, the following holds for every $y \in \mathcal{Y}$ and every $k \in \mathcal{N}_y^{(t)}$ with $\text{sign}(a_k) = y$:*

$$\mathbb{E}_{\mathbf{x}|\mathbf{y}=y\sim\mathcal{D}_{\text{train}}}[\langle\mathbf{w}_k^{(t+1)},\mathbf{x}\rangle] \leq (1-\lambda\eta)\mathbb{E}_{\mathbf{x}|\mathbf{y}=y\sim\mathcal{D}_{\text{train}}}[\langle\mathbf{w}_k^{(t)},\mathbf{x}\rangle] + O\left(\frac{\eta d_0}{m}\right). \qquad (34)$$

*Proof.* By an identical proof to Lemma 12, we have

$$\mathbb{E}_{\mathbf{x}|\mathbf{y}=y\sim\mathcal{D}_{\text{train}}}[\langle\mathbf{w}_k^{(t+1)},\mathbf{x}\rangle] = y(1-\lambda\eta)\mathbb{E}_{\mathbf{x}|\mathbf{y}=y\sim\mathcal{D}_{\text{train}}}[\langle\mathbf{w}_k^{(t)},\mathbf{x}\rangle]$$

$$+ \eta\left(g_{k,y,\mathcal{S}_{\text{core}}}^{(t)} + g_{k,-y,\mathcal{S}_{\text{core}}}^{(t)} + g_{k,y,\mathcal{S}_{\text{bg}}}^{(t)} - g_{k,-y,\mathcal{S}_{\text{bg}}}^{(t)}\right)$$

By Eq. (25), we have the upper bound $g_{k,y,\mathcal{S}_{\text{core}}}^{(t)} \leq \Theta\left(\frac{d_0}{m}\right)$. By a similar argument, we also have the upper bounds $g_{k,-y,\mathcal{S}_{\text{core}}}^{(t)} \leq \Theta\left(\frac{d_0}{m}\right)$ and $g_{k,y,\mathcal{S}_{\text{bg}}}^{(t)} \leq \Theta\left(\frac{d_0}{m}\right)$. Plugging those upper bounds and the trivial lower bound $g_{k,-y,\mathcal{S}_{\text{bg}}}^{(t)} \geq 0$ into the above equation completes the proof. $\qquad\square$

The above two lemmas directly lead to the following result saying that if a neuron is initialized to have large enough correlation to its positive examples (i.e., belonging to $\mathcal{N}_y^{(0)}$), then this large enough correlation will be retained during training.

**Lemma 14** (Neuron properties during training). *For every label $y \in \mathcal{Y}$, every iteration $t \leq \widetilde{O}(\frac{m}{\eta d_0})$, and every step size $\eta \leq \frac{1}{\text{poly}(d_0)}$, we have $\mathcal{N}_y^{(t+1)} \supseteq \mathcal{N}_y^{(t)}$.*

*Proof.* By Lemma 6, we have at initialization

$$\max_{k\in[m]}\left|\mathbb{E}_{\mathbf{x}|\mathbf{y}=y\sim\mathcal{D}_{\text{train}}}\langle\mathbf{w}_k^{(0)},\mathbf{x}\rangle\right| \leq \widetilde{O}\left(\sqrt{\frac{d_0}{d}}\right), \quad \forall y \in \mathcal{Y}.$$

By Lemma 13 and our choice of $T = \Theta(\frac{m}{\eta d_0})$, we have

$$\mathbb{E}_{\mathbf{x}|\mathbf{y}=y\sim\mathcal{D}_{\text{train}}}[\langle\mathbf{w}_k^{(t)},\mathbf{x}\rangle] \leq O\left(\frac{\eta d_0 T}{m}\right) + \max_{k\in[m]}\left|\mathbb{E}_{\mathbf{x}|\mathbf{y}=y\sim\mathcal{D}_{\text{train}}}\langle\mathbf{w}_k^{(0)},\mathbf{x}\rangle\right| = O(1).$$

By Lemma 12, we have

$$\mathbb{E}_{\mathbf{x}|\mathbf{y}=y\sim\mathcal{D}_{\text{train}}}[\langle\mathbf{w}_k^{(t+1)},\mathbf{x}\rangle] \geq (1-\lambda\eta)\mathbb{E}_{\mathbf{x}|\mathbf{y}=y\sim\mathcal{D}_{\text{train}}}[\langle\mathbf{w}_k^{(t)},\mathbf{x}\rangle] + \Theta\left(\frac{\eta d_0}{m}\right).$$

Recall that $\lambda = O(\frac{d_0}{m^{1.01}})$. Therefore, as long as $\mathbb{E}_{\mathbf{x}|\mathbf{y}=y\sim\mathcal{D}_{\text{train}}}[\langle\mathbf{w}_k^{(t)},\mathbf{x}\rangle] = \widetilde{O}(1)$,

$$\mathbb{E}_{\mathbf{x}|\mathbf{y}=y\sim\mathcal{D}_{\text{train}}}[\langle\mathbf{w}_k^{(t+1)},\mathbf{x}\rangle] - \mathbb{E}_{\mathbf{x}|\mathbf{y}=y\sim\mathcal{D}_{\text{train}}}[\langle\mathbf{w}_k^{(t)},\mathbf{x}\rangle] \geq \Theta\left(\frac{\eta d_0}{m}\right) - \lambda\eta\mathbb{E}_{\mathbf{x}|\mathbf{y}=y\sim\mathcal{D}_{\text{train}}}[\langle\mathbf{w}_k^{(t)},\mathbf{x}\rangle]$$

$$= \Theta\left(\frac{\eta d_0}{m}\right) > 0.$$

Finally, recall that $\mathbb{E}_{\mathbf{x}|\mathbf{y}=y\sim\mathcal{D}_{\text{train}}}[\langle\mathbf{w}_k^{(t)},\mathbf{x}\rangle] = \sum_{j\in\mathcal{S}_{\text{core}}}y\mu_j\langle\mathbf{w}_k^{(0)},\boldsymbol{m}_j\rangle + \sum_{j\in\mathcal{S}_{\text{bg}}}\mu_j\langle\mathbf{w}_k^{(0)},\boldsymbol{m}_j\rangle$.
By Definition 2, we immediately have $\mathcal{N}_y^{(t+1)} \supseteq \mathcal{N}_y^{(t)}$. $\qquad\square$

Finally, we introduce a lemma that bounds the expected correlation between every neuron in the network and its negative examples.

**Lemma 15** (Correlation with negative examples, all neurons). *For every iteration t, every $y \in \mathcal{Y}$, and every $k \in [m]$ such that $\operatorname{sign}(a_k) = y$, the following holds:*

$$\mathbb{E}_{\mathbf{x}|\mathbf{y}=-y\sim\mathcal{D}_{\text{train}}}[\langle\mathbf{w}_k^{(t+1)},\mathbf{x}\rangle] = (1-\lambda\eta)\mathbb{E}_{\mathbf{x}|\mathbf{y}=-y\sim\mathcal{D}_{\text{train}}}[\langle\mathbf{w}_k^{(t)},\mathbf{x}\rangle]$$
$$- \Theta\left(\frac{\eta d_0}{m}\right)\mathbf{Pr}_{\mathbf{x}|\mathbf{y}=-y}[\langle\mathbf{w}_k^{(t)},\mathbf{x}\rangle \geq 0]. \tag{35}$$

*Proof.* Similar to the proof of Lemma 12, we have

$$\mathbb{E}_{\mathbf{x}|\mathbf{y}=-y\sim\mathcal{D}_{\text{train}}}[\langle\mathbf{w}_k^{(t+1)},\mathbf{x}\rangle] - \mathbb{E}_{\mathbf{x}|\mathbf{y}=-y\sim\mathcal{D}_{\text{train}}}[\langle\mathbf{w}_k^{(t)},\mathbf{x}\rangle]$$
$$= \underbrace{-y\Big(\langle\mathbf{w}_k^{(t+1)}, \sum_{j\in\mathcal{S}_{\text{core}}}\mu_j\boldsymbol{m}_j\rangle - \langle\mathbf{w}_k^{(t)}, \sum_{j\in\mathcal{S}_{\text{core}}}\mu_j\boldsymbol{m}_j\rangle\Big)}_{\Delta_{\text{core}}^{(t)}}$$
$$+ \underbrace{\Big(\langle\mathbf{w}_k^{(t+1)}, \sum_{j\in\mathcal{S}_{\text{bg}}}\mu_j\boldsymbol{m}_j\rangle - \langle\mathbf{w}_k^{(t)}, \sum_{j\in\mathcal{S}_{\text{bg}}}\mu_j\boldsymbol{m}_j\rangle\Big)}_{\Delta_{\text{bg}}^{(t)}}.$$

For $\Delta_{\text{core}}^{(t)}$, by the SGD iteration (4) we have

$$\Delta_{\text{core}}^{(t)} = \eta y\Big\langle\nabla_{\mathbf{w}_k^{(t)}}\frac{1}{N}\sum_{i\in[N]}\ell\left(h^{(t)}(\mathbf{x}_i^{(t)}),\mathbf{y}_i^{(t)}\right), \sum_{j\in\mathcal{S}_{\text{core}}}\mu_j\boldsymbol{m}_j\Big\rangle + \lambda\eta y\langle\mathbf{w}_k^{(t)}, \sum_{j\in\mathcal{S}_{\text{core}}}\mu_j\boldsymbol{m}_j\rangle.$$

Applying Lemma 9 gives

$$\Delta_{\text{core}}^{(t)} = -\eta\left(g_{k,y,\mathcal{S}_{\text{core}}}^{(t)} + g_{k,-y,\mathcal{S}_{\text{core}}}^{(t)}\right) + \lambda\eta y\langle\mathbf{w}_k^{(t)}, \sum_{j\in\mathcal{S}_{\text{core}}}\mu_j\boldsymbol{m}_j\rangle,$$

which results in the iterative expression

$$-y\langle\mathbf{w}_k^{(t+1)}, \sum_{j\in\mathcal{S}_{\text{core}}}\mu_j\boldsymbol{m}_j\rangle = -y(1-\lambda\eta)\langle\mathbf{w}_k^{(t)}, \sum_{j\in\mathcal{S}_{\text{core}}}\mu_j\boldsymbol{m}_j\rangle - \eta\left(g_{k,y,\mathcal{S}_{\text{core}}}^{(t)} + g_{k,-y,\mathcal{S}_{\text{core}}}^{(t)}\right). \tag{36}$$

For $\Delta_{\text{bg}}^{(t)}$, by the SGD iteration (4) we have

$$\Delta_{\text{bg}}^{(t)} = -\eta\Big\langle\nabla_{\mathbf{w}_k^{(t)}}\frac{1}{N}\sum_{i\in[N]}\ell\left(h^{(t)}(\mathbf{x}_i^{(t)}),\mathbf{y}_i^{(t)}\right), \sum_{j\in\mathcal{S}_{\text{bg}}}\mu_j\boldsymbol{m}_j\Big\rangle - \lambda\eta\langle\mathbf{w}_k^{(t)}, \sum_{j\in\mathcal{S}_{\text{bg}}}\mu_j\boldsymbol{m}_j\rangle.$$

Applying Lemma 10 gives

$$\Delta_{\text{bg}}^{(t)} = \eta \left( g_{k,y,\mathcal{S}_{\text{bg}}}^{(t)} - g_{k,-y,\mathcal{S}_{\text{bg}}}^{(t)} \right) - \lambda\eta \langle \mathbf{w}_k^{(t)}, \sum_{j\in\mathcal{S}_{\text{bg}}} \mu_j \boldsymbol{m}_j \rangle,$$

which results in the iterative expression

$$\langle \mathbf{w}_k^{(t+1)}, \sum_{j\in\mathcal{S}_{\text{bg}}} \mu_j \boldsymbol{m}_j \rangle = (1-\lambda\eta)\langle \mathbf{w}_k^{(t)}, \sum_{j\in\mathcal{S}_{\text{bg}}} \mu_j \boldsymbol{m}_j \rangle + \eta \left( g_{k,y,\mathcal{S}_{\text{bg}}}^{(t)} - g_{k,-y,\mathcal{S}_{\text{bg}}}^{(t)} \right). \tag{37}$$

Combining Eqs. (36) and (37) gives

$$\mathbb{E}_{\mathbf{x}|\mathbf{y}=-y\sim\mathcal{D}_{\text{train}}}[\langle \mathbf{w}_k^{(t+1)}, \mathbf{x} \rangle] = -y\langle \mathbf{w}_k^{(t+1)}, \sum_{j\in\mathcal{S}_{\text{core}}} \mu_j \boldsymbol{m}_j \rangle + \langle \mathbf{w}_k^{(t+1)}, \sum_{j\in\mathcal{S}_{\text{bg}}} \mu_j \boldsymbol{m}_j \rangle$$

$$= -y(1-\lambda\eta)\langle \mathbf{w}_k^{(t)}, \sum_{j\in\mathcal{S}_{\text{core}}} \mu_j \boldsymbol{m}_j \rangle - \eta \left( g_{k,y,\mathcal{S}_{\text{core}}}^{(t)} + g_{k,-y,\mathcal{S}_{\text{core}}}^{(t)} \right)$$

$$+ (1-\lambda\eta)\langle \mathbf{w}_k^{(t)}, \sum_{j\in\mathcal{S}_{\text{bg}}} \mu_j \boldsymbol{m}_j \rangle + \eta \left( g_{k,y,\mathcal{S}_{\text{bg}}}^{(t)} - g_{k,-y,\mathcal{S}_{\text{bg}}}^{(t)} \right)$$

$$= (1-\lambda\eta)\mathbb{E}_{\mathbf{x}|\mathbf{y}=-y\sim\mathcal{D}_{\text{train}}}[\langle \mathbf{w}_k^{(t)}, \mathbf{x} \rangle]$$

$$+ \eta \left( g_{k,y,\mathcal{S}_{\text{bg}}}^{(t)} - g_{k,y,\mathcal{S}_{\text{core}}}^{(t)} \right) - \eta \left( g_{k,-y,\mathcal{S}_{\text{core}}}^{(t)} + g_{k,-y,\mathcal{S}_{\text{bg}}}^{(t)} \right). \tag{38}$$

For $g_{k,y,\mathcal{S}_{\text{bg}}}^{(t)} - g_{k,y,\mathcal{S}_{\text{core}}}^{(t)}$, we have

$$g_{k,y,\mathcal{S}_{\text{bg}}}^{(t)} - g_{k,y,\mathcal{S}_{\text{core}}}^{(t)} = \frac{1}{m}\mathbb{E}_{(\mathbf{x},\mathbf{y})} \mathbb{1}_{h^{(t)}(\mathbf{x})\leq 1} \mathbb{1}_{\mathbf{y}=y} \mathbb{1}_{\langle \mathbf{w}_k^{(t)},\mathbf{x}\rangle\geq 0} \left( \sum_{j\in\mathcal{S}_{\text{bg}}} \mu_j \mathbf{z}_j - \sum_{j\in\mathcal{S}_{\text{core}}} \mu_j \mathbf{z}_j \right) \leq 0. \tag{39}$$

For $g_{k,-y,\mathcal{S}_{\text{core}}}^{(t)} + g_{k,-y,\mathcal{S}_{\text{bg}}}^{(t)}$, by the law of total expectation we have

$$g_{k,-y,\mathcal{S}_{\text{core}}}^{(t)} + g_{k,-y,\mathcal{S}_{\text{bg}}}^{(t)} = \frac{1}{m}\mathbb{E}_{(\mathbf{x},\mathbf{y})\sim\mathcal{D}_{\text{train}}} \mathbb{1}_{h^{(t)}(\mathbf{x})\leq 1} \mathbb{1}_{\mathbf{y}=-y} \mathbb{1}_{\langle \mathbf{w}_k^{(t)},\mathbf{x}\rangle\geq 0} \left( \sum_{j\in\mathcal{S}_{\text{bg}}} \mu_j \mathbf{z}_j + \sum_{j\in\mathcal{S}_{\text{core}}} \mu_j \mathbf{z}_j \right)$$

$$= \frac{1}{m}\mathbb{E}_{(\mathbf{x},\mathbf{y})\sim\mathcal{D}_{\text{train}}} \mathbb{1}_{h^{(t)}(\mathbf{x})\leq 1} \mathbb{1}_{\mathbf{y}=-y} \mathbb{1}_{\langle \mathbf{w}_k^{(t)},\mathbf{x}\rangle\geq 0} \sum_{j\in[d_0]} \mu_j \mathbf{z}_j$$

$$= \frac{1}{2m}\mathbb{E}_{\mathbf{x}|\mathbf{y}=-y} \left[ \mathbb{1}_{h^{(t)}(\mathbf{x})\leq 1} \sum_{j\in[d_0]} \mu_j \mathbf{z}_j \Big| \langle \mathbf{w}_k^{(t)}, \mathbf{x} \rangle \geq 0 \right] \mathbf{Pr}_{\mathbf{x}|\mathbf{y}=-y}[\langle \mathbf{w}_k^{(t)}, \mathbf{x} \rangle \geq 0]$$

$$= \Theta\left(\frac{d_0}{m}\right) \mathbf{Pr}_{\mathbf{x}|\mathbf{y}=-y}[\langle \mathbf{w}_k^{(t)}, \mathbf{x} \rangle \geq 0] \tag{40}$$

Finally, plugging Eqs. (39) and (40) into Eq. (38) gives the desired result. □

We are now ready to introduce the proofs of our main theoretical results.

## B.1 PROOF OF THEOREM 1

For ease of presentation, we first restate the theorem:

**Theorem 1** (Activation asymmetry). *For every $\eta \leq \frac{1}{\text{poly}(d_0)}$ and every $y \in \mathcal{Y}$, there exists $T_0 = \widetilde{\Theta}(\frac{m}{\eta\sqrt{d}})$ such that with high probability, there are $\Theta(m)$ neurons whose weights $\mathbf{w}_k$ satisfy*

$$\mathbf{Pr}_{\mathbf{x}|\mathbf{y}=y\sim\mathcal{D}_{\text{train}}}[\langle \mathbf{w}_k^{(t)}, \mathbf{x} \rangle \geq 0] = 1 - o(1), \quad \mathbf{Pr}_{\mathbf{x}|\mathbf{y}=-y}[\langle \mathbf{w}_k^{(t)}, \mathbf{x} \rangle \geq 0] = o(1), \quad \forall t \geq T_0. \tag{41}$$

*Proof.* For every $y \in \mathcal{Y}$, consider the neuron set $\mathcal{N}_y^{(t)}$ defined in Definition 2. By Lemma 5 and Lemma 14, we have $|\mathcal{N}_y^{(t)}| = \Theta(m)$ for every iteration $t \leq \Theta(\frac{m}{\eta d_0})$. We then prove that after at

most $T_0$ iterations, for every neuron $k \in \mathcal{N}_y^{(T_0)}$ we have $\mathbf{Pr}_{\mathbf{x}|\mathbf{y}=y\sim\mathcal{D}_{\text{train}}}[\langle \mathbf{w}_k^{(T_0)}, \mathbf{x} \rangle \geq 0] = 1 - o(1)$ and $\mathbf{Pr}_{\mathbf{x}|\mathbf{y}=-y\sim\mathcal{D}_{\text{train}}}[\langle \mathbf{w}_k^{(T_0)}, \mathbf{x} \rangle \geq 0] = o(1)$.

**Part 1: proving $\mathbf{Pr}_{\mathbf{x}|\mathbf{y}=y\sim\mathcal{D}_{\text{train}}}[\langle \mathbf{w}_k^{(T_0)}, \mathbf{x} \rangle \geq 0] = 1 - o(1)$.**

Let $T_0 = \Theta(\frac{m\sqrt{\log md_0}}{\eta\sqrt{d}}) = \widetilde{\Theta}(\frac{m}{\eta\sqrt{d}})$. By Lemma 12 and Lemma 13 we have

$$\mathbb{E}_{\mathbf{x}|\mathbf{y}=y\sim\mathcal{D}_{\text{train}}}[\langle \mathbf{w}_k^{(t+1)}, \mathbf{x} \rangle] = (1 - \lambda\eta)\mathbb{E}_{\mathbf{x}|\mathbf{y}=y\sim\mathcal{D}_{\text{train}}}[\langle \mathbf{w}_k^{(t)}, \mathbf{x} \rangle] + \Theta\left(\frac{\eta d_0}{m}\right)$$

$$\leq \mathbb{E}_{\mathbf{x}|\mathbf{y}=y\sim\mathcal{D}_{\text{train}}}[\langle \mathbf{w}_k^{(t)}, \mathbf{x} \rangle] + \Theta\left(\frac{\eta d_0}{m}\right),$$

which gives $\mathbb{E}_{\mathbf{x}|\mathbf{y}=y\sim\mathcal{D}_{\text{train}}}[\langle \mathbf{w}_k^{(t)}, \mathbf{x} \rangle] \leq \widetilde{\Theta}(\frac{d_0}{\sqrt{d}}) \leq O(1)$. Recall that $\lambda = o(\frac{d_0}{m})$, we then have

$$\mathbb{E}_{\mathbf{x}|\mathbf{y}=y\sim\mathcal{D}_{\text{train}}}[\langle \mathbf{w}_k^{(t+1)}, \mathbf{x} \rangle] = (1 - \lambda\eta)\mathbb{E}_{\mathbf{x}|\mathbf{y}=y\sim\mathcal{D}_{\text{train}}}[\langle \mathbf{w}_k^{(t)}, \mathbf{x} \rangle] + \Theta\left(\frac{\eta d_0}{m}\right)$$

$$\geq \mathbb{E}_{\mathbf{x}|\mathbf{y}=y\sim\mathcal{D}_{\text{train}}}[\langle \mathbf{w}_k^{(t)}, \mathbf{x} \rangle] + \Theta\left(\frac{\eta d_0}{m}\right) - o\left(\frac{\eta d_0}{m}\right)$$

$$= \mathbb{E}_{\mathbf{x}|\mathbf{y}=y\sim\mathcal{D}_{\text{train}}}[\langle \mathbf{w}_k^{(t)}, \mathbf{x} \rangle] + \Theta\left(\frac{\eta d_0}{m}\right),$$

which gives $\mathbb{E}_{\mathbf{x}|\mathbf{y}=-y\sim\mathcal{D}_{\text{train}}}[\langle \mathbf{w}_k^{(T_0)}, \mathbf{x} \rangle] \geq \Theta\left(\frac{d_0\sqrt{\log md_0}}{\sqrt{d}}\right)$.

By Lemma 8, we have

$$\mathbf{Pr}_{\mathbf{x}|\mathbf{y}=y}[\langle \mathbf{w}_k^{(T_0)}, \mathbf{x} \rangle \geq 0] = \Phi\left(\frac{\mathbb{E}_{\mathbf{x}|\mathbf{y}=y}\langle \mathbf{w}_k^{(t)}, \mathbf{x} \rangle}{\Theta(1)\sqrt{\sum_{j\in[d_0]}\langle \mathbf{w}_k^{(t)}, \boldsymbol{m}_j \rangle^2}}\right) \pm O\left(\frac{1}{\sqrt{d_0}}\right)$$

$$\geq \Phi\left(\frac{\Theta\left(\frac{d_0\sqrt{\log md_0}}{\sqrt{d}}\right)}{\Theta(\sqrt{d_0})\max_{j\in[d_0]}|\langle \mathbf{w}_k^{(t)}, \boldsymbol{m}_j \rangle|}\right) \pm O\left(\frac{1}{\sqrt{d_0}}\right)$$

$$\geq \Phi\left(\frac{\Theta\left(\frac{d_0\sqrt{\log md_0}}{\sqrt{d}}\right)}{\Theta(\sqrt{d_0})\left(O\left(\sqrt{\frac{d_0\log m}{d}}\right) + \Theta\left(\frac{\sqrt{\log md_0}}{\sqrt{d}}\right)\right)}\right) \pm O\left(\frac{1}{\sqrt{d_0}}\right)$$

$$= \Phi\left(\Theta\left(\sqrt{\log d_0}\right)\right) \pm O\left(\frac{1}{\sqrt{d_0}}\right).$$

Applying Lemma 21 gives $\Phi\left(\Theta\left(\sqrt{\log d_0}\right)\right) = 1 - \Theta(\frac{1}{\sqrt{d_0}})$. We then have

$$\mathbf{Pr}_{\mathbf{x}|\mathbf{y}=y}[\langle \mathbf{w}_k^{(T_0)}, \mathbf{x} \rangle \geq 0] = 1 - O\left(\frac{1}{\sqrt{d_0}}\right) = 1 - o(1).$$

**Part 2: proving $\mathbf{Pr}_{\mathbf{x}|\mathbf{y}=-y\sim\mathcal{D}_{\text{train}}}[\langle \mathbf{w}_k^{(T_0)}, \mathbf{x} \rangle \geq 0] = o(1)$.**

By Lemma 15, we have for every $t$ and $k \in \mathcal{N}_y^{(t)}$:

$$\mathbb{E}_{\mathbf{x}|\mathbf{y}=-y\sim\mathcal{D}_{\text{train}}}[\langle \mathbf{w}_k^{(t+1)}, \mathbf{x} \rangle] = (1 - \lambda\eta)\mathbb{E}_{\mathbf{x}|\mathbf{y}=-y\sim\mathcal{D}_{\text{train}}}[\langle \mathbf{w}_k^{(t)}, \mathbf{x} \rangle]$$

$$- \Theta\left(\frac{\eta d_0}{m}\right)\mathbf{Pr}_{\mathbf{x}|\mathbf{y}=-y}[\langle \mathbf{w}_k^{(t)}, \mathbf{x} \rangle \geq 0]. \tag{42}$$

By Lemma 8, we have

$$\mathbf{Pr}_{\mathbf{x}|\mathbf{y}=-y}[\langle \mathbf{w}_k^{(t)}, \mathbf{x} \rangle \geq 0] = \Phi\left(\frac{\mathbb{E}_{\mathbf{x}|\mathbf{y}=-y}\langle \mathbf{w}_k^{(t)}, \mathbf{x} \rangle}{\Theta(1)\sqrt{\sum_{j\in[d_0]}\langle \mathbf{w}_k^{(t)}, \boldsymbol{m}_j \rangle^2}}\right) \pm O\left(\frac{1}{\sqrt{d_0}}\right). \tag{43}$$

Assume that a neuron $k \in \mathcal{N}_y^{(0)}$ satisfies $\mathbb{E}_{\mathbf{x}|\mathbf{y}=-y}\langle \mathbf{w}_k^{(t)}, \mathbf{x}\rangle \geq 0$. Then by Eq. (43), we have $\mathbf{Pr}_{\mathbf{x}|\mathbf{y}=-y}[\langle \mathbf{w}_k^{(t)}, \mathbf{x}\rangle \geq 0] \geq \frac{1}{2} \pm O(\frac{1}{\sqrt{d_0}})$, which gives

$$\mathbb{E}_{\mathbf{x}|\mathbf{y}=-y\sim\mathcal{D}_{\text{train}}}[\langle \mathbf{w}_k^{(t+1)}, \mathbf{x}\rangle] = (1-\lambda\eta)\mathbb{E}_{\mathbf{x}|\mathbf{y}=-y\sim\mathcal{D}_{\text{train}}}[\langle \mathbf{w}_k^{(t)}, \mathbf{x}\rangle] - \Theta\left(\frac{\eta d_0}{m}\right)$$

$$\leq \mathbb{E}_{\mathbf{x}|\mathbf{y}=-y\sim\mathcal{D}_{\text{train}}}[\langle \mathbf{w}_k^{(t)}, \mathbf{x}\rangle] - \Theta\left(\frac{\eta d_0}{m}\right).$$

By Lemma 6, we have at initialization $t = 0$:

$$\max_{k\in[m]} \left|\mathbb{E}_{\mathbf{x}|\mathbf{y}=-y\sim\mathcal{D}_{\text{train}}}\langle \mathbf{w}_k^{(0)}, \mathbf{x}\rangle\right| \leq \widetilde{O}\left(\sqrt{\frac{d_0}{d}}\right). \tag{44}$$

Therefore, for any step size $\eta = \frac{1}{\text{poly}(d_0)}$, after at most $T_{01} := \widetilde{\Theta}(\frac{m}{\eta\sqrt{d_0 d}})$ iterations, we must have $\mathbb{E}_{\mathbf{x}|\mathbf{y}=-y}\langle \mathbf{w}_k^{(t)}, \mathbf{x}\rangle \leq 0$ for every $k \in \mathcal{N}_y^{(t)}$.

Now, let $T_{02} := \Theta(\frac{m\sqrt{\log md_0}}{\eta\sqrt{d}})$. Suppose that $\mathbf{Pr}_{\mathbf{x}|\mathbf{y}=-y}[\langle \mathbf{w}_k^{(t)}, \mathbf{x}\rangle \geq 0] \geq \Theta(1)$ after $t = T_{01} + T_{02} = \widetilde{\Theta}(\frac{m}{\eta\sqrt{d}})$ steps. We then have for $t = T_{01}, \ldots, T_{01} + T_{02} - 1$ that

$$\mathbb{E}_{\mathbf{x}|\mathbf{y}=-y\sim\mathcal{D}_{\text{train}}}[\langle \mathbf{w}_k^{(t+1)}, \mathbf{x}\rangle] = (1-\lambda\eta)\mathbb{E}_{\mathbf{x}|\mathbf{y}=-y\sim\mathcal{D}_{\text{train}}}[\langle \mathbf{w}_k^{(t)}, \mathbf{x}\rangle] - \Theta\left(\frac{\eta d_0}{m}\right)$$

$$\geq \mathbb{E}_{\mathbf{x}|\mathbf{y}=-y\sim\mathcal{D}_{\text{train}}}[\langle \mathbf{w}_k^{(t)}, \mathbf{x}\rangle] - \Theta\left(\frac{\eta d_0}{m}\right),$$

which gives $\mathbb{E}_{\mathbf{x}|\mathbf{y}=-y\sim\mathcal{D}_{\text{train}}}[\langle \mathbf{w}_k^{(t)}, \mathbf{x}\rangle] \geq -\widetilde{O}\left(\sqrt{\frac{d_0}{d}}\right) - \Theta\left(\frac{d_0\sqrt{\log md_0}}{\sqrt{d}}\right) \geq -O(1)$. Since $\lambda = o(\frac{d_0}{m})$, we then have

$$\mathbb{E}_{\mathbf{x}|\mathbf{y}=-y\sim\mathcal{D}_{\text{train}}}[\langle \mathbf{w}_k^{(t+1)}, \mathbf{x}\rangle] = (1-\lambda\eta)\mathbb{E}_{\mathbf{x}|\mathbf{y}=-y\sim\mathcal{D}_{\text{train}}}[\langle \mathbf{w}_k^{(t)}, \mathbf{x}\rangle] - \Theta\left(\frac{\eta d_0}{m}\right)$$

$$\leq \mathbb{E}_{\mathbf{x}|\mathbf{y}=-y\sim\mathcal{D}_{\text{train}}}[\langle \mathbf{w}_k^{(t)}, \mathbf{x}\rangle] - \Theta\left(\frac{\eta d_0}{m}\right) + o\left(\frac{\eta d_0}{m}\right)$$

$$= \mathbb{E}_{\mathbf{x}|\mathbf{y}=-y\sim\mathcal{D}_{\text{train}}}[\langle \mathbf{w}_k^{(t)}, \mathbf{x}\rangle] - \Theta\left(\frac{\eta d_0}{m}\right),$$

which gives $\mathbb{E}_{\mathbf{x}|\mathbf{y}=-y\sim\mathcal{D}_{\text{train}}}[\langle \mathbf{w}_k^{(T_{01}+T_{02})}, \mathbf{x}\rangle] \leq -\Theta\left(\frac{d_0\sqrt{\log md_0}}{\sqrt{d}}\right)$. Plugging this into Eq. (43), we obtain

$$\mathbf{Pr}_{\mathbf{x}|\mathbf{y}=-y}[\langle \mathbf{w}_k^{(T_{01}+T_{02})}, \mathbf{x}\rangle \geq 0] = \Phi\left(\frac{\mathbb{E}_{\mathbf{x}|\mathbf{y}=-y}\langle \mathbf{w}_k^{(t)}, \mathbf{x}\rangle}{\Theta(1)\sqrt{\sum_{j\in[d_0]}\langle \mathbf{w}_k^{(t)}, \boldsymbol{m}_j\rangle^2}}\right) \pm O\left(\frac{1}{\sqrt{d_0}}\right)$$

$$\leq \Phi\left(-\frac{\Theta\left(\frac{d_0\sqrt{\log md_0}}{\sqrt{d}}\right)}{\Theta(\sqrt{d_0})\max_{j\in[d_0]}|\langle \mathbf{w}_k^{(t)}, \boldsymbol{m}_j\rangle|}\right) \pm O\left(\frac{1}{\sqrt{d_0}}\right)$$

$$\leq \Phi\left(-\frac{\Theta\left(\frac{d_0\sqrt{\log md_0}}{\sqrt{d}}\right)}{\Theta(\sqrt{d_0})\left(O\left(\sqrt{\frac{d_0\log m}{d}}\right) + \Theta\left(\frac{\sqrt{\log d_0}}{\sqrt{d}}\right)\right)}\right) \pm O\left(\frac{1}{\sqrt{d_0}}\right)$$

$$= \Phi\left(-\Theta\left(\sqrt{\log d_0}\right)\right) \pm O\left(\frac{1}{\sqrt{d_0}}\right).$$

Applying Lemma 21 gives $\Phi\left(-\Theta\left(\sqrt{\log d_0}\right)\right) = \Theta(\frac{1}{\sqrt{d_0}})$, which leads to

$$\mathbf{Pr}_{\mathbf{x}|\mathbf{y}=-y}[\langle \mathbf{w}_k^{(T_{01}+T_{02})}, \mathbf{x}\rangle \geq 0] = O\left(\frac{1}{\sqrt{d_0}}\right).$$

This contradicts with our assumption that $\mathbf{Pr}_{\mathbf{x}|\mathbf{y}=-y}[\langle \mathbf{w}_k^{(T_{01}+T_{02})}, \mathbf{x}\rangle \geq 0] \geq \Theta(1)$. Hence, we must have $\mathbf{Pr}_{\mathbf{x}|\mathbf{y}=-y}[\langle \mathbf{w}_k^{(T_{01}+T_{02})}, \mathbf{x}\rangle \geq 0] = o(1)$.

Finally, combining **Part 1** and **Part 2** finishes the proof. $\square$

### B.2 Proof of Theorem 2

**Theorem 2** (Learned features). *For every $\eta \leq \frac{1}{\mathrm{poly}(d_0)}$ and every $y \in \mathcal{Y}$, after $T_1 = \Theta(\frac{m}{\eta d_0})$ iterations with high probability, there are $\Theta(m)$ neurons whose weights $\mathbf{w}_k^{(T_1)}$ satisfy*

$$\sum_{j \in \mathcal{S}_{\mathrm{core}}} \mu_{j1} \langle \mathbf{w}_k^{(T_1)}, \boldsymbol{m}_j \rangle = y \cdot \Theta(1), \quad \sum_{j \in \mathcal{S}_{\mathrm{bg}}} \mu_{j1} \langle \mathbf{w}_k^{(T_1)}, \boldsymbol{m}_j \rangle = \Theta(1). \tag{45}$$

*Proof.* For every $y \in \mathcal{Y}$, consider the neuron set $\mathcal{N}_y^{(t)}$ defined in Definition 2. By Lemma 5 and Lemma 14, we have $|\mathcal{N}_y^{(t)}| = \Theta(m)$ for every iteration $t \leq T_1$. We break the subsequent proof into two parts: in the first part we prove the desired result for core features $\mathcal{S}_{\mathrm{core}}$ for all neurons $k \in \mathcal{N}_y^{(T_1)}$; in the second part we prove the desired result for background features $\mathcal{S}_{\mathrm{bg}}$ for all neurons $k \in \mathcal{N}_y^{(T_1)}$. Recall that we use the shorthand $\mu_j$ to denote $\mu_{j1} = \mathbb{E}_{\mathbf{z} \sim \mathcal{D}_{\mathrm{train}}} \mathbf{z}_j$.

**Part 1: proving $\sum_{j \in \mathcal{S}_{\mathrm{core}}} \mu_j \langle \mathbf{w}_k^{(T_1)}, \boldsymbol{m}_j \rangle = \langle \mathbf{w}_k^{(T_1)}, \sum_{j \in \mathcal{S}_{\mathrm{core}}} \mu_j \boldsymbol{m}_j \rangle = \Theta(1)$.**

The SGD update (4) gives

$$\left\langle \mathbf{w}_k^{(t+1)}, \sum_{j \in \mathcal{S}_{\mathrm{core}}} \mu_j \boldsymbol{m}_j \right\rangle - \left\langle \mathbf{w}_k^{(t)}, \sum_{j \in \mathcal{S}_{\mathrm{core}}} \mu_j \boldsymbol{m}_j \right\rangle$$

$$= -\eta \left\langle \nabla_{\mathbf{w}_k^{(t)}} \frac{1}{N} \sum_{i \in [N]} \ell\left(h^{(t)}(\mathbf{x}_i^{(t)}), \mathbf{y}_i^{(t)}\right), \sum_{j \in \mathcal{S}_{\mathrm{core}}} \mu_j \boldsymbol{m}_j \right\rangle - \lambda \eta \left\langle \mathbf{w}_k^{(t)}, \sum_{j \in \mathcal{S}_{\mathrm{core}}} \mu_j \boldsymbol{m}_j \right\rangle$$

for every $t = 0, \ldots, T_1 - 1$.

Applying Lemma 9, we obtain

$$\left\langle \mathbf{w}_k^{(t+1)}, \sum_{j \in \mathcal{S}_{\mathrm{core}}} \mu_j \boldsymbol{m}_j \right\rangle - \left\langle \mathbf{w}_k^{(t)}, \sum_{j \in \mathcal{S}_{\mathrm{core}}} \mu_j \boldsymbol{m}_j \right\rangle$$

$$= y \cdot \Theta\left(\frac{\eta d_0}{m}\right) + y g_{k,-y,\mathcal{S}_{\mathrm{core}}}^{(t)} - \lambda \eta \left\langle \mathbf{w}_k^{(t)}, \sum_{j \in \mathcal{S}_{\mathrm{core}}} \mu_j \boldsymbol{m}_j \right\rangle$$

$$= y \cdot \Theta\left(\frac{\eta d_0}{m}\right) - \lambda \eta \left\langle \mathbf{w}_k^{(t)}, \sum_{j \in \mathcal{S}_{\mathrm{core}}} \mu_j \boldsymbol{m}_j \right\rangle.$$

Without loss of generality, assume that $y = 1$ (the case of $y = -1$ is similar). By the choice of $T_1 = \Theta(\frac{m}{\eta d_0})$, we have

$$\left\langle \mathbf{w}_k^{(T_1)}, \sum_{j \in \mathcal{S}_{\mathrm{core}}} \mu_j \boldsymbol{m}_j \right\rangle \leq \Theta\left(\frac{\eta T_1 d_0}{m}\right) + \left\langle \mathbf{w}_k^{(0)}, \sum_{j \in \mathcal{S}_{\mathrm{core}}} \mu_j \boldsymbol{m}_j \right\rangle$$

$$\leq \Theta(1) + \widetilde{O}\left(\frac{d_0}{d}\right) = \Theta(1),$$

where in the second inequality we apply the concentration inequality of the maximum absolute Gaussian (Lemma 23). By our choice of $\lambda = o(\frac{d_0}{m})$, we have

$$\left\langle \mathbf{w}_k^{(t+1)}, \sum_{j \in \mathcal{S}_{\mathrm{core}}} \mu_j \boldsymbol{m}_j \right\rangle - \left\langle \mathbf{w}_k^{(t)}, \sum_{j \in \mathcal{S}_{\mathrm{core}}} \mu_j \boldsymbol{m}_j \right\rangle$$

$$= \Theta\left(\frac{\eta d_0}{m}\right) - \lambda \eta \left\langle \mathbf{w}_k^{(t)}, \sum_{j \in \mathcal{S}_{\mathrm{core}}} \mu_j \boldsymbol{m}_j \right\rangle$$

$$\geq \Theta\left(\frac{\eta d_0}{m}\right) - o\left(\frac{\eta d_0}{m}\right) = \Theta\left(\frac{\eta d_0}{m}\right).$$

Summing the above inequality from $t = 0$ to $t = T_1 - 1$ yields

$$\left\langle \mathbf{w}_k^{(T_1)}, \sum_{j \in \mathcal{S}_{\text{core}}} \mu_j \boldsymbol{m}_j \right\rangle = \Theta(1).$$

Similarly, for $y = -1$ we have $\left\langle \mathbf{w}_k^{(T_1)}, \sum_{j \in \mathcal{S}_{\text{core}}} \mu_j \boldsymbol{m}_j \right\rangle = -\Theta(1)$.

**Part 2: proving** $\sum_{j \in \mathcal{S}_{\text{bg}}} \mu_j \langle \mathbf{w}_k^{(T_1)}, \boldsymbol{m}_j \rangle = \langle \mathbf{w}_k^{(T_1)}, \sum_{j \in \mathcal{S}_{\text{bg}}} \mu_j \boldsymbol{m}_j \rangle = \Theta(1)$.

Similar to the first part of the proof, we have the SGD update

$$\left\langle \mathbf{w}_k^{(t+1)}, \sum_{j \in \mathcal{S}_{\text{bg}}} \mu_j \boldsymbol{m}_j \right\rangle - \left\langle \mathbf{w}_k^{(t)}, \sum_{j \in \mathcal{S}_{\text{bg}}} \mu_j \boldsymbol{m}_j \right\rangle$$
$$= -\eta \left\langle \nabla_{\mathbf{w}_k^{(t)}} \frac{1}{N} \sum_{i \in [N]} \ell\left(h^{(t)}(\mathbf{x}_i^{(t)}), \mathbf{y}_i^{(t)}\right), \sum_{j \in \mathcal{S}_{\text{bg}}} \mu_j \boldsymbol{m}_j \right\rangle - \lambda \eta \left\langle \mathbf{w}_k^{(t)}, \sum_{j \in \mathcal{S}_{\text{bg}}} \mu_j \boldsymbol{m}_j \right\rangle.$$

Applying Lemma 10, we obtain

$$\left\langle \mathbf{w}_k^{(t+1)}, \sum_{j \in \mathcal{S}_{\text{bg}}} \mu_j \boldsymbol{m}_j \right\rangle - \left\langle \mathbf{w}_k^{(t)}, \sum_{j \in \mathcal{S}_{\text{bg}}} \mu_j \boldsymbol{m}_j \right\rangle$$
$$= \Theta\left(\frac{\eta d_0}{m}\right) - \eta g_{k,-y,\mathcal{S}_{\text{bg}}}^{(t)} - \lambda \eta \left\langle \mathbf{w}_k^{(t)}, \sum_{j \in \mathcal{S}_{\text{bg}}} \mu_j \boldsymbol{m}_j \right\rangle,$$

where

$$g_{k,-y,\mathcal{S}_{\text{bg}}}^{(t)} = \frac{1}{m} \sum_{j \in \mathcal{S}_{\text{bg}}} \mathbb{E}_{(\mathbf{x},\mathbf{y}) \sim \mathcal{D}_{\text{train}}} \mathbb{1}_{h^{(t)}(\mathbf{x}) \leq 1} \mathbb{1}_{\mathbf{y}=-y} \mathbb{1}_{\langle \mathbf{w}_k^{(t)}, \mathbf{x} \rangle \geq 0} \mu_j \mathbf{z}_j$$
$$= \frac{1}{2m} \mathbb{E}_{\mathbf{x}|\mathbf{y}=-y} \left[ \mathbb{1}_{h^{(t)}(\mathbf{x}) \leq 1} \sum_{j \in \mathcal{S}_{\text{bg}}} \mu_j \mathbf{z}_j \middle| \langle \mathbf{w}_k^{(t)}, \mathbf{x} \rangle \geq 0 \right] \mathbf{Pr}_{\mathbf{x}|\mathbf{y}=-y} \left[ \langle \mathbf{w}_k^{(t)}, \mathbf{x} \rangle \geq 0 \right]$$
$$\leq \Theta\left(\frac{d_0}{m}\right) \mathbf{Pr}_{\mathbf{x}|\mathbf{y}=-y} \left[ \langle \mathbf{w}_k^{(t)}, \mathbf{x} \rangle \geq 0 \right].$$

Using Theorem 1, we have after at most $T_0 = \widetilde{\Theta}(\frac{m}{\eta \sqrt{d}})$ iterations, $\mathbf{Pr}_{\mathbf{x}|\mathbf{y}=-y} \left[ \langle \mathbf{w}_k^{(t)}, \mathbf{x} \rangle \geq 0 \right] = o(1)$. We thus have

$$\left\langle \mathbf{w}_k^{(t+1)}, \sum_{j \in \mathcal{S}_{\text{bg}}} \mu_j \boldsymbol{m}_j \right\rangle = (1 - \lambda \eta) \left\langle \mathbf{w}_k^{(t)}, \sum_{j \in \mathcal{S}_{\text{bg}}} \mu_j \boldsymbol{m}_j \right\rangle + \Theta\left(\frac{\eta d_0}{m}\right)$$

for every $t \geq T_0$. By a similar argument as in the first part of the proof, we have $\left\langle \mathbf{w}_k^{(T)}, \sum_{j \in \mathcal{S}_{\text{bg}}} \mu_j \boldsymbol{m}_j \right\rangle \leq \Theta(1)$ and

$$\left\langle \mathbf{w}_k^{(T_1)}, \sum_{j \in \mathcal{S}_{\text{bg}}} \mu_j \boldsymbol{m}_j \right\rangle = (T_1 - T_0) \Theta\left(\frac{\eta d_0}{m}\right) + \left\langle \mathbf{w}_k^{(T_0)}, \sum_{j \in \mathcal{S}_{\text{bg}}} \mu_j \boldsymbol{m}_j \right\rangle$$
$$\geq \Theta(1) - T_0 \cdot \Theta\left(\frac{\eta d_0}{m}\right) - \widetilde{O}\left(\sqrt{\frac{d_0}{d}}\right)$$
$$= \Theta(1).$$

Finally, combining **Part 1** and **Part 2** completes the proof. $\qquad\square$

### B.3 PROOF OF THEOREM 3

**Theorem 3** (ID and OOD risks). *For every $\eta \leq \frac{1}{\text{poly}(d_0)}$, after at most $T_2 = \widetilde{\Theta}(\frac{m}{\eta d_0})$ iterations with high probability, the trained model $h^{(T_2)}$ satisfies the following:*

$$\mathcal{R}_{\mathcal{D}_{\text{train}}}(h^{(T_2)}) \leq o(1), \quad \mathcal{R}_{\text{OOD}}(h^{(T_2)}) = \Theta(1). \tag{46}$$

*Proof.* We break the subsequent proof into two parts: in the first part we prove the desired result for the ID risk; in the second part we prove the desired result for the OOD risk.

**Part 1: proving $\mathcal{R}_{\mathcal{D}_{\text{train}}}(h^{(T_2)}) \leq o(1)$.**

By definition, we have

$$
\begin{aligned}
\mathcal{R}_{\mathcal{D}_{\text{train}}}(h^{(T_2)}) &= \mathbb{E}_{(\mathbf{x},\mathbf{y}) \sim \mathcal{D}_{\text{train}}} \max\left\{1 - \mathbf{y} h^{(T_2)}(\mathbf{x}), 0\right\} \\
&= \frac{1}{2} \underbrace{\mathbb{E}_{\mathbf{x}|\mathbf{y}=1}\left[1 - h^{(T_2)}(\mathbf{x})\Big| h^{(T_2)}(\mathbf{x}) \leq 1\right] \mathbf{Pr}_{\mathbf{x}|\mathbf{y}=1}\left[h^{(T_2)}(\mathbf{x}) \leq 1\right]}_{\mathcal{R}_1} \\
&\quad + \frac{1}{2} \underbrace{\mathbb{E}_{\mathbf{x}|\mathbf{y}=-1}\left[1 + h^{(T_2)}(\mathbf{x})\Big| h^{(T_2)}(\mathbf{x}) \geq -1\right] \mathbf{Pr}_{\mathbf{x}|\mathbf{y}=-1}\left[h^{(T_2)}(\mathbf{x}) \geq -1\right]}_{\mathcal{R}_{-1}}.
\end{aligned}
\tag{47}
$$

We first consider $\mathcal{R}_1$. By Theorem 2, we have that after $T_1 = \Theta(\frac{m}{\eta d_0})$ iterations, for every neuron $k \in \mathcal{N}_1^{(t)}$ with $t \geq T_1$, we have

$$
\sum_{j \in \mathcal{S}_{\text{core}}} \mu_j \mathbf{z}_j \langle \mathbf{w}_k^{(t)}, \boldsymbol{m}_j \rangle = \Theta(1), \qquad \sum_{j \in \mathcal{S}_{\text{bg}}} \mu_j \mathbf{z}_j \langle \mathbf{w}_k^{(t)}, \boldsymbol{m}_j \rangle = \Theta(1).
$$

We can then obtain

$$
\mathbb{E}_{\mathbf{x}|\mathbf{y}=1} \langle \mathbf{w}_k^{(t)}, \mathbf{x} \rangle = \sum_{j \in [d_0]} \mu_j \mathbf{z}_j \langle \mathbf{w}_k^{(t)}, \mathbf{x} \rangle = \Theta(1).
$$

On the other hand, by Lemma 13, we know that for every neuron $k$ satisfying $\text{sign}(a_k) = y$, its correlation grow rate is asymptotically not larger than the correlation grow rate of neurons in $\mathcal{N}_y^{(t)}$. Denoting the set of those neurons as $\mathcal{M}_y := \{k \in [m] : \text{sign}(a_k) = y\}, \forall y \in \mathcal{Y}$, we then have

$$
\mathbb{E}_{\mathbf{x}|\mathbf{y}=1} \langle \mathbf{w}_k^{(t)}, \mathbf{x} \rangle = O(1), \forall k \in \mathcal{M}_1, t \geq T_1.
$$

Meanwhile, for all neurons $k \in \mathcal{M}_{-1}$, by Lemma 15 and Theorem 1 we have for all $t \geq T_0 = \widetilde{\Theta}(\frac{m}{\eta\sqrt{d}})$,

$$
\mathbf{Pr}_{\mathbf{x}|\mathbf{y}=1}[\langle \mathbf{w}_k^{(t)}, \mathbf{x} \rangle \geq 0] = o(1).
$$

Therefore, we have

$$
\begin{aligned}
\mathbb{E}_{\mathbf{x}|\mathbf{y}=1} h^{(T_1)}(\mathbf{x}) &= \frac{1}{m} \sum_{k \in \mathcal{M}_1} \mathbb{E}_{\mathbf{x}|\mathbf{y}=1}\left[\text{ReLU}\left(\langle \mathbf{w}_k^{(T_1)}, \mathbf{x} \rangle\right)\right] - \frac{1}{m} \sum_{k \in \mathcal{M}_{-1}} \mathbb{E}_{\mathbf{x}|\mathbf{y}=1}\left[\text{ReLU}\left(\langle \mathbf{w}_k^{(T_1)}, \mathbf{x} \rangle\right)\right] \\
&= \frac{1}{m} \sum_{k \in \mathcal{M}_1} \Theta(1) - \frac{1}{m} \sum_{k \in \mathcal{M}_{-1}} o(1) \\
&= \Theta(1).
\end{aligned}
$$

Now, suppose that $\mathcal{R}_1 \geq \Theta(1)$. Choose $T_2 = \Theta(\frac{m\sqrt{\log m}}{\eta d_0}) = \widetilde{\Theta}(\frac{m}{\eta d_0})$. Then, for every $t = T_1, \ldots, T_2 - 1$ we have

$$
\mathbf{Pr}_{\mathbf{x}|\mathbf{y}=1}\left[h^{(T_2)}(\mathbf{x}) \leq 1\right] = \Theta(1).
$$

This further leads to

$$
\begin{aligned}
&\frac{1}{m} \sum_{k \in \mathcal{M}_1} \mathbb{E}_{\mathbf{x}|\mathbf{y}=1}\left[\text{ReLU}\left(\langle \mathbf{w}_k^{(t+1)}, \mathbf{x} \rangle\right)\right] - \frac{1}{m} \sum_{k \in \mathcal{M}_1} \mathbb{E}_{\mathbf{x}|\mathbf{y}=1}\left[\text{ReLU}\left(\langle \mathbf{w}_k^{(t)}, \mathbf{x} \rangle\right)\right] \\
&= \frac{1}{m} \sum_{k \in \mathcal{M}_1} \mathbb{E}_{\mathbf{x}|\mathbf{y}=1}\left[\langle \mathbf{w}_k^{(t+1)}, \mathbf{x} \rangle - \langle \mathbf{w}_k^{(t)}, \mathbf{x} \rangle\right] \\
&= \frac{1}{m} \sum_{k \in \mathcal{N}_1^{(t)}} \mathbb{E}_{\mathbf{x}|\mathbf{y}=1}\left[\langle \mathbf{w}_k^{(t+1)}, \mathbf{x} \rangle - \langle \mathbf{w}_k^{(t)}, \mathbf{x} \rangle\right] + \frac{1}{m} \sum_{k \in \mathcal{M}_1 \setminus \mathcal{N}_1^{(t)}} \mathbb{E}_{\mathbf{x}|\mathbf{y}=1}\left[\langle \mathbf{w}_k^{(t+1)}, \mathbf{x} \rangle - \langle \mathbf{w}_k^{(t)}, \mathbf{x} \rangle\right]
\end{aligned}
\tag{48}
$$

For the first term in the RHS of the last equality, by Lemma 12 we have

$$\frac{1}{m} \sum_{k \in \mathcal{N}_1^{(t)}} \mathbb{E}_{\mathbf{x}|\mathbf{y}=1} \left[ \left\langle \mathbf{w}_k^{(t+1)}, \mathbf{x} \right\rangle - \left\langle \mathbf{w}_k^{(t)}, \mathbf{x} \right\rangle \right]$$

$$= \frac{1}{m} \sum_{k \in \mathcal{N}_1^{(t)}} \left( \Theta \left( \frac{\eta d_0}{m} \right) - \lambda \eta \mathbb{E}_{\mathbf{x}|\mathbf{y}=1} \left\langle \mathbf{w}_k^{(t)}, \mathbf{x} \right\rangle \right)$$

$$= \Theta \left( \frac{\eta d_0}{m} \right),$$

where in the last equality we use $|\mathcal{N}_1^{(t)}| = \Theta(m)$, $\lambda = O(\frac{d_0}{m^{0.01}})$ and $\mathbb{E}_{\mathbf{x}|\mathbf{y}=1} \left\langle \mathbf{w}_k^{(t)}, \mathbf{x} \right\rangle = \widetilde{O}(1)$ for $t \leq T_2$.

For the second term in the RHS of the last equality in (48), by Lemma 13 we have

$$\frac{1}{m} \sum_{k \in \mathcal{M}_1 \setminus \mathcal{N}_1^{(t)}} \mathbb{E}_{\mathbf{x}|\mathbf{y}=1} \left[ \left\langle \mathbf{w}_k^{(t+1)}, \mathbf{x} \right\rangle - \left\langle \mathbf{w}_k^{(t)}, \mathbf{x} \right\rangle \right] \leq O \left( \frac{\eta d_0}{m} \right).$$

Therefore,

$$\mathbb{E}_{\mathbf{x}|\mathbf{y}=1} h^{(T_2)}(\mathbf{x}) = \mathbb{E}_{\mathbf{x}|\mathbf{y}=1} h^{(T_1)}(\mathbf{x}) + \Theta \left( \frac{\eta d_0 (T_2 - T_1)}{m} \right) \pm o(1)$$

$$= \Theta(1) + \Theta(\sqrt{\log m}) \pm o(1) = \Theta(\sqrt{\log m}).$$

Applying one-sided Bernstein's inequality (Lemma 19) then gives

$$\mathbf{Pr}_{\mathbf{x}|\mathbf{y}=1} \left[ h^{(T_2)}(\mathbf{x}) \leq 1 \right] = O \left( \frac{1}{\sqrt{m}} \right),$$

which contradicts with $\mathbf{Pr}_{\mathbf{x}|\mathbf{y}=1} \left[ h^{(T_2)}(\mathbf{x}) \leq 1 \right] = \Theta(1)$. Hence, we must have $\mathcal{R}_1 = o(1)$. By a similar argument, we also have $\mathcal{R}_{-1} = o(1)$. We then have that $\mathcal{R}_{\mathcal{D}_{\text{train}}}(h^{(T_2)}) = o(1)$ holds.

**Part 2: proving $\mathcal{R}_{\text{OOD}}(h^{(T)}) = \Theta(1)$.**

This part of the proof directly follows from Theorem 2. Since after $t = T_1$ iterations we have $\sum_{j \in \mathcal{S}_{\text{bg}}} \mu_j \langle \mathbf{w}_k^{(t)}, \boldsymbol{m}_j \rangle = \Theta(1)$ for every neuron $k \in \mathcal{N}_y^{(t)}$, it can be shown that perturbing each $\mu_j$ from $\Theta(1)$ to $-\Theta(1)$ changes the output of the network by at least $-\frac{1}{m} \sum_{k \in \mathcal{N}_y^{(t)}} \Theta(1) = -\Theta(1)$ using the fact that $|\mathcal{N}_y^{(t)}| = \Theta(m)$ for every $t$ (using Lemma 5 and Lemma 14). By the definition of the OOD risk we then arrive at the desired result.

Finally, combining **Part 1** and **Part 2** completes the proof. $\qquad\square$

## C   SEPARATION BETWEEN LINEARITY AND NON-LINEARITY: PROOF OF THEOREM 4

### C.1   TWO-LAYER LINEAR NETWORKS: PROOF OF THEOREM 4

Before providing the main proof, we first introduce some lemmas that characterize the gradients of the two-layer linear network. In general, the gradients of two-layer linear networks take a similar form to those of two-layer ReLU networks except for not having the ReLU derivative. We can thus reuse some of our lemmas in Section A and Section B in the analysis of the gradients.

**Notation.** In this section, we overload the notation from the previous sections such as $h^{(t)}(\mathbf{x})$, $\mathbf{w}_k^{(t)}$ and let them also denote the linear network model/weights.

**Lemma 16** (Gradient of linear networks). *For every example $(x, y) \in \mathcal{X} \times \mathcal{Y}$, every $k \in [m]$, and every iteration $t$, the following holds:*

$$\nabla_{\mathbf{w}_k^{(t)}} \ell \left( h(x), y \right) = -a_k y \mathbb{1}_{h(x) \leq 1} x. \tag{49}$$

*Proof.* The proof follows from simple calculation. □

**Lemma 17** (Gap between empirical and population gradients). *For every $k \in [m]$, every $j \in [d]$, and every iteration $t$, if the batch size $N = \mathrm{poly}(d)$ for some sufficiently large polynomial, then the following holds with probability at least $1 - e^{-\Omega(d)}$:*

$$\left| \left\langle \nabla_{\mathbf{w}_k^{(t)}} \frac{1}{N} \sum_{i \in [N]} \ell\left(h^{(t)}(\mathbf{x}_i^{(t)}), \mathbf{y}_i^{(t)}\right), \boldsymbol{m}_j \right\rangle - \left\langle \nabla_{\mathbf{w}_k^{(t)}} \mathbb{E}_{(\mathbf{x},\mathbf{y}) \sim \mathcal{D}_{\mathrm{train}}} \ell\left(h(\mathbf{x}), \mathbf{y}\right), \boldsymbol{m}_j \right\rangle \right| \leq \frac{1}{\mathrm{poly}(d)}.$$
(50)

*Proof.* The proof is nearly identical to Lemma 2, hence we omit here. □

Since in linear models we do not need to consider the activation probability (equivalently, this can be viewed as each neuron being fully activated for every example), we can analyze the gradient projections for all neurons without resorting to characterizing a subset of neurons as in Definition 2.

**Lemma 18** (Gradient projection onto background features, linear networks). *For every iteration $t \leq O(\frac{m}{\eta d_0})$, every $k \in [m]$, and every $j \in \mathcal{S}_{\mathrm{bg}}$, the following holds:*

$$\left| \left\langle \nabla_{\mathbf{w}_k^{(t)}} \frac{1}{N} \sum_{i \in [N]} \ell\left(h^{(t)}(\mathbf{x}_i^{(t)}), \mathbf{y}_i^{(t)}\right), \boldsymbol{m}_j \right\rangle \right| = \frac{1}{\mathrm{poly}(d)},$$
(51)

*Proof.* Applying Lemma 16 and Lemma 17, we obtain

$$-\left\langle \nabla_{\mathbf{w}_k^{(t)}} \frac{1}{N} \sum_{i \in [N]} \ell\left(h^{(t)}(\mathbf{x}_i^{(t)}), \mathbf{y}_i^{(t)}\right), \boldsymbol{m}_j \right\rangle$$

$$= -\left\langle \nabla_{\mathbf{w}_k^{(t)}} \mathbb{E}_{(\mathbf{x},\mathbf{y}) \sim \mathcal{D}_{\mathrm{train}}} \ell\left(h^{(t)}(\mathbf{x}), \mathbf{y}\right), \boldsymbol{m}_j \right\rangle \pm \frac{1}{\mathrm{poly}(d)}$$

$$= a_k \mathbb{E}_{(\mathbf{x},\mathbf{y}) \sim \mathcal{D}_{\mathrm{train}}} \mathbf{y} \mathbb{1}_{h^{(t)}(\mathbf{x}) \leq 1} \mathbf{z}_j \pm \frac{1}{\mathrm{poly}(d)}$$

$$= a_k \mathbb{E}_{(\mathbf{x},\mathbf{y}) \sim \mathcal{D}_{\mathrm{train}}} \mathbb{1}_{h^{(t)}(\mathbf{x}) \leq 1} \mathbb{1}_{\mathbf{y}=1} \mathbf{z}_j$$

$$\qquad - a_k \mathbb{E}_{(\mathbf{x},\mathbf{y}) \sim \mathcal{D}_{\mathrm{train}}} \mathbb{1}_{h^{(t)}(\mathbf{x}) \leq 1} \mathbb{1}_{\mathbf{y}=-1} \mathbf{z}_j \pm \frac{1}{\mathrm{poly}(d)}$$

$$= \pm \frac{1}{\mathrm{poly}(d)}.$$
(52)

This gives the desired result. □

We are now ready to prove Theorem 4.

**Theorem 4** (Linear networks). *If we replace the ReLU functions in the network with identity functions and keep other conditions the same as in Theorem 2, then with high probability, we have $|\langle \mathbf{w}_k^{(T_1)}, \boldsymbol{m}_j \rangle| \leq \widetilde{O}(\frac{1}{\sqrt{d}})$ for every $k \in [m]$ and every $j \in \mathcal{S}_{\mathrm{bg}}$.*

*Proof.* For every $k \in [m]$ and every $j \in \mathcal{S}_{\mathrm{bg}}$, by the SGD update (4) we have

$$\langle \mathbf{w}_k^{(t+1)}, \boldsymbol{m}_j \rangle = -\eta \left\langle \nabla_{\mathbf{w}_k^{(t)}} \frac{1}{N} \sum_{i \in [N]} \ell\left(h^{(t)}(\mathbf{x}_i^{(t)}), \mathbf{y}_i^{(t)}\right), \boldsymbol{m}_j \right\rangle + (1 - \lambda\eta)\left\langle \mathbf{w}_k^{(t)}, \sum_{j \in \mathcal{S}_{\mathrm{core}}} \mu_j \boldsymbol{m}_j \right\rangle.$$

By Lemma 18, we obtain

$$\langle \mathbf{w}_k^{(t+1)}, \boldsymbol{m}_j \rangle = (1 - \lambda\eta)\langle \mathbf{w}_k^{(t)}, \boldsymbol{m}_j \rangle \pm \frac{\eta}{\mathrm{poly}(d)}.$$

By Lemma 23, with probability at least $1 - O(\frac{1}{m})$, we have at initialization

$$\max_{k \in [m]} |\langle \mathbf{w}_k^{(0)}, \boldsymbol{m}_j \rangle| \leq 2\sqrt{\frac{\log m}{d}}.$$

Recall that $\lambda = O(\frac{d_0}{m^{1.01}})$. Combining the above equations gives the desired result. □

**Remark.** Similar to our analysis of two-layer ReLU networks, for two-layer linear networks we can also analyze the correlation growth between every neuron and the core features and show that SGD can converge to a solution with small ID risk. Since Theorem 4 indicates that linear networks do not have feature accompaniment (i.e., background features do not accumulate in the weights), we can show that the network would also have small OOD risk at convergence. Since this analysis has a similar procedure to (and is also much simpler than) our analysis on two-layer ReLU networks we do not include it here.

## D  PROBABILITY THEORY LEMMAS

In this section, we provide the probability theory lemmas used in our proofs for completeness. Since these lemmas are all standard results in the probability theory, we omit the proofs of them.

We first state an one-sided form of Bernstein's inequality.

**Lemma 19** (One-sided Bernstein's inequality). *Given $n$ independent random variables $\{\mathbf{x}_i\}_{i \in [n]}$ with $\mathbf{x}_i \leq b$ almost surely for every $i \in [n]$, the following holds for every $\delta \geq 0$:*

$$\mathbf{Pr}\Big[ \sum_{i \in [n]} (\mathbf{x}_i - \mathbb{E}\mathbf{x}_i) \geq n\delta \Big] \leq \exp\left( -\frac{n\delta^2}{\frac{1}{n}\sum_{i \in [n]} \mathbb{E}\mathbf{x}_i^2 + \frac{b\delta}{3}} \right). \tag{53}$$

Note that the above result can also be used to derive bounds on the lower tail by applying it to the random variables $\{-\mathbf{x}_i\}_{i \in [n]}$ if each $\mathbf{x}_i$ is bounded from below.

We then state a matrix extension of Bernstein's inequality; such type of inequalities is useful for bounding the gradients of the network in our proofs.

**Lemma 20** (Matrix Bernstein's inequality (Oliveira, 2010; Tropp, 2012)). *Given $n$ independent random matrices $\{\mathbf{X}_i\}_{i \in [n]}$ with dimension $d_1 \times d_2$ and $\mathbb{E}\mathbf{X}_i = \mathbf{0}$, $\|\mathbf{X}_i\|_2 \leq b$ almost surely for every $i \in [n]$, define the sum $\mathbf{S} := \sum_{i \in [n]} \mathbf{X}_i$ and let $v(\mathbf{S})$ denote the matrix variance statistic of the sum:*

$$v(\mathbf{S}) := \max \{\|\mathbb{E}[\mathbf{S}\mathbf{S}^*]\|_2, \|\mathbb{E}[\mathbf{S}^*\mathbf{S}]\|_2\}, \tag{54}$$

*where $\|\cdot\|_2$ denotes the spectral norm a matrix or the $\ell_2$ norm of a vector (when $d_1 = 1$ or $d_2 = 1$). Then, the following holds for every $\delta \geq 0$:*

$$\mathbf{Pr}\big[\|\mathbf{S}\|_2 \geq \delta\big] \leq (d_1 + d_2) \cdot \exp\left( -\frac{\delta^2}{2v(\mathbf{S}) + \frac{2b\delta}{3}} \right). \tag{55}$$

We then state a basic result for bounding the cumulative distribution function of the unit Gaussian distribution that is repeatedly used in deriving neuron properties in initialization.

**Lemma 21** (Bounds for unit Gaussian variables). *Let $\mathbf{x} \sim \mathcal{N}(0, 1)$ be a unit Gaussian random variable. Then, the following holds for every $\delta > 0$:*

$$\frac{1}{\sqrt{2\pi}}\frac{\delta}{\delta^2 + 1} \exp\left( -\frac{\delta^2}{2} \right) \leq \mathbf{Pr}[\mathbf{x} \geq \delta] \leq \frac{1}{\sqrt{2\pi}}\frac{1}{\delta} \exp\left( -\frac{\delta^2}{2} \right). \tag{56}$$

Finally, we state a result for lower bounding the upper tail of the cumulative distribution function for binomial variables using Hoeffding's inequality:

**Lemma 22** (Tail bound for binomial variables). *Let $\mathbf{x} \sim \mathcal{B}(n, p)$ be a binomial random variable with trial number $n$ and success probability $p \in [0, 1]$. Then, the following holds for every $n, p$ and integer $k \leq np$:*

$$\mathbf{Pr}[\mathbf{x} \geq k] \geq 1 - \exp\left( -2n\left( p - \frac{k-1}{n} \right)^2 \right). \tag{57}$$

**Lemma 23** (Concentration of the maximum of absolute Gaussian). *Let $\mathbf{x}_1, \ldots, \mathbf{x}_n$ be i.i.d. random variables that follow the zero-mean Gaussian distribution $\mathcal{N}(0, \sigma^2)$. Then, the following holds for every positive integer $n$:*

$$\mathbf{Pr}\Big[\max_{i \in [n]} |\mathbf{x}_i| \geq 2\sigma\sqrt{\log n}\Big] \leq \frac{2}{n}. \tag{58}$$

**Lemma 24** (Berry–Esseen theorem)**.** *Let* $\mathbf{x}_1, \ldots, \mathbf{x}_n$ *be independent random variables with* $\mathbb{E}\mathbf{x}_i = 0$, $\mathbb{E}\mathbf{x}_i^2 = \sigma_i^2 > 0$, *and* $\rho_i := \mathbb{E}|\mathbf{x}_i^3| < \infty$. *Also, define the normalized sum*

$$\mathbf{s}_n := \frac{\sum_{i \in [n]} \mathbf{x}_i}{\sqrt{\sum_{i \in [n]} \sigma_i^2}}. \tag{59}$$

*Denote* $\Phi$ *the cumulative distribution function of* $\mathcal{N}(0, 1)$. *Then, there exists a constant* $C_0 \in [0.40, 0.56]$ *such that*

$$\sup_{\delta \in \mathbb{R}} |\mathbf{Pr}[\mathbf{s}_n < \delta] - \Phi(\delta)| \leq C_0 \left( \sum_{i=1}^{n} \sigma_i^2 \right)^{-\frac{3}{2}} \sum_{i=1}^{n} \rho_i. \tag{60}$$

# APPENDIX II: EXPERIMENTAL DETAILS AND ADDITIONAL EMPIRICAL RESULTS

In this part of the appendix, we provide the details of the experiments in the main text and include additional empirical results in both real-world datasets and synthetic distribution shift settings. A quick overview of the structure of this part is as follows:

- In Section E, we provide the implementation details and more results of the representation distillation experiments in Section 2 in the main text.

- In Section F, we present numerical experiments in both classification and regression (representation distillation) settings as well as additional feature visualizations on a variant of the CIFAR-10 dataset.

- In Section G, we provide empirical evidence that supports Conjecture 1 in the main text and some further discussion.

## E    REPRESENTATION DISTILLATION DETAILS

### E.1    NATURAL DISTRIBUTION SHIFTS OF IMAGENET

**Datasets.**    Following (Taori et al., 2020; Radford et al., 2021; Wortsman et al., 2022), we consider 5 natural distribution shift test sets of ImageNet that are representative of real-world distribution shifts without artificial perturbations to images, including ImageNetV2 (Recht et al., 2019), ImageNet-R (Hendrycks et al., 2021a), ObjectNet (Barbu et al., 2019), ImageNet Sketch (Wang et al., 2019), and ImageNet-A (Hendrycks et al., 2021b). Compared to the original training and validation (ID test) sets of ImageNet, those test sets are reflective of changes in data distribution due to natural variations in the data collection process such as lighting, geographic location, image background, and styles.

**Pre-trained models.**    We used pre-trained checkpoints provided by CLIP (Radford et al., 2021), which is reported to exhibit remarkable robustness to distribution shifts of ImageNet. The official CLIP repository provide CLIP models pre-trained on the same dataset with varying sizes and architectures (ResNets and ViTs). In our experiments, we used five different CLIP models, including four ResNets and one ViT: CLIP-ResNet-50 (CLIP-RN50), CLIP-ResNet-101 (CLIP-RN101), CLIP-ResNet-50x4 (CLIP-RN50x4), CLIP-ResNet-50x16 (CLIP-RN50x16), and CLIP-ViT-B/16. For linear probing, we freezed the weights of the pre-trained models and trained randomly-initialized linear classification heads on top of the extracted representations on the ImageNet training set for 10 epochs. Following the hyperparameters used by Wortsman et al. (2022), we used the AdamW optimizer (Loshchilov & Hutter, 2019) with learning rate 0.001, $\ell_2$ weight decay 0.1, batch size 256, and a cosine learning rate scheduler (Loshchilov & Hutter, 2017). The results are reported based on the model with the best ID validation accuracy.

**Representation distillation.**    For each pre-trained CLIP model (teacher model), we freezed its weights and randomly initialized another model with identical architecture to the teacher model. We used the Mean Squared Error (MSE) loss to train the student model on the ImageNet training set, minimizing the mean Euclidean distance between the representations extracted by the student model and the representations extracted by the teacher model. We did not perform extensive grid search on the distillation hyperparameters and sticked to the following hyperparameter choices based on our preliminary experiments:

- CLIP-RN50: AdamW optimizer with learning rate 0.001, $\ell_2$ weight decay 0.05, batch size 256, and a cosine learning rate schedular with warmup for 10000 steps; 100 distillation epochs.

- CLIP-RN101: AdamW optimizer with learning rate 0.001, $\ell_2$ weight decay 0.1, batch size 256, and a cosine learning rate scheduler with warmup for 10000 steps; 100 distillation epochs.

- CLIP-RN50x4 and CLIP-RN50x16: AdamW optimizer with learning rate 0.0001, $\ell_2$ weight decay 0.5, batch size 256, and a cosine learning rate scheduler with warmup for 10000 steps; 100 distillation epochs.

- CLIP-ViT-B/16: AdamW optimizer with learning rate 0.0001, $\ell_2$ weight decay 0.1, batch size 256, and a cosine learning rate scheduler with warmup for 10000 steps; 200 distillation epochs. Besides minimizing the difference between final representations (i.e., the output of the last layer of the networks) of student and teacher networks, we also minimized the difference between student and teacher network's intermediate representations of each residual attention block with a weighting coefficient 0.1.

In the linear probing stage, we freezed the parameters of the student models and trained a randomly initialized linear classification head for each student model on the ImageNet training set for 10 epochs. We used the AdamW optimizer with learning rate 0.001, $\ell_2$ weight decay of 0.001, batch size 256, and a cosine learning rate scheduler. The results are reported based on the model with the best ID validation accuracy.

**Baseline models.** We reported the results of baseline models provided by the testbed of Taori et al. (2020). In their testbed, Taori et al. (2020) catogory the models into different types, where some type of models are trained with more data than the original ImageNet training set. Since our aim is to explore the limit of representation learning using only ID data, we omit the results of those models trained with more data. We also omit the results of models with significantly lower accuracy than common ImageNet models, such as linear classifier on pixels or random features, classifiers based on nearest neighbors, and low accuracy CNNs. Concretely, we reported the results of the following two types of models defined by Taori et al. (2020):

- `STANDARD`: models obtained by standard training (i.e., ERM) on the ImageNet training set.
- `ROBUST_INTV`: models trained with existing robust intervention techniques on the ImageNet training set.

**Detailed results.** We list detailed OOD generalization performance of linear probes on pre-trained and distilled representations on all 5 distribution shift test sets as well as the ID generalization results on the original ImageNet validation set in Table 1.

Table 1: Detailed ID and OOD top-1 accuracy (%) of linear probes on pre-trained and distilled representations on ImageNet-based test sets. "IN" refers to "ImageNet".

|                    | IN (ID) | OOD Avg. | INV2  | IN-R  | ObjectNet | IN Sketch | IN-A  |
|--------------------|---------|----------|-------|-------|-----------|-----------|-------|
| CLIP-RN50          | 70.37   | 39.42    | 59.03 | 51.18 | 37.72     | 31.87     | 17.31 |
| Distilled RN50     | 69.85   | 31.63    | 57.97 | 38.22 | 32.72     | 20.97     | 8.25  |
| CLIP-RN101         | 72.33   | 45.27    | 61.70 | 59.92 | 43.07     | 37.93     | 23.73 |
| Distilled RN101    | 72.28   | 35.18    | 60.46 | 44.09 | 35.89     | 23.88     | 11.56 |
| CLIP-ViT-B/16      | 79.40   | 57.59    | 69.72 | 72.42 | 51.85     | 47.33     | 46.64 |
| Distilled ViT-B/16 | 73.58   | 37.14    | 62.45 | 44.43 | 35.52     | 23.83     | 19.47 |
| CLIP-RN50x4        | 76.18   | 51.45    | 65.83 | 64.80 | 48.74     | 42.19     | 35.67 |
| Distilled RN50x4   | 76.25   | 41.40    | 65.20 | 49.22 | 42.71     | 29.23     | 20.64 |
| CLIP-RN50x16       | 80.24   | 60.61    | 70.13 | 73.67 | 56.92     | 48.52     | 53.79 |
| Distilled RN50x16  | 79.65   | 48.26    | 68.49 | 55.03 | 48.90     | 32.93     | 35.97 |

## E.2 iWildCam-WILDS

**Dataset.** We used the official version of the dataset provided by WILDS (Koh et al., 2021).

**Pre-trained models.** In order to obtain a feature extractor that exhibits sufficient generalization ability on the dataset, we explored different pre-trained models including ViTs in CLIP (Radford et al., 2021), RegNets in SWAG (Singh et al., 2022) as well as ResNets pre-trained on ImageNet (Deng et al., 2009). In the end, we chose a fine-tuned ResNet-50 (RN50) that is pre-trained on ImageNet as the teacher model since we observed that ImageNet-scale pre-training already leads to considerable robustness improvements compared to models trained from scratch on this dataset

(also reported by Miller et al. (2021)), while being consistent to the network architecture used in the official WILDS repository. For linear probing, we freezed the parameters of the pre-trained model and trained a randomly initialized linear classification head using the hyperparameters provided by the official WILDS repository. The results are reported based on the model with the best OOD validation accuracy, following the protocol used by the WILDS paper (Koh et al., 2021).

**Representation distillation.** We freezed the weights of the teacher model and randomly initialized a ResNet-50 as the student model. We trained the student model by minimizing the Euclidean distance between its extracted representations and the representations extracted by the teacher model using the MSE loss on the training domains of iWildCam-WILDS. The student model was trained for 150 epochs using AdamW with batch size 128, learning rate 0.0001, and $\ell_2$ weight decay 0.1. In the linear probing stage, we freezed the parameters of the student model and trained a randomly initialized linear classification head using the hyperparameters provided by the official WILDS repository. The results are reported based on the model with the best OOD validation accuracy, following the protocol used by the WILDS paper.

**Baseline models.** We reported the results of baseline models provided by (Miller et al., 2021). In their result file, Miller et al. (2021) report both results for ImageNet-pre-trained neural networks (corresponding to models with `model_type` as "Neural Network" in the result file) and results for neural networks trained from scratch (corresponding to models with `model_type` as "ImageNet Pretrained Network"). Since our aim is to explore the limit of representation learning using only ID data, we omit the results of the models with pre-training.

**Detailed results.** We list detailed ID and OOD generalization performance of linear probes on pre-trained and distilled representations on iWildCam-WILDS in Table 2.

Table 2: Detailed ID and OOD Macro F1 of linear probes on pre-trained and distilled representations on iWildCam-WILDS.

|  | ID Macro F1 | OOD Macro F1 |
|---|---|---|
| ImageNet RN50 | 49.30 | 32.46 |
| Distilled RN50 | 32.32 | 13.83 |

### E.3 CAMELYON17-WILDS

**Dataset.** We used the official version of the dataset provided by WILDS (Koh et al., 2021).

**Pre-trained models.** After preliminary experiments, we chose a ViT-B/16 pre-trained by CLIP as our teacher model. For linear probing, we freezed the parameters of the pre-trained model and trained a randomly initialized linear classification head using the hyperparameters provided by the official WILDS repository. The results are reported based on the model with the best OOD validation accuracy, following the protocol used by the WILDS paper (Koh et al., 2021).

**Representation distillation.** We freezed the weights of the teacher model and randomly initialized a ViT-B/16 with identical architecture to the teacher model as the student model. We trained the student model by minimizing the Euclidean distance between its extracted representations and the representations extracted by the teacher model using the MSE loss on the training domains of Camelyon17-WILDS. The student model was trained for 120 epochs with batch size 128, learning rate 0.0001 and $\ell_2$ weight decay 0.1 using AdamW. For linear probing, we freezed the parameters of the student model and trained a randomly initialized linear classification head using the hyperparameters provided by the official WILDS repository. The results are reported based on the model with the best OOD validation accuracy, following the protocol used by the WILDS paper.

**Baseline models.** We reported the results of all algorithms from the offcial WILDS leaderboard (accessed at September 26th, 2023) that do not use custom data augmentation or pre-training (including "SGD (Freeze-Embed)" that uses CLIP pre-training and "ContriMix", "MBDG", "ERM

w/ targeted aug" and "ERM w/ H&E jitter" that use custom, task-specific data augmentations) as baseline results.

**Detailed results.**  We list detailed ID and OOD generalization performance of linear probes on pre-trained and distilled representations on Camelyon17-WILDS in Table 3.

Table 3: Detailed ID validation and OOD test accuracy (%) of linear probes on pre-trained and distilled representations on Camelyon17-WILDS.

|  | ID Validation Accuracy | OOD Test Accuracy |
|---|---|---|
| CLIP-ViT-B/16 | 98.22 | 92.88 |
| Distilled ViT-B/16 | 98.28 | 89.83 |

### E.4   DOMAINNET

**Dataset.**  Following the setup of Tan et al. (2020); Kumar et al. (2022), we used a pruned version of the original DomainNet dataset (Peng et al., 2019). The pruned dataset consists of 4 domains {Clipart, Painting, Real, Sketch} and 40 commonly occurring classes, selected from the original DomainNet which consists of 6 domains and 345 classes.

**Implementation details.**  We adhered to the experimental settings as in DomainBed (Gulrajani & Lopez-Paz, 2021), which encompassed protocols for data augmentation, dataset partitioning, and hyperparameter search strategies. We opted for the widely adopted training domain validation for the model selection criterion. To reduce the computational cost, without loss of generality, we chose the Sketch domain with the largest distributional shifts as the test domain (OOD), while training on the other three domains (ID). For both our model and baseline models, we performed random searches on the hyperparameters with three different random seeds, each involving 5 trials.

**Pre-trained models.**  We used the official ResNet-50 (RN50), ResNet-101 (RN101), and ViT-B/32 pre-trained checkpoints provided by CLIP.

**Representation distillation.**  Due to limitations imposed by the scale of the dataset, we exclusively employed the CLIP-RN50 as the teacher model—it turns out in our preliminary experiments that distilling the other two pre-trained models results in *worse* performance both ID and OOD, which we believe is because the scale of the dataset is too small for distilling larger models. In the distillation stage, we froze the pre-trained CLIP-RN50 as the teacher model and used the MSE loss to train the student RN50 model with the exact same structure as CLIP-RN50 but with random initialization. We used the AdamW optimizer with a cosine scheduler and learning rate 0.0003, $\ell_2$ weight decay 5e-5, batch size 32, and trained the student model for 95000 iterations. In the linear probe stage, we froze the parameters of the student model and add a randomly initialized single-layer linear classifier. We trained the linear probe on the training sets of the three training domains and performed zero-shot evaluation on the test domain. We ultimately select the checkpoints with the highest accuracy on the validation set from the training domain. During this stage, we used the Adam optimizer (Kingma & Ba, 2015) with a cosine scheduler and learning rate 0.003, $\ell_2$ weight decay 1e-6, batch size 32, and trained the linear probe for 5000 iterations.

**Baseline models.**  We generally followed the settings of DomainBed, with the exception of using a modified RN50 model with the same structure as CLIP-RN50 but randomly initialized. Additionally, we introduced a cosine scheduler with a warmup to enhance the convergence of models trained from scratch. We conducted extensive experiments with 15 representative domain generalization algorithms, including ERM (Vapnik, 1999), IRM (Arjovsky et al., 2019), GroupDRO (Sagawa et al., 2020a), Mixup (Zhang et al., 2018), MLDG (Li et al., 2018), Deep CORAL (Sun & Saenko, 2016), DANN (Ganin et al., 2016), SagNet (Nam et al., 2021), ARM (Zhang et al., 2021), VREx (Krueger et al., 2021), RSC (Huang et al., 2020), SelfReg (Kim et al., 2021), IB-ERM (Ahuja et al., 2021a), and IB-IRM (Ahuja et al., 2021a), and Fish (Shi et al., 2022). We increased the training iterations from the default 5000 to 20000 to ensure the convergence of all methods.

**Detailed results.** We list detailed ID and OOD generalization performance of linear probes on pre-trained and distilled representations on DomainNet in Table 4.

Table 4: Detailed ID test and OOD test accuracy (%) of linear probes on pre-trained and distilled representations on DomainNet.

|  | ID Test Accuracy | OOD Test Accuracy |
|---|---|---|
| CLIP-RN101 | 92.30 | 87.34 |
| CLIP-ViT-B/32 | 92.35 | 87.60 |
| CLIP-RN50 | 87.02 | 82.58 |
| Distilled RN50 | 77.91 | 64.78 |

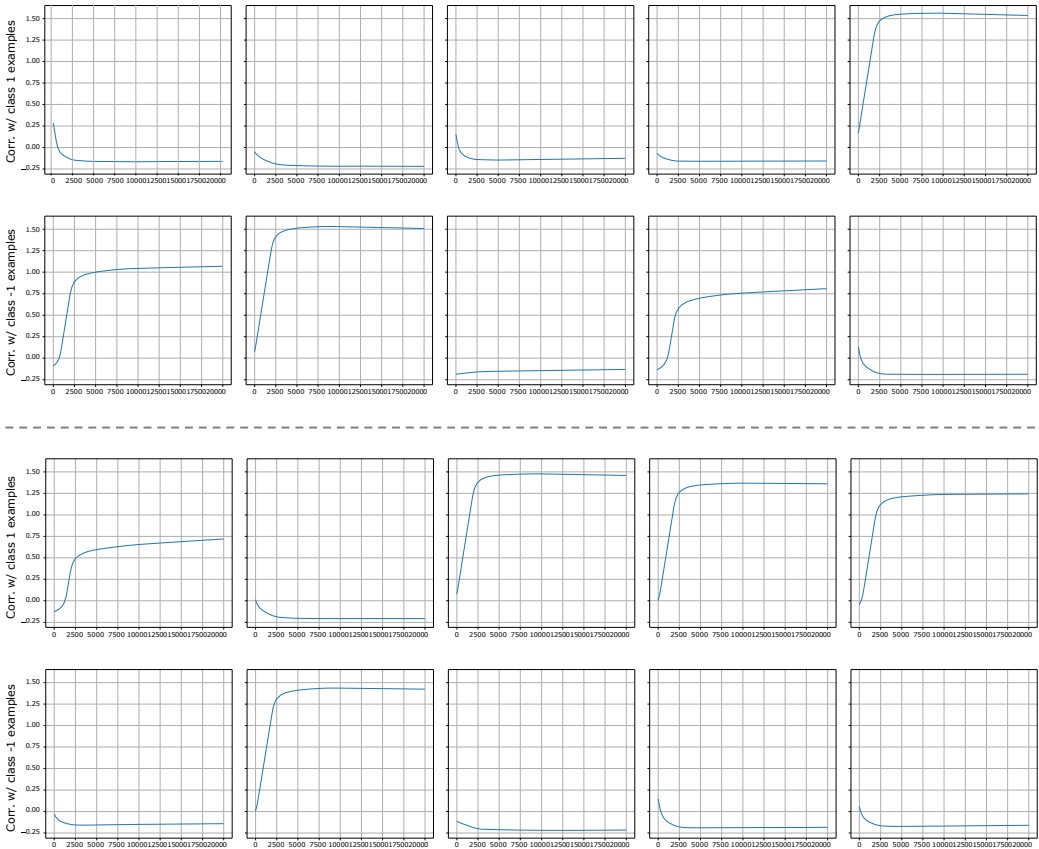

Figure 4: **(Activation asymmetry)** The average correlation between 10 random neurons and examples from both classes as a function of training iterations in the classification setting. In each column, the top plot above shows the average correlation between the weight (learned feature) of the neuron and the examples from class $y = 1$, while the bottom plot shows the average correlation between the weight (learned feature) of the neuron and the examples from class $y = -1$. As the training goes on, each neuron evolves to have positive correlation with at most one class of examples, which leads to activation asymmetry.

# F  ADDITIONAL EXPERIMENTS

## F.1  NUMERICAL EXPERIMENTS

In this section, we present the results of our numerical experiments. The numerical experiments were conducted with parameters $d_{\text{core}} = d_{\text{bg}} = 32$, $d = 256$, $m = 256$, and $N = 1000$. During training,

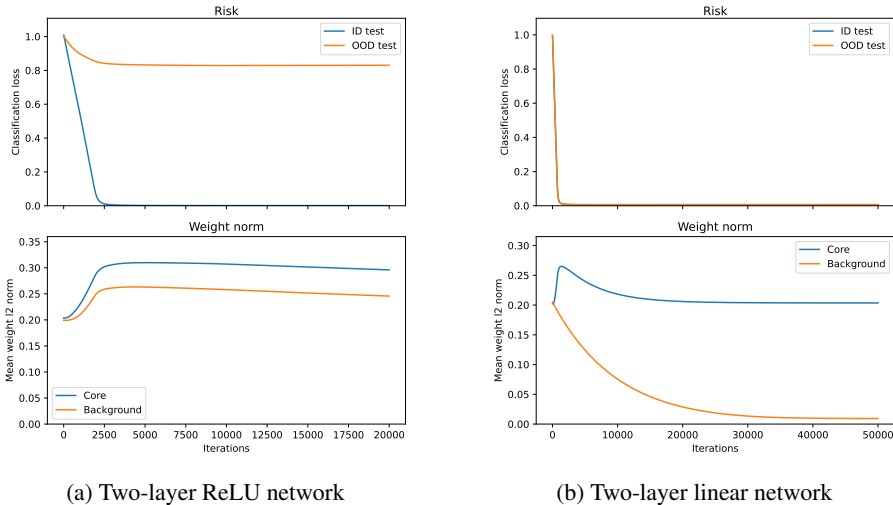

(a) Two-layer ReLU network

(b) Two-layer linear network

Figure 5: The ID and OOD risks (**top**) and the norm of weight projections onto core and background features (**bottom**) in the **classification** setting.

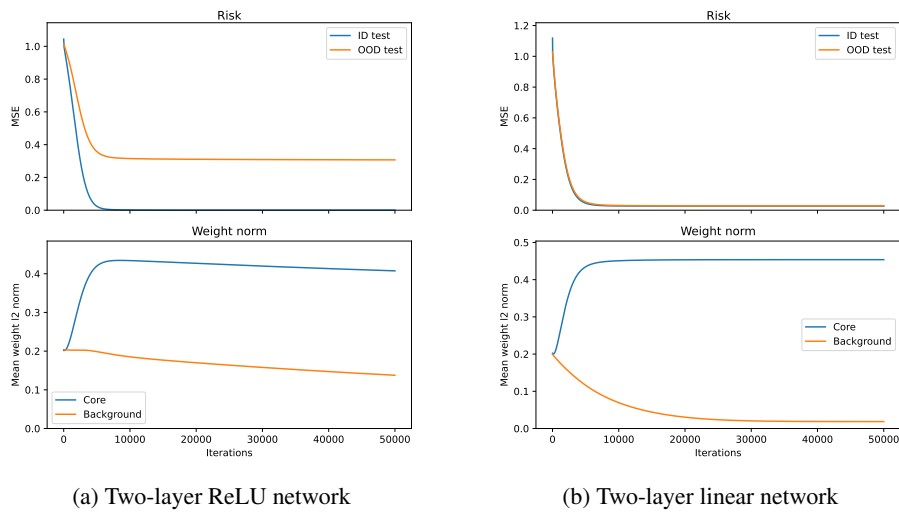

(a) Two-layer ReLU network

(b) Two-layer linear network

Figure 6: The ID and OOD risks (**top**) and the norm of weight projections onto core and background features (**bottom**) in the **regression** setting.

each $\mathbf{z}_i$, $i \in [d_0]$ was sampled from the uniform distribution on its support $[0, 1]$; during testing, each $\mathbf{z}_i$, $i \in \mathcal{S}_{\text{core}}$ was sampled from the same distribution as in training, while each $\mathbf{z}_i$, $i \in \mathcal{S}_{\text{bg}}$ was sampled from the uniform distribution on $[-1, 0]$. We considered two experimental settings:

- **Classification:** We trained a two-layer ReLU network to predict the binary label for each input, which matches our theoretical setting in Section 4. As an ablation, we also trained a two-layer linear network for the same task, replacing the ReLU functions in the network by identity functions.

- **Regression (representation distillation):** We trained a two-layer ReLU network to predict the vector $(\mathbf{z}_i)_{i \in \mathcal{S}_{\text{core}}}$ for each input—note that this is an optimal representation for OOD generalization, which matches the setting as our real-world representation distillation experiments in Section 2. As an ablation, we also trained a two-layer linear network.

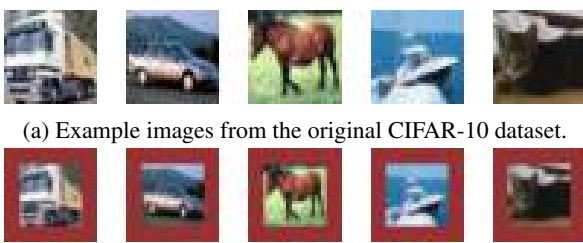

(a) Example images from the original CIFAR-10 dataset.

(b) Example images from the modified CIFAR-10 dataset.

Figure 7: Our modifications to CIFAR-10 by padding colored pixels for all images.

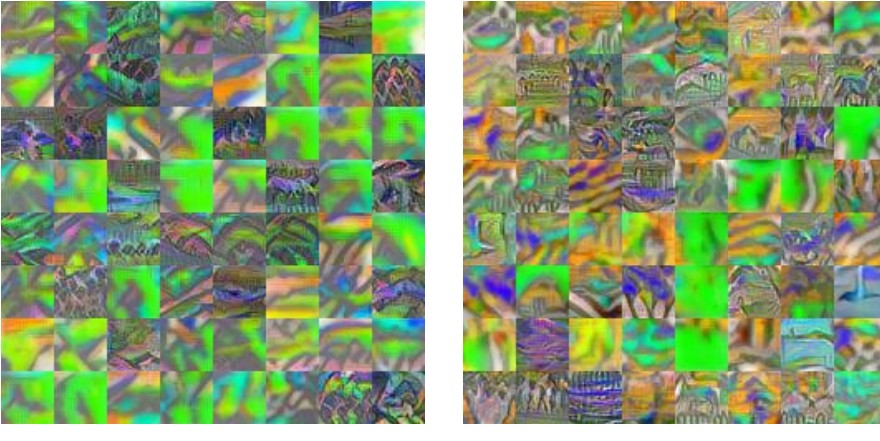

(a) Learned features of a ResNet-32 trained on the original CIFAR-10 dataset.

(b) Learned features of a ResNet-32 trained on the modified CIFAR-10 dataset.

Figure 8: Visualizations of the learned deep features.

In both settings, we trained the network using SGD with learning rate 0.001 and $\ell_2$ weight decay 0.01. The results are in Figure 4, Figure 5, and Figure 6, which corroborate our theoretical results on

- **Activation asymmetry:** as shown by Figure 4, each neuron evolves to have positive correlations with at most one class of examples during training.

- **Feature accompaniment happens for non-linear networks:** as shown by Figure 5a (classification) and Figure 6a (regression), two-layer ReLU networks indeed accumulate weight projections onto the background features during training, leading to small ID risk yet large OOD risk.

- **Feature accompaniment does not happen for linear networks:** as shown by Figure 5b (classification) and Figure 6b (regression), two-layer linear networks does not accumulate weight projections onto the background features during training, leading to both small ID risk and small OOD risk when the concept class is linearly separable.

## F.2 FEATURE VISUALIZATIONS ON A VARIANT OF CIFAR-10

To investigate the presence of feature accompaniment in real-world datasets, we conducted a experiment based on a variant of the CIFAR-10 dataset that is explicitly modified to incorporate background features that have *no correlation* with the label. Concretely, we augmented the CIFAR-10 training set by padding brick red pixels to the original images from CIFAR-10 and resized the padded images to the size of the original images, as shown in Figure 7. Since our padding does not impact the original image contents, it follows the "orthogonal" setting in our theoretical model where the core features (the original image contents) and the background features (the padded pixels) are independent.

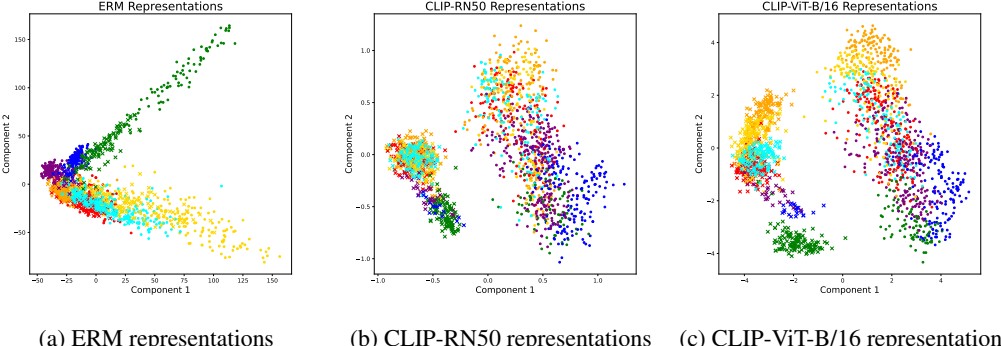

(a) ERM representations      (b) CLIP-RN50 representations      (c) CLIP-ViT-B/16 representations

Figure 9: Visualizations of ERM and CLIP representations after PCA dimensionality reduction to two dimensions. Circles refer to image representations in the training domains, while crosses refer to image representations in the test domain. Different colors represent different classes. Compared to ERM representations where the examples from training and test domains are visually mixed, CLIP representations exhibit strong *linear separability* of different domains.

We then visualize the learned features of a ResNet-32 network trained on the original CIAFR-10 dataset and another ResNet-32 trained on our modified dataset. Following the visualization technique in Allen-Zhu & Li (2021), we performed adversarial training using the method proposed by Salman et al. (2019) and visualized the features learned by the network's convolutional kernels in the 31st layer using the same hyperparameters as described in Allen-Zhu & Li (2021). As shown by Figure 8, we observe notable differences in the learned color information between models trained on the original CIFAR-10 dataset and its modified variant. Meanwhile, we note that there are no obvious geometric patterns in the red areas, which we conjecture is due to the augmentations used during training such as random cropping and flipping. In general, the visualization results suggest that background features are indeed learned by deep neural networks despite having no correlation with the label, which corroborates our theory and indicates that *feature accompaniment also happens in **deep features** learned from real-world image data*.

## G  EMPIRICAL EVIDENCE THAT SUPPORTS CONJECTURE 1

In this section, we provide preliminary empirical evidence that supports Conjecture 1 stated in the main text. For ease of presentation, here we restate this conjecture:

**Conjecture 1.** *Pre-training on a sufficiently large and diverse dataset alleviates feature accompaniment and leads to more linearized representations, hence improving OOD genenralization.*

Table 5: Detailed ID test accuracy, OOD test accuracy, and domain classification error (%) of linear probes on pre-trained and distilled representations on PACS.

|  | ID Test Acc. | OOD Test Acc. | Domain Classification Error |
|---|---|---|---|
| CLIP-ViT-B/16 | 99.68 | 91.59 | 0.06 |
| CLIP-RN50 | 97.35 | 85.67 | 0.19 |
| ERM-RN50 | 99.28 | 76.47 | 1.02 |

### G.1  LARGE-SCALE PRE-TRAINING LEADS TO LINEAR SEPARABILITY OF DOMAINS

To empirically test this conjecture, we first examined the properties of the pre-trained representations from CLIP and the representations learned by ERM on a domain generalization dataset PACS (Li et al., 2017) for image classification. The images in PACS are divided into four domains, namely Photo, Art painting, Cartoon, and Sketch, with seven common categories. We trained a ResNet-50 ERM model using the examples from the first three domains (ID) and the Sketch domain was treated

as the OOD domain. To evaluate the robustness of CLIP representations, we fitted a linear probe on top of frozen CLIP representations on ID domains and evaluated the learned linear probe on the OOD domain.

We begin by a 2-dimensional visualization of both the learned ERM representations and the CLIP representations using PCA dimensionality reduction. As shown in Figure 9, *ERM representations and CLIP representations exhibit quite different properties in terms of domain separability*: while examples from training and test domains are visually mixed in ERM representations, examples from training and test domains are *strongly linearly separable* in CLIP representations.

We then quantitatively examined this linear separability by fitting linear classifiers on top of ERM and CLIP representations for *domain classification*. Concretely, we trained linear classifiers with the original "class" label of each example substituted by its domain index. We then evalute the accuracy of this classifier on a hold-out validation set. As shown in Table 5, domain classifiers on CLIP representations have considerably smaller error than domain classifiers on ERM representations, which is consistent with visualization. This phenomenon is related to recent work on unsupervised domain adaptation based on contrastive learning (Shen et al., 2022; HaoChen et al., 2022), where it has been shown that contrastive learning can learn representations that disentangle domain and class information, enabling generalization that they refer to as "linear transferability" (HaoChen et al., 2022). However, their analysis requires that unlabeled examples from the target domain are seen by the contrastive learning algorithm during training, while large-scale pre-training in our context seems to achieve a similar disentangling effect even without explicitly trained on the target distribution. Further theoretical explanations of this phenomenon is an important future work.

In summary, the results in this section suggest that the representations learned by large-scale pre-training is highly linearized, with features representing different factors of variation not as non-linearly coupled as in our analysis on feature accompaniment. We believe that such high linearity of representations plays a critical role in the OOD capability of pre-trained models.

### G.2    LARGE-SCALE PRE-TRAINING LEADS TO DENSER NEURON ACTIVATION

In this section, we study property of pre-trained representations from another angle of neuron activation. As we have formally proved in Section 4, feature accompaniment causes the neurons to learn non-linearly coupled features. The activation of each neuron is thus likely to involve multiple feature vectors due to this coupling. By the above deduction, if pre-training alleviates feature accompaniment and learns more linearized features, then the activation of different feature vectors would be more likely to involve different neurons, resulting in an increase in the total number of activated neurons for each input.

Empirically, we confirmed the above hypothesis by calculating the histogram of the neuron's expected activation value in pre-trained and distilled models from the ImageNet experiments in Section 2. We considered the CLIP-RN50 teacher model and its corresponding student model obtained from representation distillation, and maintained an estimate of the average activation value for each output ReLU activation in the first residual block during one evaluation run. We plot the histogram of the neuron's average activation value in Figure 10. As shown by the figure, the pre-trained CLIP model indeed have considerably denser neuron activation than the distilled model, even on the ID ImageNet validation set where their top-1 accuracy is nearly the same (70.37% for the pre-trained CLIP model and 69.85% for the distilled model). This suggests that pre-trained models learn more "decoupled" features than models trained solely on the ID data.

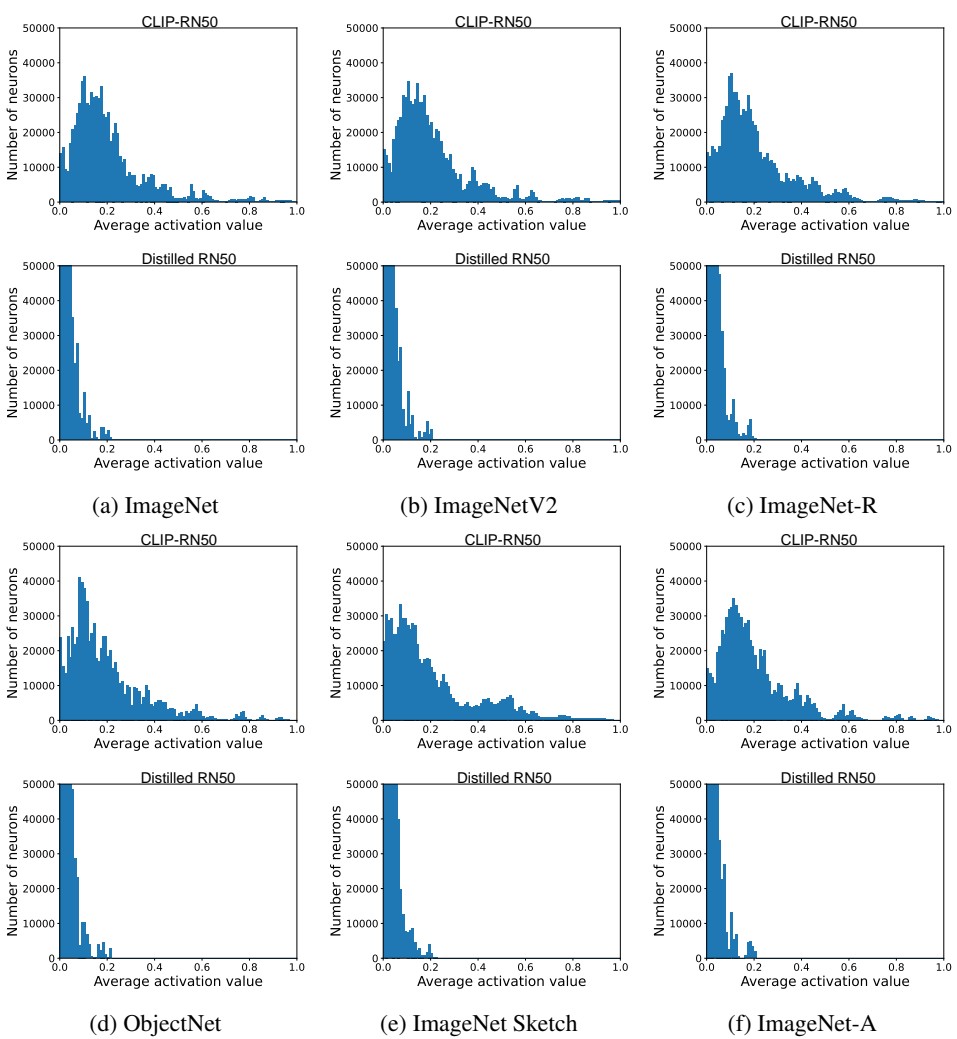

(a) ImageNet      (b) ImageNetV2      (c) ImageNet-R

(d) ObjectNet      (e) ImageNet Sketch      (f) ImageNet-A

Figure 10: Histograms of average neuron activations of both pre-trained CLIP models and distilled models on ImageNet-based distribution shift datasets. In each subfigure, the top plot shows the histogram of CLIP, and the bottom plot shows the histogram of the distilled model.

