# OpenReview forum: "Feature Accompaniment: Is It Feasible to Learn Out-of-Distribution Generalizable Representations with In-Distribution Data?"
_ICLR.cc/2024/Conference — Submitted to ICLR 2024_

### Official Review · Reviewer_jEjy · 2023-11-01

**Soundness:** 3 good
**Presentation:** 4 excellent
**Contribution:** 3 good
**Rating:** 6
**Confidence:** 4

**Summary:**

The paper considers the question of whether it is feasible to learn good representations for OOD generalization with only ID data, without considering inductive biases of the architecture and learning algorithm. First, the paper looks at an experiment where models are trained to learn the features of pretrained models that exhibit good OOD performance. It is found that these distilled models have OOD performance better than standard models only trained on the ImageNet training set, but not as good as the original pretrained models, suggesting that it is not possible to learn good OOD representations from ID data, even given access to “oracle” representations known to perform well OOD. Via theoretical analysis of 2-layer ReLU networks, the paper then unveils a novel failure mode of OOD generalization called feature accompaniment. This failure mode is shown theoretically to stem from inductive biases of nonlinear networks, and is absent in deep linear models.

**Strengths:**

* The paper identifies a novel and intuitive failure mode of out-of-distribution generalization, distinct from the prevailing attention on spurious correlations
* The paper provides principled theoretical foundations that prove the existence of this failure mode for 2-layer
* The takeaways of the study are applicable to future theoretical study of OOD generalization. In particular, the paper attempts to make the highly of-interest case that existing theoretical models of OOD generalization may not cover why OOD generalization failure happens in practice.

**Weaknesses:**

* The claim in Section 2 of the existence of an OOD generalization failure mode beyond the reach of generalization theory, and related to feature learning may not be fully justified by the empirical results in the section. Please see Question 1 below.
* The study does not suggest how one might make the findings actionable in an empirical setting to improve or predict OOD generalization ability. It is thus unclear how significant the results or the identified failure mode are.
* Relatedly, the paper does not make a case for how much OOD generalization failure is attributable to this failure mode in empirical settings, if any at all.

**Questions:**

Question 1:
I am unconvinced that the empirical results of section 2 imply the existence of a failure mode related to the feature learning process. The empirical result may not necessarily be due to nonlinear feature-learning dynamics in this experiment, but rather just that pre-trained CLIP models contain features covering a much larger data distribution than is captured by models distilled on ImageNet. In particular, if you distill a model from CLIP on the ImageNet training set, are some CLIP features that are not represented in ImageNet not likely to be left out? These features could still be helpful in OOD classification. For example, the OOD bird and car could have core features that look different from the core features seen in-distribution. Pretrained models may contain these features, while distilled models may not have learned them if they do not appear in the training set.


Question 2:
Could there be discussion on how feature accompaniment relates to the previous studies on simplicity bias and gradient starvation [1,2], which find that networks rely on simple features and ignore more complex features? In particular, work on Gradient Starvation [2] suggests that an increase in strength of a simpler feature inhibits the learning of other more complex features. Are these results contradictory to those suggested by feature accompaniment?



[1] The Pitfalls of Simplicity Bias in Neural Networks, https://arxiv.org/abs/2006.07710

[2] Gradient Starvation: A Learning Proclivity in Neural Networks, https://arxiv.org/abs/2011.09468

---

> ### Author Response · Authors · 2023-11-15
> **Response to Reviewer jEjy**
>
> Thank you for your detailed and insightful review! Your feedback has led to some fruitful discussions among the authors and is helpful for us to improve our paper. In general, there are three major concerns raised in your review:
>
> - Connections between feature accompaniment and the empirically observed OOD generalization failure and the significance of our results.
>
> - Can the empirical observations in Sec. 2 be explained by the fact that CLIP may have learned some OOD core features that are absent in the ID data?
>
> - Discussion on how feature accompaniment relates to the simplicity bias of neural networks.
>
> Since the first concern is common among all reviewers, we provide a general response to it in our "General response to all reviewers" panel, along with a discussion on the significance of our contributions. Please see our general response for more details.
>
> For the second concern, we believe that **our experimental results cannot be solely explained by CLIP having the ability to extract OOD core features due to large-scale pre-training**. This is due to our linear probe evaluation protocol: first train the linear probe on top of given (CLIP/distilled) representations _**on the training set (ID data)**_, then test its accuracy on both ID and OOD test sets. In other words, _this linear probe still needs to solve an OOD generalization task_---although with inputs as CLIP/distilled representations instead of raw images.
>
> - Having this in mind, now assume that the distribution shift robustness of CLIP is attributed to it having the ability to extract OOD core features from OOD images. If this is indeed the case, then the linear probe based on CLIP representations should **not** exhibit very good OOD generalization performance. This is because those OOD features are **absent** in the ID images (and hence also absent in the CLIP representations of ID images as well) and thus the linear probe trained using only ID data is **not** able to learn to use those OOD features to achieve OOD generalization.
>
> - As a piece of empirical evidence that supports the above explanation, we have visualized the CLIP representations for both ID and OOD images (see Fig. 9 in Sec. G for details). The visualization shows that ID and OOD representations exhibit the geometry where the distribution shift on the features is roughly in a direction **orthogonal** to the classification boundary, so it is not likely that the CLIP's advantage in robustness is due to learning "more" OOD features that provide additional information for prediction.
>
> For the third concern, we believe that feature accompaniment and simplicity bias are two **complementary** feature learning biases of neural networks that may manifest in different data distributions: both works [1] and [2] consider a data model involving multiple features that are **equally predictive** for the task; the network may then only learn some of them due to the simplicity bias. By contrast, the background features in our data model are assumed to be **not predictive at all** for the task, yet we theoretically show that the network may still learn them due to feature accompaniment---note that _this does not contradict the simplicity bias_ since the resulting solution can still be "simple", e.g., learning sparse core/background features. Instead, **we believe that feature accompaniment as a novel inductive bias of neural networks helps extend our general understanding of the feature learning biases of neural networks**. Technically, we would also like to note that the theory in [1] only considers a specific dataset (LSN dataset with one linear coordinate and one 3-slab coordinate) while we consider a more general data model, and the theory in [2] is under the neural tangent kernel regime, in which the network is essentially linearized and the features are **not** learned (they stay at random initialization).
>
> **We hope that you can consider raising the score if our response has addressed your concerns and we are willing to engage in further discussion if any of our explanations are still unclear.** We are looking forward to your reply.
>
> ---
>
> [1] The Pitfalls of Simplicity Bias in Neural Networks.
>
>
> [2] Gradient Starvation: A Learning Proclivity in Neural Networks.

---

> > ### Author Response · Authors · 2023-11-21
> > **A kind reminder**
> >
> > Dear Reviewer jEjy:
> >
> > As the author's response period is coming to an end, we would like to kindly query whether our response has addressed your concerns and we are willing to answer your further questions if necessary (since we cannot answer them after Nov 22nd). Thank you!
> >
> > Best,
> >
> > Authors of Submission1827

---

### Official Review · Reviewer_9qmH · 2023-11-02

**Soundness:** 3 good
**Presentation:** 3 good
**Contribution:** 4 excellent
**Rating:** 6
**Confidence:** 4

**Summary:**

This paper tries to study whether it is possible to learn OOD-generalizable representations with only in-distribution data. The authors discover a new failure model that they refer to as feature accompaniment, which is caused by the inductive biases of training process of nonlinear neural networks.

**Strengths:**

* This paper studies OOD from the perspective of inductive bias, which has rarely considered in existing literature. They consider the training process of neural network, which is more practical than directly considering the global minimum.
* From their theoretical analyses, they find an interesting failure mode ''feature accompaniment''. In my understanding, this means that due to the asymmetry of activation, each neuron tends to correlate more with one class than another. Then, this can further make the projection of gradients onto background features non-zero, which makes the final model also use background features to classify. I think this ''feature accompaniment'' may be a very fundamental phenomenon caused by the asymmetry of activation, which can also be used to understand other properties of neural network. I think my own research can draw some inspiration from it.

I hope to obtain more insights from upcoming discussions with the authors and I'm happy to further raise my score.

**Weaknesses:**

* I think the experiment part in Section 2 is a bit disconnected from the theoretical part in Section 4. They consider different settings and different learning objective. I don't think theory in Section 4 can explain the experimental results in Section 2. I know that Section 2 is probably just a starting point of studying whether OOD-generalizable representations are learnable, so this's okay. But I think it could be better to connect them more in the writing.
* The training process the authors mainly studied is based on ERM with regularization. Since there is only one training domain, this is okay. But I want to know if there are several domains whose background distributions are different, will the training process of Equation (1) (instead of ERM) still cause "feature accompaniment"? Are there any cases even if we have multiple domains we can still not learn a OOD-generalizable neural network?

**Questions:**

Please see Cons.

---

> ### Author Response · Authors · 2023-11-15
> **Response to Reviewer 9qmH**
>
> Thank you for your positive review and your approval of our work! We are very glad to know that you found our theoretical results novel, foundamental and may benefit understanding the learning process of neural networks in other scenarios. In general, your review has raised two major concerns:
>
> - The experiment part and the theoretical part seem a bit disconnected.
>
> - What will happen if we have **multiple** training domains (with different background feature distributions) instead of only one training domain?
>
> Since the first concern is common among all reviewers, we provide a general response to it in the "General response to all reviewers" panel, along with a discussion on the significance of our contributions. Please see our general response for more details.
>
> For the second concern, we further break it down into two questions as mentioned in your review:
>
>  1. Are there any cases where we cannot achieve OOD generalization even with multiple domains?
>
>  2. If we do have multiple domains and run some more advanced algorithms rather than ERM, will feature accompaniment still happen?
>
> For Q1, our answer is yes: note that "OOD generalization with multiple training domains" is typically studied in the context of _domain generalization_ (see [1,2] for some examples). Indeed, prior studies in this area [1,2] have pointed out that large OOD generalization gaps remain even with multiple training domains and more advanced algorithms than ERM---in fact, they show that under a fair evaluation protocol, even ERM is often comparable to many of the recently published OOD generalization algorithms! Also, our experiments in the DomainNet dataset (results are in the last panel of Fig. 1) are based on this domain generalization setup where we have three training domains and one test domain, yet the distilled model still underperforms OOD.
>
> For Q2, we believe the answer is also yes for the following reasons:
>
> - As we mentioned above, current algorithmic improvement over ERM is still limited---we believe this fact indicates that other objectives may also suffer from similar generalization difficulty as ERM.
>
> - Our empirical evidence indicates that feature accompaniment is a general phenomenon that also happens beyond the binary classification setting used by our theory (see our general response for more details). We thus believe that feature accompaniment can also happen for many objectives other than ERM.
>
> - For the particular objective mentioned in your review, i.e., directly minimizing Eq. (1) (note that in practice we need to substitute $\mathbb{D}$ with $\mathbb{D}_\mathrm{train}$ since we only have access to the training domains), our analysis can directly apply: Proposition 2 in [3] proves that minimizing this objective is equivalent to minimizing a weighted mixture of the risk in each training domain, which is then equivalent to performing ERM on the corresponding weighted data distribution. Our results then trivially hold if this weighted training distribution satisfies Def. 1. Empirically, our experiments in the DomainNet dataset also consider this objective as one of the standard models (referred to as "GroupDRO"; please see Sec. E.4 for more details), which does not yield good OOD generalization performance.
>
> **We hope that you can consider raising the score if our response has addressed your concerns and we are willing to engage in further discussion if any of our explanations are still unclear.** We are looking forward to your reply.
>
> ---
>
> [1] I. Gulrajani and D. Lopez-Paz. In search of lost domain generalization. In ICLR, 2021.
>
>
> [2] P. W. Koh et al. Wilds: A benchmark of in-the-wild distribution shifts. In ICML, 2021.
>
>
> [3] M. Arjovsky, L. Bottou, I. Gulrajani, and D. Lopez-Paz. Invariant risk minimization. arXiv preprint arXiv:1907.02893, 2019.

---

> > ### Author Response · Authors · 2023-11-21
> > **A kind reminder**
> >
> > Dear Reviewer 9qmH:
> >
> > As the author's response period is coming to an end, we would like to kindly query whether our response has addressed your concerns and we are willing to answer your further questions if necessary (since we cannot answer them after Nov 22nd). Thank you!
> >
> > Best,
> >
> > Authors of Submission1827

---

### Official Review · Reviewer_Lh7e · 2023-11-10

**Soundness:** 2 fair
**Presentation:** 3 good
**Contribution:** 2 fair
**Rating:** 5
**Confidence:** 4

**Summary:**

This work tries to answer "Can we learn OOD generalizable representations from in-distribution data?" empirically and theoretically.

In the empirical part, the term **OOD generalizable representation" mainly indicates a representation that contains rich features. The author investigates the OOD linear probing performance of three kinds of pretrained models: 1) a CLIP pretrained model on super large&diverse dataset; 2) a supervised pretrained model on Imagenet dataset; 3) a supervised pretrained model on Imagenet dataset with more objective information (i.e. prediction the representation of a CLIP model).

The author treats the 3rd model as the oracle objective function --- *"representation learning objective itself cannot be further improved in general"*. Hence concludes that *"OOD generalizable representations may not be learnable using only ID data without explicitly taking into account the inductive biases of the model or the task."*

In the theoretical part, however, the term **OOD generalizable representation** changes to indicate "a representation that doesn't contain spurious signals (or background feature signals) and only contains invariant signals (or core feature signals). The author uses a 2-layers Relu network to show that --- a non-convex network (especially with asymmetric activations) could "learn and store" some background feature signals in the representation even though these background features have no correlation with the target label.

**Strengths:**

- It is interesting to investigate out-of-distribution generalization problem through the rich-representation (a representation contains a rich set of features that could be redundant in-distribution but crucial out-of-distributation) point of view.

- It is also interesting to show that a non-convex network could "learn and store" some irrelative signals (per-example level spurious features) in the representation even though these signals are not (or weakly) correlated with the target label in the whole-dataset level.

**Weaknesses:**

- As I commented in the **Summary**, the empirical part and theoretical part use different principles. So that they can not support each other. Please check **Summary** for details. In my opinion, that is the biggest weakness.

- In the empirical part, this work treats "good OOD linear probing performance" as "good generalization representation" (Figure 1). The principle here is "rich-representation"[1][2]. I suggest the author clarify the principle.

- The author treats -- a supervised pretrained model on Imagenet dataset with more objective information (i.e. prediction the representation of a CLIP model) -- as the **oracle** objective function. By comparing this model (pretrained on Imagenet) with CLIP (pretrained on a large dataset), the author concludes that *"OOD generalizable representations may not be learnable using only ID data without explicitly taking into account the inductive biases of the model or the task."*

On one hand, this comparison doesn't support the conclusion. From the rich-representation's principle (which is used in the linear probing experiment), OOD linear probing benefits from a representation that contains diverse and simple features. Indeed, CLIP (pretrained on a large dataset) contains rich features. But please remember that CLIP uses more data. It is possible that the model  above (pretrained on Imagenet) is already the best (by say "best", I mean a model that achieves the best OOD linear probing performance) imagenet pretraining model. In short, CLIP model (pretrained on a large dataset) should not be assumed as an achievable upper bound of other Imagenet pretrained model.

On the other hand, this Imagenet pretrained model is not **oracle** in terms of rich-representation. Compared with Imagenet's 1k target categories, indeed this object contains more supervision information (with the help of CLIP and CLIP's pretraining dataset). But how about 22k target categories, for example?

- The theoretical section didn't discuss the relationship between works about SGD and features, e.g. [3][4][5].

[1] Zhang, J., Lopez-Paz, D., & Bottou, L. (2022, June). Rich feature construction for the optimization-generalization dilemma. In International Conference on Machine Learning (pp. 26397-26411). PMLR.
[2] Zhang, J., & Bottou, L. (2023, July). Learning useful representations for shifting tasks and distributions. In International Conference on
[3]Andriushchenko, M., Varre, A. V., Pillaud-Vivien, L., & Flammarion, N. (2023, July). Sgd with large step sizes learns sparse features. In International Conference on Machine Learning (pp. 903-925). PMLR.
[4]Blanc, G., Gupta, N., Valiant, G., & Valiant, P. (2020, July). Implicit regularization for deep neural networks driven by an ornstein-uhlenbeck like process. In Conference on learning theory (pp. 483-513). PMLR.
[5]Pezeshki, M., Kaba, O., Bengio, Y., Courville, A. C., Precup, D., & Lajoie, G. (2021). Gradient starvation: A learning proclivity in neural networks. Advances in Neural Information Processing Systems, 34, 1256-1272.

**Questions:**

- please see weaknesses

---

> ### Author Response · Authors · 2023-11-15
> **Response to Reviewer Lh7e**
>
> Thank you for your detailed and constructive review! We have found your feedback very helpful in improving the quality of our work. However, _**we believe that there is a key misunderstanding of our experimental protocol in your review, which may have led to some misinterpretations of our work**_. We apologize if it was our current writing that caused the misunderstanding and will clarify our experimental protocol according to your feedback to avoid potential misunderstanding from future readers.
>
> - Your review mentioned that _"In the empirical part, this work treats 'good **OOD linear probing** performance' as 'good generalization representation'. The principle here is 'rich-representation' [1,2]."_ We are aware of the insightful works by Zhang et al. [1,2] and would like to clarify that **our evaluation protocol is _not OOD linear probing_** (i.e., training and testing the linear probe on both OOD data, similar to the linear probing protocol in [2]), **but _OOD testing of the linear probe_** (i.e., training the linear probe on **ID** data and testing it on OOD data). Note that this is a **fundamental distinction**: in OOD linear probing, the linear probe essentially solves an **ID generalization** task since its training and testing are on the **same** distribution---in this case, the richness of the representation indeed plays a key role (under the theoretical framework of [2]). However, in our experiments, the linear probe still needs to solve an **OOD generalization** task, albeit with inputs as some pre-trained representations rather than raw images. In this setting, _better linear probing performance does not necessarily mean a 'richer' representation_ (see below for more discussion).
>
> - Given the above clarification, "an OOD-generalizable representation (oracle representation)" in our experiments should be viewed as "a representation based on which an ID-trained (linear) classifier can generalize OOD". In this regard, **the CLIP representation is indeed a proper approximation of the "oracle representation"** since it has been known for exhibiting strong robustness to distribution shifts [3]. Our experiments thus **do support our conclusion** that "OOD-generalizable representations may not be learnable using only ID data".
>
> We then address your other concerns regarding (1) the "rich-representation" interpretation of our empirical results and (2) comparison with other theoretical work on neural network features.
>
> - **On the rich-representation interpretation:** as mentioned in your review, a potential explanation of the generalizability of the CLIP representation is its richness (in the sense that it _"contains a rich set of features that could be redundant in-distribution but crucial out-of-distribution"_, taken from your review). We **do agree** that this representation richness contributes to generalizability and hypothesize that this may be one of the reasons for the distilled model outperforming standard models. However, _if the generalizability of CLIP can be solely attributed to its richness, then the distilled model _should also achieve good OOD generalization_ since it **directly learns**_ this rich representation during training (instead of learning the representation by a _proxy_ classification task, which may fail to capture _all_ useful features, as shown by [1,2]). However, our experiments suggest that this is **not** the case. Instead, we believe that **feature accompaniment can explain the remaining large gap between distilled models and CLIP**; please see our general response for more details.
>
>
> - **Comparison with work on neural network features:** To our knowledge, our discovery of feature accompaniment is novel, and **no** existing theoretical study on SGD-trained neural networks has explored this feature learning bias. As shown by the cited works in your review and **Reviewer jEjy**, one of the major focuses of existing theory on the inductive biases of neural networks is the **simplicity bias**, i.e., the bias of finding solutions that are "simple" or sparse. Feature accompaniment **complements existing observations** by showing that neural networks also have the bias of learning "_irrelevant_ features" in certain tasks (note that this does not contradict with sparsity).
>
> Again, we are very thankful for your feedback and will add the discussion on the rich-representation to the revised paper.
> **We hope that you can consider raising the score if our response has addressed your main concerns and we are willing to engage in further discussion if any of our explanations are still unclear.** We are looking forward to your reply.
>
> ---
>
> [1] Zhang, J., Lopez-Paz, D., & Bottou, L. Rich feature construction for the optimization-generalization dilemma. In ICML, 2022.
>
>
> [2] Zhang, J., & Bottou, L. Learning useful representations for shifting tasks and distributions. In ICML, 2023.
>
>
> [3] A. Radford et al. Learning transferable visual models from natural language supervision. In ICML, 2021.

---

> > ### Author Response · Authors · 2023-11-21
> > **A kind reminder**
> >
> > Dear Reviewer Lh7e:
> >
> > As the author's response period is coming to an end, we would like to kindly query whether our response has addressed your concerns and we are willing to answer your further questions if necessary (since we cannot answer them after Nov 22nd). Thank you!
> >
> > Best,
> >
> > Authors of Submission1827

---

> > ### Comment · Reviewer_Lh7e · 2023-11-23
> >
> > Thanks for your answer!
> >
> > The linear probing here is trained on ID and evaluated on both ID and OOD. I got you. Thank you.
> >
> > Let's list the experiments in this paper (before the theory part):
> > 1) CLIP representation, trained on large & diverse data, achieves the best OOD performance.
> > 2) Supervised representation, trained on ID data (imagenet) and imagenet 1k labels, achieves the worst OOD performance.
> > 3) Supervised representation, trained on ID data (imagenet) and CLIP generated continuous targets (so called representation), locates in between.
> > 4) During the comparisons above, ID performance is aligned.
> >
> > I agree with your comment **"the CLIP representation is indeed a proper approximation of the "oracle representation""**. We can treat CLIP representation in 1) as a good oracle representation. That is totally fine.
> >
> > My question in my previous review is about **"oracle objective function"**. The paper treats 3) as an oracle objective function. (e.g.  - *"representation learning objective itself cannot be further improved in general"*.) The paper further concludes the main point *"OOD generalizable representations may not be learnable using only ID data without explicitly taking into account the inductive biases of the model or the task."* based on this "oracle objective function" assumption. Is it an oracle objective function? What if I use both imagenet 1k labels and this CLIP generated continuous targets?
> >
> > On the other hand, the ID (in-distribution) concept should include both inputs (e.g. images) and targets (e.g. labels). In method 3), the target is changed, so it is not *ID data* any more...

---

> ### Author Response · Authors · 2023-11-23
> **Continual response to Reviewer Lh7e**
>
> **Thank you for your continual engagement! We are glad that we have solved your concerns regarding our linear probing protocol and the comparison between our protocol and the rich-representation principle after our clarification.** We address your remaining concerns in the following:
>
> - **On the "oracle" objective:** Thank you for your clarification on your comment. Yes, we believe that our representation distillation objective **is** "oracle" in terms of representation learning (if we do not have additional prior knowledge of the model/the task and thus treat the model as a black box). We can actually argue this mathematically: consider each model $h$ to be composed of an encoder $\Phi$ and a (linear) classifier $f$. Given each input $x$ (e.g., an image), we treat $\Phi(x)$ as **the representation of $x$** and $f$ as the classifier on top of the representations extracted by $\Phi$. Given a training set $(X,Y) = {(x_i,y_i)}_{i=1}^N$, **the aim of an ideal representation learning process is that the learned representation $\Phi(x_i)$ recovers some oracle representation** $\Phi^*({x_i})$ **for every** $x_i \in X$, where we approximate $\Phi^*$ using CLIP. In this regard, our representation distillation objective (i.e., $\mathrm{minimize}\  \lVert \sum_i \Phi(x_i) - \Phi^*(x_i) \rVert_2^2$ ) is indeed consistent with the above idealization since its minimizer does uniquely recovers $\Phi^*(x)$ for every $x$---we thus refer to it as the **oracle objective** in the above sense. **By contrast**, existing objectives perform representation learning using proxy tasks such as a classification objective defined over the final output of $h(x)$, or other auxiliary objectives based on $h(x)$ and/or $\Phi(x)$. However, minimizing such objectives does **not** guarantee the unique recovery of $\Phi(x) = \Phi^*(x)$ as guaranteed by our oracle objective. Of course, even by minimizing the oracle objective, we cannot guarantee $\Phi = \Phi^*$ since we only have $\Phi(x) = \Phi^*(x)$ for a finite training set, which leads to the remaining gap between the distilled model and CLIP---our theoretical results suggest that feature accompaniment plays an important role in this gap when $\Phi$ is instantiated as a non-linear neural network trained by SGD.
>
>    **Empirically**, in our preliminary experiments, we have also tested using both the original labels and the CLIP representations in representation learning, as the ablation mentioned in your review. Our results suggested that this variant indeed does **not** outperform only using CLIP representations. Due to time limitations, we will add those results to the camera-ready version of the paper if it is accepted.
>
> - **On the definition of ID and OOD:** This is a good point! Yes, strictly speaking, the notion of "distribution" in both "ID" and "OOD" is the _joint distribution_ over $\mathcal{X}\times\mathcal{Y}$. However, in a practical OOD generalization regime (and also in existing benchmarks), "ID" and "OOD" often essentially stem from the distribution difference over the **input space** $\mathcal{X}$ since the ground-truth labeling function (i.e., $p(y|x)$) is often assumed to be consistent across different distributions. Therefore, we consider the notions "ID data" and "OOD data" mainly as "ID inputs" and "OOD inputs" but not their (conditional) label distributions. Note that this setting is natural and consistent with existing algorithmic studies since many existing OOD generalization algorithms also construct auxiliary objectives (e.g., by using domain labels [1, 2, 3, 4]) other than the original labels.
>
> We hope that the above clarification can solve your concerns. Also, **we would like to note that the theoretical part constitutes a more significant part of the overall contributions of our work beyond the experiments** (please see our general response for a discussion on the significance of our contributions).
>
> ---
>
> [1] Y. Ganin et al. Domain-adversarial training of neural networks. Journal of Machine Learning Research, vol. 17, no. 59, pp. 1–35, 2016.
>
> [2] D. Krueger et al. Out-of-distribution generalization via risk extrapolation (rex). In ICML, 2021.
>
> [3] M. Arjovsky, L. Bottou, I. Gulrajani, and D. Lopez-Paz. Invariant risk minimization. arXiv preprint arXiv:1907.02893, 2019.
>
> [4] A. Rame, C. Dancette, and M. Cord. Fishr: Invariant gradient variances for out-of-distribution generalization. In ICML, 2022.

---

### Author Response · Authors · 2023-11-15
**General response to all reviewers (1/3)**

Dear reviewers,

We sincerely appreciate your insightful feedback on our work! Meanwhile, we notice that there are some common concerns among the reviewers, in particular regarding the following two points:

- Connections between our theory and experiments.

- The significance of our empirical and theoretical results.

 In the following, we provide a general response to the above common concerns. **For targeted response to individual concerns raised by each reviewer, please refer to the panel under each review.**

---

> ### Author Response · Authors · 2023-11-15
> **General response to all reviewers (2/3)**
>
> ## Our experiments and theory are consistent
>
> As mentioned by the reviewers, our experiments and theory use different settings:
>
>  - Our experiments are based on a representation distillation setting where we first train a model by distilling the "oracle" representations (approximated by CLIP representations), then perform classification by training a linear probe on top of the distilled representations.
>
>  - Our theory is derived in a binary classification setting based on a structured data generation model with core/background features.
>
> Hence, a natural question is whether our experiments and theory are disconnected. We believe that this is also partially due to our current presentation, where we opt to present the empirical results separately since we believe that those results are of interest by themselves and that they call for theoretical analyses on the inductive biases in OOD generalization, which naturally elicits our theory. Here we would like to emphasize that **our experiments and theory are consistent** in the following sense: _(1) our theoretical finding also holds in our empirical setting, and (2) our empirical observations can be explained by feature accompaniment but **_not_** by existing theory/explanations in OOD generalization_. We apologize that the current writing may be a bit confusing on this important point, which may have led to some misunderstandings. In the following we will clarify this:
>
>  - **Our theoretical finding of feature accompaniment also extends to our empirical setting (representation distillation).** While our theorems are formally proved only in binary classification (which is quite common in theoretical studies since it is often more amenable to theoretical analysis than distillation), _we empirically observed that feature accompaniment does hold both in classification and representation distillation settings, as shown by our numerical experiments in Sec. F.1_. We also noted this in footnote 6 (footnote 5 in the revised paper) in the main text.
>
>  - **Feature accompaniment explains our empirical observations while existing explanations cannot.** Since we have no access to the generation model of natural image data, it is very hard to formally attribute the observed empirical OOD failure in existing benchmarks to any theoretically established failure mode (this is also why most studies in OOD generalization have to _assume_ a data model to begin with). Nonetheless, we have provided **additional empirical evidence** supporting our explanation based on feature accompaniment for the observed OOD failure in our experiments:
>
>     - In Fig. 2, we visualize the prediction heatmaps of the linear probes on CLIP representations/distilled representations, revealing a novel "weakening effect" in the prediction heatmaps of the ID-distilled model on OOD data. In particular, our visualization also includes a _synthetic_ OOD scenario based on image style transfer, which **_closely matches our theoretical data model_** (keeping core features intact while _only_ changing background features). As shown in Fig. 2 (right), the prediction heatmaps on synthetic images exhibit visually similar weakening activation patterns as in those of natural OOD images, suggesting that _our theoretical data model indeed captures some important characteristics of real-world image data with distribution shifts_. Moreover, _the above weakening effect on prediction heatmaps matches the effect of feature accompaniment, i.e., the coupling of core and background features in neuron activations_ (see the discussion after Theorem 3 in the main text).
>     - In Sec. F.2, we visualize the _learned features_ of a deep convolutional network based on a variant of CIFAR-10 with manually injected label-irrelevant background features. Although this experiment does not follow the distillation setting as in Sec. 2, it indicates that feature accompaniment also happens for _deep features_ learned by practical neural networks as predicted by our theory.
>
>   - On the other hand, **the above empirical evidence cannot be directly explained by existing explanations of OOD failure** such as spurious correlations (discussed in Sec. 2). As raised by **reviewer Lh7e** and **reviewer jEjy**, another possible explanation here is the "richness" of representations---we have also discussed it in the response and argue that this viewpoint also fails to explain our empirical observations; please refer to our individual response for details.
>
> Given the above evidence, while we cannot claim that feature accompaniment is the _only_ reason for the empirical OOD failure, **we do believe that feature accompaniment indeed plays a key role in the observed OOD failure in our experiments**.
>
> We are grateful to all reviewers for letting us know the existing disconnection between experiments and theory in the current paper; we are working on revising our paper to make them more connected and will upload the revised paper once we finish the revision.

---

> ### Author Response · Authors · 2023-11-15
> **General response to all reviewers (3/3)**
>
> ## Significance of our contributions
>
>
> Our work has both experimental and theoretical parts and is therefore dense in content. Due to space limitations, the significance of some of our contributions may not be discussed in the current paper, which we believe may have led to some confusion and misunderstandings. Therefore, in the following, we would like to add a more detailed discussion on the significance of our contributions.
>
> - **Significance to OOD generalization theory:** As a novel failure mode in OOD generalization, feature accompaniment is **provably** due to the non-linear feature learning process of neural networks, being **one of the very first results** that take into account the inductive biases of SGD-trained neural networks in OOD generalization. This contrasts to the majority of prior theoretical work based on linear models, which is often _over-simplified_ and may not reflect some important facets of OOD generalization in practice.
>
>
> - **Significance to OOD generalization in practice:** In the OOD generalization community, it is often **assumed** that _it is spurious correlations that cause (or at least play a critical role in) the OOD generalization failure of machine learning models_: for example, many pitoval algorithmic works in the recent surge of OOD generalization research (such as [1,2,3] and their follow-ups) are based on the spurious correlation formulation, and so are most of recent theoretical studies (such as [4, 5] and many others). In the introduction of many influential works we can see statements like: _"...failing to generalize out-of-distribution is failing to capture the causal factors of variation in data, clinging instead to easier-to-fit spurious correlations" [6]_. However, **our empirical results challenge this common belief**, suggesting there is a large OOD generalization gap that _cannot_ be explained by spurious correlations (or other existing explanations) in existing benchmarks. By contrast, we provide empirical evidence supporting the explanation based on feature accompaniment, suggesting that **feature accompaniment is not a pure theoretical finding** and it also plays a key role in the OOD generalization failure observed in practice. While our work does not provide a direct solution to this failure mode, we believe that our insights can benefit future algorithm design in OOD generalization. For example, in Sec. G.2 we empirically show that feature accompaniment can lead to more concentrated neuron activations compared to CLIP; it can then inspire designing activation-level regularizers when fine-tuning pre-trained models.
>
>
> - **Significance to general neural network theory:** To the best of our knowledge, even outside the context of OOD generalization, there is **no prior theoretical work** that explores the feature accompaniment phenomenon in neural networks. Therefore, at a high level, _our theoretical finding of feature accompaniment as a **novel inductive bias of SGD-trained neural networks** can serve as a new perspective for understanding the existing success/shortcomings of deep learning_, complementing known inductive biases of neural networks such as the "simplicity bias" as mentioned by **reviewer Lh7e** and **reviewer jEjy**. As pointed out by **reviewer 9qmH**, feature accompaniment may also be used to understand the properties of neural networks in different settings beyond OOD generalization.
>
> ---
>
> [1] M. Arjovsky, L. Bottou, I. Gulrajani, and D. Lopez-Paz. Invariant risk minimization. arXiv preprint arXiv:1907.02893, 2019.
>
>
> [2] D. Krueger et al. Out-of-distribution generalization via risk extrapolation (rex). In ICML, 2021.
>
>
> [3] S. Sagawa, P. W. Koh, T. B. Hashimoto, and P. Liang. Distributionally robust neural networks for group shifts: On the importance of regularization for worst-case generalization. In ICLR, 2020.
>
>
> [4] E. Rosenfeld, P. Ravikumar, and A. Risteski. The risks of invariant risk minimization. In ICLR, 2021.
>
>
> [5] V. Nagarajan, A. Andreassen, and B. Neyshabur. Understanding the failure modes of out-of-distribution generalization. In ICLR, 2021.
>
>
> [6] I. Gulrajani and D. Lopez-Paz. In search of lost domain generalization. In ICLR, 2021.

---

### Author Response · Authors · 2023-11-18
**Paper revision**

Dear reviewers,

We have uploaded a revision of our paper according to your feedback, incorporating the major points in the author's response. In general, we add more discussion on the connections between our theory and experiments and highlight the significance of our results. We sincerely hope that you can reevaluate our contributions if we have addressed your main concerns. The main changes are as follows, highlighted in orange in the revised paper:

- In Section 1.1, we add more discussion on the significance of our empirical and theoretical results.

- In Section 2, we add more detailed explanations of our experimental protocol and more discussion on the alternative explanations of our empirical results.

- In Section 3, we add a discussion on the connection between our theoretical data model and our experiments.

- In Section 4, we add more discussion on the connection between feature accompaniment and our empirical observations and refine the previous discussion on the explanations of our visualization in Figure 2.

Thank you again for the great efforts made in reviewing our work; we would greatly appreciate it if you could let us know whether our response and the revised paper have addressed your concerns and we will be very happy to engage in further discussion.

---

### Meta-Review · Area_Chair_JWYn · 2023-12-14

**Metareview:**

This paper introduces the notion of "feature accompaniment", a potential obstacle to learning representations which generalize to out-of-distribution data in models trained via stochastic gradient descent. The results are supported by both a theoretical and empirical analysis.

Overall, this paper shows a strong contribution. However there were a number of difficulties encountered by reviewers in the initial submitted version of the paper, including: understanding the link between the theoretical and experimental results; the precise setting for the empirical studies; potential explanations for the empirical results due to differences in dataset sizes and tasks; and overall practical considerations and impact.

To quite some degree these have been addressed in the author comments and the revised draft. However, without significant engagement from the reviewers to evaluate the new sections and the new draft, then although borderline in terms of scores the paper should probably not be accepted to ICLR at this time, and instead should go through another round of review.

**Justification For Why Not Higher Score:**

There were various concerns from the reviewers regarding the relation between experiments and theory, as well as the experimental setting. While these may have been addressed in the revision, they probably require further review.

**Justification For Why Not Lower Score:**

N/A

---

### Decision · Program_Chairs · 2024-01-16

Reject